# Mirror Descent Under Generalized Smoothness

**Dingzhi Yu** [1 2] **Wei Jiang** [3] **Hongyi Tao** [1 2] **Yuanyu Wan** [4 5] **Lijun Zhang** [1 2]

## Abstract

Smoothness is crucial for attaining fast rates in first-order optimization. However, many optimization problems in modern machine learning involve non-smooth objectives. Recent studies relax the smoothness assumption by allowing the Lipschitz constant of the gradient to grow with respect to the gradient norm, which accommodates a broad range of objectives in practice. Despite this progress, existing generalizations of smoothness are restricted to Euclidean geometry with $\ell_2$-norm and only have theoretical guarantees for optimization in the Euclidean space. In this paper, we address this limitation by introducing a new $\ell*$-smoothness concept that measures the norm of Hessians in terms of a general norm and its dual, and establish convergence for mirror-descent-type algorithms, matching the rates under the classic smoothness. Notably, we propose a generalized self-bounding property that facilitates bounding the gradients via controlling suboptimality gaps, serving as a principal component for convergence analysis. Beyond deterministic optimization, we establish sharp convergence for stochastic mirror descent, matching state-of-the-art under classic smoothness. Our theory also extends to non-convex and composite optimization, which may shed light on practical usages of mirror descent, including pre-training and post-training of LLMs.

## 1. Introduction

This paper considers the optimization problem $\min_{\mathbf{x} \in \mathcal{X}} f(\mathbf{x})$, where $f$ is a convex differentiable function defined on the domain $\mathcal{X}$. It is well-known that gradient descent converges at the rate of $O(1/\sqrt{T})$ for any Lipschitz continuous $f$ (Nesterov, 2013). If $f$ is smooth, i.e., has a Lipschitz continuous gradient, then a faster rate of $O(1/T)$ can be obtained. Furthermore, an optimal $O(1/T^2)$ rate is attained by incorporating acceleration schemes (Nesterov, 1983). However, optimization problems arising from modern machine learning (ML) are typically non-smooth (Defazio & Bottou, 2019). Even for standard objectives such as $\ell_2$-regression, the global smoothness parameter could still be unbounded (Gorbunov et al., 2025).

Recently, efforts have been made to bridge this gap between theory and practice. Zhang et al. (2020b) propose a generalized $(L_0, L_1)$-smooth condition, which assumes $\left\|\nabla^2 f(\mathbf{x})\right\|_2 \leq L_0 + L_1 \left\|\nabla f(\mathbf{x})\right\|_2$ for nonnegative constants $L_0, L_1$, on the basis of extensive experimental findings on language and vision models. Under this condition, they analyze the convergence of gradient clipping (Mikolov et al., 2012) and theoretically justify its advantages over gradient descent during neural network training. Later, Li et al. (2023a) propose $\ell$-smoothness which replaces the affine function specified by $(L_0, L_1)$ with an arbitrary non-decreasing sub-quadratic function $\ell(\cdot)$, i.e., $\left\|\nabla^2 f(\mathbf{x})\right\|_2 \leq \ell(\left\|\nabla f(\mathbf{x})\right\|_2)$. The mild condition on $\ell(\cdot)$ ensures compatibility with objectives in modern ML and sheds light on new algorithmic designs.

However, existing studies on generalized smoothness suffer from a critical limitation: the definition is limited to $\ell_2$-norm and thus can only relate to optimization algorithms based on gradient descent in the Euclidean space. In particular, none of the existing work has considered mirror descent (MD) (Beck & Teboulle, 2003), a powerful optimization algorithm tailored for non-Euclidean problems. Over the years, algorithmic adaptivity to diverse geometries has received tremendous success in traditional ML (Ben-Tal et al., 2001; Zhang et al., 2020c), reinforcement learning (Montgomery & Levine, 2016; Wang et al., 2019; Tomar et al., 2022; Lan, 2023; Alfano et al., 2025), network quantization (Ajanthan et al., 2021), over-parameterization regimes (Sun et al., 2022), watermarking diffusion models (Liu et al., 2023a), pretraining and post-training of LLMs (Xie et al., 2023; Munos et al., 2024; Zhang et al., 2025d;e; Wu et al., 2026). Given the indispensable role of MD in these advancements, a natural question arises:

---
[1]State Key Laboratory of Novel Software Technology, Nanjing University [2]School of Artificial Intelligence, Nanjing University [3]School of Computer Science and Engineering, Nanjing University of Science and Technology [4]School of Software Technology, Zhejiang University [5]Hangzhou High-Tech Zone (Binjiang) Institute of Blockchain and Data Security. Correspondence to: Lijun Zhang <zhanglj@lamda.nju.edu.cn>.

*Proceedings of the 43rd International Conference on Machine Learning*, Seoul, South Korea. PMLR 306, 2026. Copyright 2026 by the author(s).

*Table 1.* Summary of our main results.

| Algorithm | Convergence Rate | Type[*] | Convexity |
|---|---|---|---|
| Mirror Descent (Beck & Teboulle, 2003) | $O(1/T)$ (Theorem 3.5) | a&l | ✓ |
| Accelerated Mirror Descent (Lan, 2020) | $O(1/T^2)$ (Theorem 3.7) | l | ✓ |
| Optimistic Mirror Descent (Chiang et al., 2012) | $O(1/T)$ (Theorem 3.9) | a | ✓ |
| Mirror Prox (Nemirovski, 2004) | $O(1/T)$ (Theorem 3.10) | a | ✓ |
| Stochastic Mirror Descent (Nemirovski et al., 2009) | $\widetilde{O}(1/\sqrt{T})$ (Theorem 4.2) | l | ✓ |
| Composite Mirror Descent (Duchi et al., 2010) | $O(1/T)$ (Theorem 5.1) | g | ✗ |

[*] Types of convergence, either average-iterate (a), last-iterate (l), or gradient-mapping (g).

> Can we extend mirror descent under the generalized smoothness condition?

We answer this question affirmatively by devising a new notion of generalized smoothness called $\ell*$-smoothness, which is defined under an arbitrary norm rather than $\ell_2$-norm, with full coverage of the $\ell$-smoothness proposed by Li et al. (2023a). To be more specific, we provide a fine-grained characterization of the matrix norm by utilizing a general norm and its dual. Under the $\ell*$-smoothness condition, we establish the convergence of mirror descent, accelerated mirror descent, optimistic mirror descent, and mirror prox algorithms, all matching classic results under $L$-smoothness in deterministic convex optimization. The key ingredient in our analysis is to bridge the gradients' dual norm with the suboptimality gap. To this end, we establish a generalized self-bounding property under $\ell*$-smoothness (Lemma 3.4), which bounds the gradient's dual norm by the product of the suboptimality gap and the local Lipschitz constant. For different types of mirror descent algorithms, we employ various strategies to demonstrate that the suboptimality gaps along the optimization trajectory can be bounded by the initial one, which may be of independent interest.

Furthermore, we extend our methodology to stochastic convex optimization and non-convex composite optimization. For the former, we effectively manage the gradient norms by delving into the last-iterate behavior of stochastic mirror descent (SMD) and provide a simple proof based on the "chain-of-events" (Section 4.2) analysis. We derive a state-of-the-art high probability convergence of $O(\sqrt{\log(T)}/\sqrt{T})$, matching the rate for standard $L$-smooth functions. For the latter, we show that the standard $O(1/T)$ convergence for $L$-smooth functions also holds for $\ell*$-smooth function class.

The convergence rates for mirror descent methods are shown in Table 1. Our contributions can be summarized as follows.

1. We propose $\ell*$-smoothness, the first non-Euclidean generalized smoothness model that fully encompasses existing smoothness models, including $L$-smooth, $(L_0, L_0)$-smooth, and $\ell$-smooth conditions.

2. We establish sharp convergence guarantees in Table 1 for mirror descent and its variants under $\ell*$-smoothness for convex, non-convex, and stochastic regimes, all of which match the rates under classic settings.

3. We provide both theoretical and empirical justifications for our $\ell*$-smoothness model, with strong empirical evidence from LLM and CNN experiments showing its broad applicability in real-world ML practice.

**Related Work** We briefly introduce two seminal works on generalized smoothness, with the comprehensive literature review deferred to Appendix A. Zhang et al. (2020b) first observe in LSTM and ResNet training that the loss functions satisfy an affine bound on the Hessian norm, which they formalize as $(L_0, L_1)$-smoothness. Under this assumption, they show that simple gradient clipping finds an $\epsilon$-stationary point in $O(\epsilon^{-2})$ iterations for deterministic non-convex problems and $O(\epsilon^{-4})$ for the stochastic counterpart provided bounded noise. More recently, Li et al. (2023a) propose a broader notion called $\ell$-smoothness, where the Hessian is bounded by a non-decreasing function of the gradient norm. This condition captures a wider range of practical models, and under it, they recover the classic convergence rates: logarithmic in $\epsilon^{-1}$ for strongly convex problems, $O(\epsilon^{-1})$ for convex ones, and $O(\epsilon^{-2})$ for non-convex objectives. They further prove that (modified) Nesterov's accelerated method achieves $O(\epsilon^{-1/2})$ on convex losses, and that SGD still matches the optimal $O(\epsilon^{-4})$ rate in stochastic non-convex settings (Arjevani et al., 2023).

## 2. Generalized Smooth Function Class

In this section, we formally introduce a refined notion of generalized smoothness on the basis of Li et al. (2023a), so as to accommodate non-Euclidean geometries. First, we introduce the following notations used throughout this paper.

**Notations** Let $\|\cdot\|$ represent a general norm on a finite-dimensional Banach space $\mathcal{E}$, with its corresponding dual norm defined as $\|\mathbf{x}\|_* := \sup_{\mathbf{y} \in \mathcal{E}} \{\langle \mathbf{x}, \mathbf{y} \rangle \mid \|\mathbf{y}\| \leq 1\}$ for

any $\mathbf{x} \in \mathcal{E}^*$, where $\mathcal{E}^*$ is the dual space of $\mathcal{E}$. The optima of the objective is denoted by $\mathbf{x}_* :\in \operatorname{argmin}_{\mathbf{x} \in \mathcal{X}}$ and that $f^* := f(\mathbf{x}_*)$, where $\mathcal{X} \subseteq \mathcal{E}$. We denote the ball centered at $\mathbf{x}_0$ with radius $r_0$ by $\mathcal{B}(\mathbf{x}_0, r_0) := \{\mathbf{x} \in \mathcal{E} \mid \|\mathbf{x} - \mathbf{x}_0\| \leq r_0\}$. We denote the set of non-negative real numbers by $\mathbb{R}_+$ and the set of positive real numbers by $\mathbb{R}_{++}$. Positive integers and natural numbers are represented by $\mathbb{N}_+$ and $\mathbb{N}$, respectively. The $n$-dimensional all-ones vector and the one-hot vectors are denoted by $\mathbf{1}_n$ and $\{\mathbf{e}_i\}_{i=1}^n$, respectively. Additionally, we use the $\widetilde{O}(\cdot)$ notation to hide absolute constants and poly-logarithmic factors in $t$.

## 2.1. Definitions and Properties

Note that in the Euclidean case, characterizing smoothness either by gradient or Hessian is straightforward due to the nice property $\|\cdot\|_2 = \|\cdot\|_{2,*}$. For the non-Euclidean case where $\|\cdot\| \neq \|\cdot\|_*$, the previous formulation $\left\|\nabla^2 f(\mathbf{x})\right\|_2 \leq \ell(\|\nabla f(\mathbf{x})\|_2)$ in Li et al. (2023a) fails to account for the inherent heterogeneity between the norm and its dual. To clarify, we first rewrite the definition in Li et al. (2023a) into $\sup_{\mathbf{h} \in \mathcal{E} \setminus \{\mathbf{0}\}} \left\{\left\|\nabla^2 f(\mathbf{x})\mathbf{h}\right\|_2 / \|\mathbf{h}\|_2\right\} \leq \ell(\|\nabla f(\mathbf{x})\|_2)$. Then, we underline that the mapping induced by the Hessian matrix (i.e., $\nabla^2 f(\mathbf{x})\mathbf{h}$) and the gradient (i.e., $\nabla f(\mathbf{x})$) lie in the dual space $\mathcal{E}^*$ (Nesterov et al., 2018, Theorem 2.1.6), necessitating the use of the dual norm. To this end, we reformulate the definition into $\sup_{\mathbf{h} \in \mathcal{E} \setminus \{\mathbf{0}\}} \left\{\left\|\nabla^2 f(\mathbf{x})\mathbf{h}\right\|_* / \|\mathbf{h}\|\right\} \leq \ell(\|\nabla f(\mathbf{x})\|_*)$, where the dual norm $\|\cdot\|_*$ is imposed to measure the size of $\nabla^2 f(\mathbf{x})\mathbf{h}$ and $\nabla f(\mathbf{x})$. The formal definition is given below.

**Definition 2.1.** ($\ell*$-smoothness) A differentiable function $f : \mathcal{X} \to \mathbb{R}$ belongs to the class $\mathcal{F}_\ell(\|\cdot\|)$ w.r.t. a non-decreasing continuous link function $\ell : \mathbb{R}_+ \to \mathbb{R}_{++}$ if

$$\forall \mathbf{h} \in \mathcal{X}, \; \left\|\nabla^2 f(\mathbf{x})\mathbf{h}\right\|_* \leq \ell(\|\nabla f(\mathbf{x})\|_*) \|\mathbf{h}\| \quad (1)$$

holds almost everywhere for $\mathbf{x} \in \mathcal{X}$.

*Remark* 2.2. Through careful analysis, we discover that it suffices to take the supremum over the domain $\mathcal{X}$ rather than the whole space $\mathcal{E}$, indicating that Definition 2.1 enjoys a slightly weaker condition than Li et al. (2023a).

*Remark* 2.3. When $\|\cdot\| = \|\cdot\|_2$, Definition 2.1 simplifies to the $\ell$-smoothness definition in Li et al. (2023a), which recovers the $L$-smooth condition (Nesterov et al., 2018) and recently proposed $(L_0, L_1)$-smoothness (Zhang et al., 2020b) with $\ell(\alpha) \equiv L$ and $\ell(\alpha) = L_0 + L_1\alpha$, respectively. In fact, due to the equivalence of norms (Boyd & Vandenberghe, 2004), $\ell$-smoothness can be transformed into $\ell*$-smoothness with a rescaled $\ell(\cdot)$. Whereas, the rescaling brings an additional factor dependent on the problem dimension, ultimately giving rise to *dimension-dependent* convergence rates.

In general, our formulation encompasses a broader class of

functions and also plays a pivotal role in applying mirror-descent-type optimization algorithms. We emphasize that strict twice-differentiability is not required here, as Definition 2.1 permits (1) to fail on a set of measure zero[1]. Definition 2.1 can be seen as a global condition on the Hessian, but it lacks local characterization of the gradients, which proves to be essential for convergence analysis (Zhang et al., 2020b). Therefore, we introduce the following definition to capture the local Lipschitzness of the gradient, which is equivalent to Definition 2.1 under mild conditions, as illustrated in the sequel.

**Definition 2.4.** $((\ell, r)*$-smoothness) A differentiable function $f : \mathcal{X} \to \mathbb{R}$ belongs to the class $\mathcal{F}_{\ell, r}(\|\cdot\|)$ w.r.t. a non-decreasing continuous link function $\ell : \mathbb{R}_+ \to \mathbb{R}_{++}$ and a non-increasing continuous radius function $r : \mathbb{R}_+ \to \mathbb{R}_{++}$ if (i) $\forall \mathbf{x} \in \mathcal{X}, \mathcal{B}(\mathbf{x}, r(\|\nabla f(\mathbf{x})\|_*)) \subseteq \mathcal{X}$; (ii) $\forall \mathbf{x}_1, \mathbf{x}_2 \in \mathcal{B}(\mathbf{x}, r(\|\nabla f(\mathbf{x})\|_*))$,

$$\|\nabla f(\mathbf{x}_1) - \nabla f(\mathbf{x}_2)\|_* \leq \ell(\|\nabla f(\mathbf{x})\|_*) \|\mathbf{x}_1 - \mathbf{x}_2\|. \quad (2)$$

Similar to Li et al. (2023a), we introduce the following standard assumption to bridge the gap between two function classes.

**Assumption 2.5.** The objective function $f$ is differentiable and *closed* within its *open* domain $\mathcal{X}$.

$f$ is closed if its sub-level sets $\{\mathbf{x} \in \mathcal{X} | f(\mathbf{x}) \leq a\}$ are closed for all $a \in \mathbb{R}$ (Boyd & Vandenberghe, 2004). A continuous $f$ satisfying Assumption 2.5 if and only if $f(\mathbf{x})$ tends to $+\infty$ when $\mathbf{x}$ approaches the boundary of $\mathcal{X}$ (Liu et al., 2023b). Note that when the optimization problem degenerates to the unconstrained setting, this assumption holds naturally for all continuous functions (Boyd & Vandenberghe, 2004). Now we present the following proposition, indicating that $\mathcal{F}_\ell(\|\cdot\|)$ and $\mathcal{F}_{\ell, r}(\|\cdot\|)$ are nearly equivalent.

**Proposition 2.6.** *(Equivalence between two definitions of generalized smoothness)*

(i) $\mathcal{F}_{\ell, r}(\|\cdot\|) \subseteq \mathcal{F}_\ell(\|\cdot\|)$;

(ii) *under Assumption 2.5, if* $f \in \mathcal{F}_\ell(\|\cdot\|)$, *then* $f \in \mathcal{F}_{\widetilde{\ell}, \widetilde{r}}(\|\cdot\|)$ *where* $\widetilde{\ell}(\alpha) = \ell(\alpha + G)$ *and* $\widetilde{r}(\alpha) = G/\widetilde{\ell}(\alpha)$ *for any* $G \in \mathbb{R}_{++}$.

For simplicity, we restrict our focus to $\mathcal{F}_\ell(\|\cdot\|)$ for the algorithms discussed hereafter. Using Proposition 2.6, we can transform Definition 2.1 into Definition 2.4, enabling a quantitative depiction of the local smoothness property

---

[1]One can also follow the similar arguments as in Zhang et al. (2020b) to relax the twice-differentiability condition by considering $\limsup_{\mathbf{h} \to \mathbf{0}} \|\nabla f(\mathbf{x} + \mathbf{h}) - \nabla f(\mathbf{x})\|_* / \|\mathbf{h}\| \leq \ell(\|\nabla f(\mathbf{x})\|_*)$.

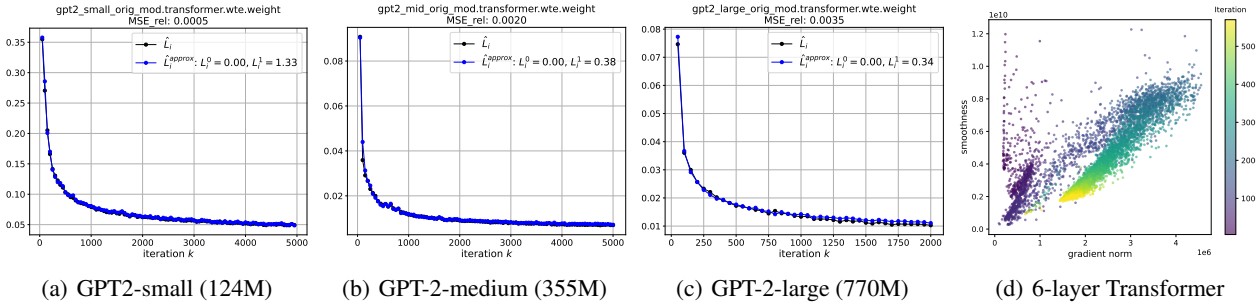

*Figure 1.* Empirical validation of $\ell*$-smoothness on language modeling tasks with GPT-2 models and a 6-layer Transformer.

around a point $\mathbf{x} \in \mathcal{X}$, as demonstrated in the following lemma.

**Lemma 2.7.** *For any* $f \in \mathcal{F}_\ell(\|\cdot\|)$ *satisfying* $\|\nabla f(\mathbf{x})\|_* \le G \in \mathbb{R}_+$ *and given* $\mathbf{x} \in \mathcal{X}$, *the following properties hold:*

*(i)* $\mathcal{B}(\mathbf{x}, G/L) \subseteq \mathcal{X}$ *where* $L := \ell(2G)$;

*(ii)* $\forall \mathbf{x}_1, \mathbf{x}_2 \in \mathcal{B}(\mathbf{x}, G/L),\ \|\nabla f(\mathbf{x}_1) - \nabla f(\mathbf{x}_2)\|_* \le L \|\mathbf{x}_1 - \mathbf{x}_2\|$;

*(iii)* $\forall \mathbf{x}_1, \mathbf{x}_2 \in \mathcal{B}(\mathbf{x}, G/L),\ f(\mathbf{x}_1) \le f(\mathbf{x}_2) + \langle \nabla f(\mathbf{x}_2), \mathbf{x}_1 - \mathbf{x}_2 \rangle + L \|\mathbf{x}_1 - \mathbf{x}_2\|^2 / 2$.

Lemma 2.7 provides (i) an optimistic estimate of the local Lipschitz parameter $\ell(2 \|\nabla f(\mathbf{x})\|_*)$, and (ii) functional estimation bounds analogous to $L$-smoothness. This establishes a connection between $\mathcal{F}_\ell(\|\cdot\|)$ and the $L$-smooth function class, demonstrating that the former exhibits similar properties as the latter in a local region. As Lemma 2.7 requires $\|\nabla f(\mathbf{x})\|_* \le G$, the boundedness of gradients plays a critical role in this transformation, which will be discussed further in Section 3.2.

### 2.2. Examples

Li et al. (2023a) provide a series of concrete examples satisfying the $\ell$-smooth condition. Since our formulation generalizes theirs, these example functions also satisfy $\ell*$-smoothness. Nonetheless, we provide additional examples to illustrate the potential emergence of *dimension-dependent* factors, highlighting the advantages of the algorithmic adaptivity brought by mirror descent. The comprehensive discussions and omitted details can be found in Appendix C.

We present following example function to *(i) justify the necessity of generalized smoothness,* and *(ii) show the advantage of* $\ell*$*-smoothness over Euclidean* $\ell$*-smoothness.*

$$f(\mathbf{x}) := (\mathbf{1}_n^\top \mathbf{x})^4 / 4, \quad \mathbf{x} \in \mathbb{R}^n. \quad (3)$$

This type of structure frequently appears in non-linear regression, polynomial neural networks, and self-attention mechanism approximations. The gradient and Hessian are

$$\nabla f(\mathbf{x}) = (\mathbf{1}_n^\top \mathbf{x})^3 \mathbf{1}_n \text{ and } \nabla^2 f(\mathbf{x}) = 3(\mathbf{1}_n^\top \mathbf{x})^2 \mathbf{1}_n \mathbf{1}_n^\top. \quad (4)$$

Classic smoothness constant is given by $L = \left\| \nabla^2 f(\mathbf{x}) \right\|_2 = 3n(\mathbf{1}_n^\top \mathbf{x})^2$, which can grow arbitrarily large. This renders the classic smoothness model unsuitable for the function defined in (3). Instead, the following proposition implies that $\ell$- and $\ell*$-smoothness remedies the unbounded Hessian, and is thus more appropriate. Proof is postponed to Appendix C.1.

**Proposition 2.8.** *(i)* $f \in \mathcal{F}_{\widehat{\ell}}(\|\cdot\|_2)$ *with* $\widehat{\ell}(\alpha) = n + 2\sqrt{n}\alpha$; *(ii)* $f \in \mathcal{F}_{\widetilde{\ell}}(\|\cdot\|_1)$ *with* $\widetilde{\ell}(\alpha) = 1 + 2\alpha$.

Evidently, our $\ell*$-smoothness can be significantly tighter than $\ell$-smoothness of Li et al. (2023a), with a worst-case dimensional factor of $\sqrt{n}$. In Appendices C.2 and C.3, we further exhibit examples where $\widetilde{\ell}(\alpha)/\widehat{\ell}(\alpha) = O(1/n)$, $O(1/\sqrt{n})$, $O(\sqrt{\log n}/\sqrt{n})$, and $O(n^{-0.4})$, demonstrating even stronger gains. Since both Li et al. (2023a) and our Theorem 3.5 yield convergence rates proportional to the link function $\ell$, mirror descent under $\ell*$-smoothness achieves improved rates by these *dimension-dependent* factors (see Appendix C.5 for detailed discussions).

The examples and the justifications discussed above are not confined to unbounded domains like $\mathbb{R}^n$. They remain valid in bounded domains, as elaborated in Appendix C.1.3. We also discuss the constant-link function example in Appendix C.2, the numerical example in Appendix C.3, and the block diagonal Hessian example in Appendix C.4, demonstrating a wide applicability of our $\ell*$-smoothness model.

### 2.3. Empirical Demonstration

Consider the layer-wise non-Euclidean smoothness model proposed in Riabinin et al. (2025, Assumption 1):

$$\frac{\|\nabla_i f(X) - \nabla_i f(Y)\|_{(i)\star}}{\|X_i - Y_i\|_{(i)}} \le L_i^0 + L_i^1 \|\nabla_i f(X)\|_{(i)\star}, \quad (5)$$

where $i \in [K]$ denotes the $i$-th layer of a deep neural network. Notice that our Definitions 2.1 and 2.4 actually cover the above assumption since setting $\|\cdot\|$ in (2) to $\|\cdot\|_{(i)}$ directly generates (5). Motivated by this, we pretrain the GPT-2 model series (Radford et al., 2019) on the FineWeb

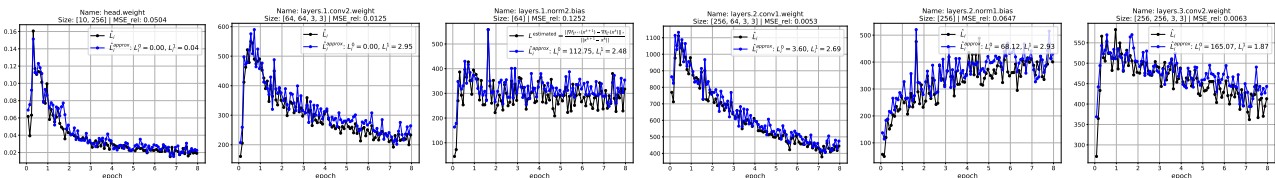

Figure 2. Empirical validation of $\ell*$-smoothness on computer vision tasks with CNN models.

dataset (Penedo et al., 2024) to verify the $\ell*$-smoothness, with results shown in Figures 1(a) to 1(c). A strong agreement between real and approximated $L_i$ values suggests that our $\ell*$-smooth assumption aligns well with the real optimization trajectory of LLMs. Following Crawshaw et al. (2022), we also experiment on a 6-layer Transformer model (Vaswani et al., 2017) on the WMT'16 dataset (Bojar et al., 2016). Figure 1(d) provides a clear visualization of how the model's local curvature (measured in $\ell_1$-norm) grows w.r.t. the gradient (measured in $\ell_\infty$-norm), which further strengthens our Definitions 2.1 and 2.4.

As suggested by Zhang et al. (2024c), CNN and Transformer typically exhibit distinct Hessian spectrums and smoothness properties. Therefore, we also investigate the smoothness properties of CNN models for the standard computer vision task: multi-class image classification on CIFAR-10 (Krizhevsky, 2009). Similar to Figures 1(a) to 1(c), we leverage the approximation in (5) for different layers of the CNN. The results in Figure 2 consistently support our non-Euclidean generalized smoothness model.

Empirically, we follow the implementation in Riabinin et al. (2025) to approximate the model in (5) along the optimization trajectory. The curvature quantity is calculated via (21), incurring negligible computation overhead. To verify (5) more closely, we further replace the mini-batch-based calculation in (21) by a *full-batch* calculation to ablate the effect of gradient noise. This is done by choosing a batch size equal to the dataset size. The results in Figure 4, deferred to Appendix B, remain consistent with Figure 2. We only perform this *full-batch* experiment for CNNs, since the analogue is computationally infeasible for LLMs. Further experimental details can be found in Appendix B.

## 3. Mirror Descent Meets $\ell*$-smoothness

This section formally presents the convergence results for a series of first-order optimization algorithms based on mirror descent, whose analyses can be found in Appendices E to H.

### 3.1. Preliminaries

First, we state the general setup of mirror descent algorithms (Beck & Teboulle, 2003).

**Definition 3.1.** We call a continuous function $\psi : \mathcal{X} \rightarrow \mathbb{R}_+$

a distance-generating function with modulus $\alpha$ w.r.t. $\|\cdot\|$, if (i) the set $\mathcal{X}^o = \{\mathbf{x} \in \mathcal{X} | \partial\psi(\mathbf{x}) \neq \emptyset\}$ is convex; (ii) $\psi$ is continuously differentiable and $\alpha$-strongly convex w.r.t. $\|\cdot\|$, i.e., $\langle \nabla\psi(\mathbf{x}_1) - \nabla\psi(\mathbf{x}_2), \mathbf{x}_1 - \mathbf{x}_2 \rangle \geq \alpha \|\mathbf{x}_1 - \mathbf{x}_2\|^2$, $\forall \mathbf{x}_1, \mathbf{x}_2 \in \mathcal{X}^o$.

**Definition 3.2.** Define the Bregman function $B : \mathcal{X} \times \mathcal{X}^o \rightarrow \mathbb{R}_+$ associated with the distance-generating function $\psi$ as

$$B(\mathbf{x}, \mathbf{x}^o) = \psi(\mathbf{x}) - \psi(\mathbf{x}^o) - \langle \nabla\psi(\mathbf{x}^o), \mathbf{x} - \mathbf{x}^o \rangle.$$

We equip domain $\mathcal{X}$ with a distance-generating function $\psi(\cdot)$, which is, WLOG, 1-strongly convex w.r.t. a certain norm $\|\cdot\|$ endowed on $\mathcal{E}$. We also define the corresponding Bregman divergence according to Definition 3.2. Then, we introduce the following assumptions on the objective function, which controls the growth rate of $\ell(\cdot)$, as also required in Li et al. (2023a).

**Assumption 3.3.** The objective satisfies $f \in \mathcal{F}_\ell(\|\cdot\|)$, where $\ell$ is sub-quadratic, i.e., $\lim_{\alpha\to\infty} \alpha^2/\ell(\alpha) = \infty$.

For convenience, we also define the following prox-mapping function (Nemirovski, 2004). For any $\mathbf{g} \in \mathcal{E}, \mathbf{y} \in \mathcal{X}$,

$$\mathcal{P}_{\mathbf{y}}(\mathbf{g}) := \underset{\mathbf{x} \in \mathcal{X}}{\operatorname{argmin}} \{\langle \mathbf{g}, \mathbf{x} \rangle + B(\mathbf{x}, \mathbf{y})\}, \qquad (6)$$

which serves as the basic component of the optimization algorithms being considered.

### 3.2. Main Idea and Technical Challenges

As discussed previously, bounding the gradients is a prerequisite for Lemma 2.7, which enables us to reduce the $\ell*$-smoothness analysis to the conventional analysis of $L$-smooth functions. In the Euclidean case, Li et al. (2023a) bound the gradients along the optimization trajectory by tracking the gradients *directly*. More specifically, they manage to show that for gradient descent, the $\ell_2$-norm of the gradients decreases monotonically provided that the learning rates are sufficiently small. However, this vital discovery builds upon the geometry of the Euclidean setting, i.e., $\langle \mathbf{x}, \mathbf{x} \rangle = \|\mathbf{x}\|_2^2$, which is invalid in the non-Euclidean setting.

**Technical challenges** The underlying problem is that for mirror descent, the dual norms of the gradients do not necessarily form a monotonic decreasing sequence. The difficulty stems from the nature of mirror descent, where the

descent step is performed in the dual space, making the explicit analysis of the dual norms very difficult and rarely addressed. Even under $L$-smoothness, tracking the gradients directly for mirror descent remains challenging, as discussed in Zhang & He (2018); Lan (2020); Huang et al. (2021) and references therein, unless the problem exhibits special structures (Zhou et al., 2017) or extra assumptions are exerted (Lei & Zhou, 2017).

**Main idea** To tackle this problem, we introduce the following *generalized version* of the reversed Polyak-Łojasiewicz inequality (Polyak, 1963; Lojasiewicz, 1963), aka the self-bounding property of smooth functions (Srebro et al., 2010; Zhang et al., 2017; Zhang & Zhou, 2019).

**Lemma 3.4.** *For any $f \in \mathcal{F}_\ell(\|\cdot\|)$, any $\mathbf{x} \in \mathcal{X}$, it holds that*

$$\|\nabla f(\mathbf{x})\|_*^2 \leq 2\ell(2\|\nabla f(\mathbf{x})\|_*)(f(\mathbf{x}) - f^*).$$

Lemma 3.4 measures the growth of the gradient's dual norm by the suboptimality gap $f(\mathbf{x}) - f^*$. By Assumption 3.3, if we further assume $\ell(\alpha) = (\alpha/2)^\beta, \beta \in [0, 2)$, then $\|\nabla f(\mathbf{x})\|_* \leq 2[f(\mathbf{x}) - f^*]^{1/(2-\beta)}$ is immediately implied. Hence, Lemma 3.4 suggests that $\|\nabla f(\mathbf{x})\|_*$ is effectively under control if $\ell(\cdot)$ does not grow too rapidly. In this way, the gradient can be bounded by the suboptimality gap of the objective function, which is generally much easier in the context of mirror descent. We rigorously show that the suboptimality gap at an arbitrary iterate is bounded by that of the starting point (see Lemma E.2), which further suggests the boundness of the gradients. Once the gradients have an upper bound, we can construct an effective smoothness parameter $L$ that serves as an optimistic estimation of the local curvatures along the trajectory. Formally, we define

$$G := \sup \left\{ \alpha \in \mathbb{R}_+ | \alpha^2 \leq 2\ell(2\alpha)(f(\mathbf{x}_0) - f^*) \right\}, \\ \text{and } L := \ell(2G), \quad (7)$$

where $\mathbf{x}_0 = \operatorname{argmin}_{\mathbf{x} \in \mathcal{X}} \psi(\mathbf{x})$ is selected as the starting point for all optimization algorithms considered in this paper, whose trajectory provably satisfies $\|\nabla f(\mathbf{x}_t)\|_* \leq G$ (will be stated rigorously in the theorems to come) and consequently, $\ell(2\|\nabla f(\mathbf{x}_t)\|_*) \leq L$. We also emphasize that both $G$ and $L$ are absolute constants, i.e., $G, L < \infty$ (Lemma D.3), which is essential to the validity of our theoretical derivations.

### 3.3. Mirror Descent and Accelerated Mirror Descent

In this section, we provide theoretical guarantees for mirror descent and its accelerated variant under $\ell*$-smoothness. We start with the basic mirror descent algorithm (Nemirovski & Yudin, 1983; Beck & Teboulle, 2003) given by

$$\mathbf{x}_{t+1} = \mathcal{P}_{\mathbf{x}_t}(\eta \nabla f(\mathbf{x}_t)), \quad (8)$$

where $\eta$ represents the learning rate, and the prox-mapping function $\mathcal{P}.(\cdot)$ is defined in (6). Our convergence analysis relies on the interesting fact that the suboptimality gap along the trajectory is a non-increasing sequence: $f(\mathbf{x}_t) - f^* \leq f(\mathbf{x}_{t-1}) - f^* \leq \cdots \leq f(\mathbf{x}_0) - f^*$, provided that learning rates are chosen appropriately. Consequently, the gradients along the trajectory are bounded by Lemma 3.4, and so are the local Lipschitz constants. The formal statements are illustrated in the following theorem.

**Theorem 3.5.** *Under Assumptions 2.5 and 3.3, if $0 < \eta \leq 1/L$, then the gradients along the trajectory generated by (8) satisfy $\|\nabla f(\mathbf{x}_t)\|_* \leq G, \forall t \in \mathbb{N}$, and the average-iterate as well as the last-iterate convergence rate is given by*

$$\max \left\{ f(\overline{\mathbf{x}}_T) - f^*, f(\mathbf{x}_T) - f^* \right\} \leq \frac{B(\mathbf{x}_*, \mathbf{x}_0)}{\eta T}, \quad (9)$$

*where $\overline{\mathbf{x}}_T = \frac{1}{T} \sum_{t=1}^{T} \mathbf{x}_t$.*

Our convergence result recovers the classical $O(1/T)$ rate for deterministic convex optimization (Bubeck et al., 2015; Lan, 2020), for both the average-iterate and the last-iterate.

*Remark* 3.6. The fact that $G$ is a finite constant is proven *after* we analyze the trajectory, rather than being used as an explicit *prior* assumption to analyze the trajectory. At first glance, it will incur a "circular" analysis. However, this seemingly circularity is exactly the key underlying our analysis techniques, where we break rigorously in Lemmas D.3 and E.2 and Theorem 3.5 by induction.

Next, we investigate the accelerated version of mirror descent. The acceleration scheme for smooth convex optimization originates from Nesterov (1983) and has numerous variants (Allen-Zhu & Orecchia, 2017; d'Aspremont et al., 2021). In this paper, we study one of its simplest versions (Lan, 2020):

$$\begin{aligned} \mathbf{y}_t &= (1 - \alpha_t)\mathbf{x}_{t-1} + \alpha_t \mathbf{z}_{t-1}, \\ \mathbf{z}_t &= \mathcal{P}_{\mathbf{z}_{t-1}}(\eta_t \nabla f(\mathbf{y}_t)), \\ \mathbf{x}_t &= (1 - \alpha_t)\mathbf{x}_{t-1} + \alpha_t \mathbf{z}_t. \end{aligned} \quad (10)$$

We present the following theoretical guarantee, which achieves the optimal $O(1/T^2)$ convergence rate of first-order methods (Nesterov et al., 2018).

**Theorem 3.7.** *Under Assumptions 2.5 and 3.3, define[2]*

$$L := \ell(4G), \tau := \left\lceil \frac{4\sqrt{2B(\mathbf{x}_*, \mathbf{x}_0)L}}{G} \right\rceil - 1, \alpha_t := \frac{2}{t+1},$$

$$\eta := \min \left\{ 1, \frac{3(\tau - 1)}{2(\tau - 3)(\tau - 2)} \right\}, \eta_t := \frac{t\eta}{2L}.$$

*Then the convergence rate of (10) is given by*

$$f(\mathbf{x}_T) - f^* \leq \frac{LB(\mathbf{x}_*, \mathbf{x}_0)}{\eta T(T + 1)}, \quad (11)$$

---

[2]Due to technical reasons, $L = \ell(4G)$ needs to be slightly larger than the one used in mirror descent. This adjustment affects only accelerated mirror descent; other algorithms still use $L = \ell(2G)$.

*and the gradients along the trajectory satisfy* $\max\{\|\nabla f(\mathbf{y}_t)\|_*, 2\|\nabla f(\mathbf{x}_t)\|_*\} \le 2G$ *for all* $t \in \mathbb{N}$.

*Remark* 3.8. The general idea is similar to mirror descent, that is, bounding the gradients along the trajectory via controlling the suboptimality gaps. However, as (10) maintains three sequences, the analysis becomes significantly more challenging compared to the single sequence $\{\mathbf{x}_t\}_{t\in\mathbb{N}}$ in (8). While $\|\nabla f(\mathbf{x}_t)\|_* \lesssim G$ follows directly from the convergence in (11), it cannot be directly extended to the sequence $\{\mathbf{y}_t\}_{t\in\mathbb{N}}$, whose gradients also need to be bounded. To address this issue, we focus on the key quantity $e_t := \|\mathbf{y}_t - \mathbf{x}_{t-1}\|$, which measures the distance between the two sequences. If $e_t \lesssim G/L$, we can readily deduce $\|\nabla f(\mathbf{y}_t)\|_* \lesssim G$ by leveraging the local smoothness property in Definition 2.4. To achieve this goal, we develop a "time partition" technique illustrated in Lemma F.3. Specifically, when $t$ is below the threshold $\tau$, we bound $e_t$ by a contraction mapping, which effectively limits its growth for small $t$. Conversely, when $t$ exceeds $\tau$, $e_t$ provably decays hyperbolically. Combining the two cases, we can derive that $e_t \lesssim G/L$, which suggests the boundness of the gradients.

**Comparison with Li et al. (2023a, Theorem 4.4)** For $f \in \mathcal{F}_\ell(\|\cdot\|_2)$, Li et al. (2023a) study a variant of Nesterov's accelerated gradient algorithm (NAG) and also recover the optimal bound. They introduce an auxiliary sequence into the original NAG, which aims to aggressively stabilize the optimization trajectory. The stabilization mainly refers to $\|\mathbf{y}_t - \mathbf{x}_t\|$ in their language, sharing a similar spirit as $e_t$ in our paper. To this end, they have to use a smaller step size ($\eta \simeq 1/L^2$) and impose a more complex analysis than the original NAG. Unlike their framework, we concentrate on a more stable acceleration scheme in (10), without manually tuning down the base learning rate $\eta$ or bringing in any additional algorithmic components. Moreover, our proof is much simpler and more intuitive, as reflected in our constructive ways of analyzing the stability term $e_t$ by partitioning the timeline. It is also worth mentioning that we do not need the fine-grained characterization on the modulus of $\ell$, while Li et al. (2023a) further assume $\ell(u) = O(u^\alpha), \alpha \in (0, 2]$.

### 3.4. Optimistic Mirror Descent and Mirror Prox

In this subsection, we consider another genre of the classic optimization algorithm: optimistic mirror descent and mirror prox. Since these algorithms are often treated as a single framework in the online learning community (Chen et al., 2023a; 2024; Wang et al., 2024b), we adopt the terminology in Mokhtari et al. (2020); Azizian et al. (2021) to avoid potential confusion.

Unlike mirror descent, optimistic mirror descent and mirror prox perform two prox-mappings per iteration and incorporate current observed information to estimate the functional curvature. Firstly, we consider the following optimistic mir-

ror descent algorithm (Popov, 1980; Chiang et al., 2012; Rakhlin & Sridharan, 2013a;b), which requires 1 gradient query per iteration.

$$\mathbf{y}_t = \mathcal{P}_{\mathbf{x}_{t-1}}(\eta\nabla f(\mathbf{y}_{t-1})), \ \mathbf{x}_t = \mathcal{P}_{\mathbf{x}_{t-1}}(\eta\nabla f(\mathbf{y}_t)), \quad (12)$$

where we initialize $\mathbf{y}_0 = \mathbf{x}_0 = \operatorname{argmin}_{\mathbf{x}\in\mathcal{X}} \psi(\mathbf{x})$. The following theorem establishes the classic $O(1/T)$ convergence rate for this algorithm.

**Theorem 3.9.** *Under Assumptions 2.5 and 3.3, if* $0 < \eta \le 1/(3L)$, *then the gradients along the trajectory generated by* (12) *satisfy* $\max\{\|\nabla f(\mathbf{y}_t)\|_*, \|\nabla f(\mathbf{x}_t)\|_*\} \le G, \forall t \in \mathbb{N}$, *and the convergence rate is given by*

$$f(\overline{\mathbf{y}}_T) - f^* \le \frac{B(\mathbf{x}_*, \mathbf{x}_0)}{\eta T}, \quad where \ \overline{\mathbf{y}}_T = \frac{1}{T}\sum_{t=1}^T \mathbf{y}_t. \ (13)$$

Next, we investigate the mirror prox algorithm (Korpelevich, 1976; Nemirovski, 2004; Nesterov, 2007), which is defined via the following updates:

$$\mathbf{y}_t = \mathcal{P}_{\mathbf{x}_{t-1}}(\eta\nabla f(\mathbf{x}_{t-1})), \ \mathbf{x}_t = \mathcal{P}_{\mathbf{x}_{t-1}}(\eta\nabla f(\mathbf{y}_t)). \quad (14)$$

The only difference between (12) and (14) is that the latter does not reuse gradient information, which results in 2 gradient queries per iteration. For a comprehensive overview and comparison between these two algorithms, one may refer to Azizian et al. (2021); Cai et al. (2022b). The same $O(1/T)$ convergence rate is also obtained, as shown in the theorem below.

**Theorem 3.10.** *Under Assumptions 2.5 and 3.3, if* $0 < \eta \le 1/(2L)$, *then the gradients along the trajectory generated by* (14) *satisfy* $\max\{\|\nabla f(\mathbf{y}_t)\|_*, \|\nabla f(\mathbf{x}_t)\|_*\} \le G, \forall t \in \mathbb{N}$, *and the convergence rate is given by*

$$f(\overline{\mathbf{y}}_T) - f^* \le \frac{B(\mathbf{x}_*, \mathbf{x}_0)}{\eta T}, \quad where \ \overline{\mathbf{y}}_T = \frac{1}{T}\sum_{t=1}^T \mathbf{y}_t. \ (15)$$

The high-level idea behind Theorems 3.9 and 3.10 aligns with that in Section 3.3, i.e., bounding the dual norms of the gradients by analyzing suboptimality gaps. However, the implication from the last-iterate convergence in Theorems 3.5 and 3.7 to the bounded suboptimality gap does not apply for optimistic mirror descent and mirror prox, as elaborated below.

**Hardness results on the last-iterate** Different from the algorithms in Section 3.3 which enjoys a non-asymptotic last-iterate convergence under $L$-smoothness, the last-iterate of (12) as well as (14) had been only known to converge asymptotically, as shown by Popov (1980); Hsieh et al. (2019) for (12); Korpelevich (1976); Facchinei & Pang

(2007) for (14). Even in the Euclidean setting, the finite-time last-iterate convergence is very challenging (Golowich et al., 2020; Wei et al., 2021; Gorbunov et al., 2022) and requires complicated analysis as well as intricate techniques (Cai et al., 2022a), e.g., computer-aided proofs based on sum-of-squares programming (Nesterov, 2000; Parrilo, 2003).

**Circumvent the last-iterate analysis**   Recall the discussions in Section 3.2 that it suffices to ensure the boundness of the suboptimality gap, which can be implied by the last-iterate convergence. Since the latter one is a stronger statement, we can *bypass* the last-iterate convergence analysis, and control the suboptimality gap *directly* through careful analysis. Therefore, the cumbersome procedures in Cai et al. (2022a) can be avoided. Below, we briefly outline our approach.

**Solution**   For the mirror prox method, we establish the descent property for both sequences (Lemma H.2), i.e., $\max\{f(\mathbf{x}_t), f(\mathbf{y}_t)\} \leq f(\mathbf{x}_{t-1})$, which guarantees the bounded suboptimality gap. For the optimistic mirror descent method, we track the stability term $\|\mathbf{y}_t - \mathbf{y}_{t-1}\|$ and show that it grows no faster than the geometric series (Lemma G.2). The upper bound on stability terms suggests that the optimization trajectory is effectively stabilized, ultimately yielding $\max\{f(\mathbf{x}_t), f(\mathbf{y}_t)\} \leq f(\mathbf{x}_0)$. Consequently, we bound both the suboptimality gap and the gradient.

## 4. Stochastic Convex Optimization

In this section, we extend from deterministic optimization to stochastic convex optimization.

### 4.1. Theoretical Results

Consider the following stochastic mirror descent (SMD) algorithm (Nemirovski et al., 2009):

$$\mathbf{x}_{t+1} = \mathcal{P}_{\mathbf{x}_t}(\eta_{t+1}\mathbf{g}_t), \tag{16}$$

where $\mathbf{g}_t$ is an noisy estimate of the true gradient $\nabla f(\mathbf{x}_t)$. Let $\boldsymbol{\epsilon}_t := \mathbf{g}_t - \nabla f(\mathbf{x}_t)$ denote the noise. We make the following sub-Gaussian assumption on $\boldsymbol{\epsilon}_t$, which is common for the high probability convergence (Juditsky et al., 2011; Vershynin, 2018; Zhang et al., 2023; Liu & Zhou, 2024).

**Assumption 4.1.** For all $t \in \mathbb{N}$, the stochastic gradients are unbiased: $\mathbb{E}_{t-1}[\boldsymbol{\epsilon}_t] = 0$, where the expectation $\mathbb{E}_{t-1}$ is conditioned on the past stochasticity $\{\boldsymbol{\epsilon}_s\}_{s=0}^{t-1}$. The noise level further satisfies:

$$\mathbb{E}_{t-1}\left[\exp\left(\lambda \|\boldsymbol{\epsilon}_t\|_*^2\right)\right] \leq \exp\left(\lambda\sigma^2\right), \forall\lambda \in [0, \sigma^{-2}].$$

Now, we formally introduce the convergence of SMD, where the analysis is postponed to Appendix I.

**Theorem 4.2.** *Under Assumptions 2.5, 3.3 and 4.1, for any* $0 < \delta < 1$, *define*

$$G := \sup\left\{\alpha \in \mathbb{R}_+ | \alpha^2 \leq 2\ell(2\alpha)\left(f(\mathbf{x}_0) - f^* + \frac{64\sigma}{\sqrt{\delta}}\right)\right\},$$

$$\eta = \min\left\{\frac{64}{G}, \sqrt{\frac{B(\mathbf{x}_*, \mathbf{x}_0)}{\sigma^2 \log\left(\frac{1}{\delta}\right)\log(T)}}\right\}, \quad L := \ell(2G).$$

*Set* $\eta_t = \min\left\{\frac{1}{2L}, \frac{\eta}{\sqrt{T}}\right\}$, *with probability at least* $1 - \delta$:

$$f(\mathbf{x}_T) - f^*$$
$$\leq O\left(\frac{LB(\mathbf{x}_*, \mathbf{x}_0)}{T} + \sigma\sqrt{\frac{B(\mathbf{x}_*, \mathbf{x}_0)\log\left(\frac{1}{\delta}\right)\log(T)}{T}}\right).$$

*Moreover,* $\|\nabla f(\mathbf{x}_t)\|_* \leq G < +\infty$ *holds for all* $0 \leq t \leq T - 1$ *with probability at least* $1 - \delta$.

The above theorem matches the state-of-the-art last-iterate convergence rate in Liu & Zhou (2024). Notably, our result is valid under $\ell*$-smooth function class, which is more general than the $(L, M)$-function class[3] in Liu & Zhou (2024). More importantly, our analysis is adaptive to noise, i.e., when $\sigma = 0$ degenerates to deterministic optimization, the convergence in Theorem 4.2 encompasses that in Theorem 3.5 (deterministic mirror descent). Our theory answers affirmatively to the application of SMD to train real-world ML models, which is typically non-smooth and may satisfy our $\ell*$-smoothness condition (see our nanoGPT experiments in Section 2.3 as well as Appendix B).

Apart from the aforementioned convergence under sub-Gaussian noise, we also propose a new generalized bounded noise model (Assumption $\widetilde{J.1}$) and establish a time-uniform convergence rate of $\widetilde{O}(1/\sqrt{t})$ (Theorem J.3). Due to space limitations, we defer this part to Appendix J.

### 4.2. Proof Sketch

Similar to the deterministic case, there are two major challenges underlying the convergence analysis for generalized smooth functions: (i) ensure in each step $t$, the iterates $\mathbf{x}_{t+1}, \mathbf{x}_t$ are close enough to exploit the effective smooth property in Lemma 2.7; and (ii) ensure the local curvatures $\ell(2\|\nabla f(\mathbf{x}_t)\|_*)$ to be bounded uniformly by an absolute constant. For the second challenge, we can achieve this goal by bounding gradient norms $\{\|\nabla f(\mathbf{x}_t)\|_*\}_{t=0}^{T-1}$, sharing a similar spirit as in Section 3. For the first challenge, the deterministic case begins with $\|\mathbf{x}_{t+1} - \mathbf{x}_t\| \leq \eta\|\nabla f(\mathbf{x}_t)\|_*$. Since we have already controlled $\|\nabla f(\mathbf{x}_t)\|_*$ in the second challenge, the first challenge has been resolved simultaneously if the learning rate $\eta$ is set appropriately. However,

---

[3]Liu & Zhou (2024, Assumption 3) state that $f$ is $(L, M)$-smooth such that $f(\mathbf{x}) \leq f(\mathbf{y}) + \langle\mathbf{g_y}, \mathbf{x} - \mathbf{y}\rangle + \frac{L}{2}\|\mathbf{x} - \mathbf{y}\|^2 + M\|\mathbf{x} - \mathbf{y}\|, \forall\mathbf{x}, \mathbf{y} \in \mathcal{X}, \mathbf{g_y} \in \partial f(\mathbf{y})$.

for SCO, we write

$$\|\mathbf{x}_{t+1} - \mathbf{x}_t\| \le \eta_{t+1} \|\mathbf{g}_t\|_* \le \eta_{t+1} \|\nabla f(\mathbf{x}_t)\|_* + \eta_{t+1} \|\boldsymbol{\epsilon}_t\|_*.$$

The gradient norm term $\|\nabla f(\mathbf{x}_t)\|_*$ is effectively managed due to challenge (ii). For the noise term $\|\boldsymbol{\epsilon}_t\|_*$, we seek help from Assumption 4.1, which has good properties. We show that throughout $T$ iterations, the probability of "bad" noise estimates (with large noise norms $\|\boldsymbol{\epsilon}_t\|_*$) under Assumption 4.1 can be relatively low.

**How do we manage $\|\nabla f(\mathbf{x}_t)\|_*$ and $\|\boldsymbol{\epsilon}_t\|_*$?** We define the following events

$$A_t := \{f(\mathbf{x}_t) - f^* \le F\}, B_t := \left\{ \|\boldsymbol{\epsilon}_t\|_* \le \frac{G}{2\eta L} \right\},$$

and show that $\cup_{t=0}^{T-1} A_t$ and $\cup_{t=0}^{T-1} B_t$ happen simultaneously with high probability along the optimization trajectory. For $A_t$, it is achieved by carefully analyzing the last-iterate behavior of (16): $f(\mathbf{x}_t) - f(\mathbf{x}) \le b_t(B(\mathbf{x}, \mathbf{x}_0))$, where $b_t$ denotes the bound depending on $B(\mathbf{x}, \mathbf{x}_0)$ for step $t$. We take the reference point $\mathbf{x} = \mathbf{x}_0$ to enable cancellation of terms, and obtain $f(\mathbf{x}_t) - f(\mathbf{x}_0) \le b_t(0)$. By showing that $b_t$ is non-increasing in $t$, we are able to bound $f(\mathbf{x}_t) - f^* \le f(\mathbf{x}_0) - f^* + \max_t b_t = O(1)$. For $B_t$, we use Chebyshev's inequality and take a union bound over the time horizon. The most subtle point underlying the analysis is that we have to perform a "chain-of-event" analysis, rather than taking union bounds brute-force. This is because when we analyze the failure probability of $A_t$, previous events *must happen*, otherwise we are unable to exploit generalized smoothness. Such "chain-like" characteristics require taking union bounds based on mathematical induction.

## 5. Non-convex Optimization

In this section, we extend $\ell*$-smoothness to non-convex composite optimization (Ghadimi et al., 2016; Lan, 2020), which can be formulated as

$$F^* := \min_{\mathbf{x} \in \mathcal{X}} \{F(\mathbf{x}) := f(\mathbf{x}) + \phi(\mathbf{x})\} \tag{17}$$

where $f \in \mathcal{F}_\ell(\|\cdot\|)$ and is possibly non-convex; $\phi : \mathcal{X} \to \mathbb{R}_+$ is a simple convex regularizer, but possibly non-smooth (e.g., $\phi(\mathbf{x}) \equiv 0$ or $\phi(\mathbf{x}) = \|\mathbf{x}\|_1$). (17) can model a wide range of problems in ML since the objective $F$ is neither convex nor smooth. To optimize (17), the *de facto* method is composite mirror descent (CMD) (Duchi et al., 2010; Ghadimi et al., 2016; Lei & Tang, 2018; Lan, 2020; Wang et al., 2024b):

$$\mathbf{x}_{t+1} = \underset{\mathbf{x} \in \mathcal{X}}{\operatorname{argmin}} \left\{ \langle \nabla f(\mathbf{x}_t), \mathbf{x} \rangle + \frac{B(\mathbf{x}, \mathbf{x}_t)}{\eta} + \phi(\mathbf{x}) \right\},$$

where we assume access to exact gradients. In the sequel, we study the convergence behavior of CMD under

$\ell*$-smoothness and show that it still achieves the classic rate under our relaxed smoothness model.

To start, we define the following gradient mapping function (Nesterov et al., 2018) as the convergence criterion:

$$\mathcal{G}_t := \frac{\mathbf{x}_t - \mathbf{x}_{t+1}}{\eta}, \tag{18}$$

which is widely used in the context of mirror descent (Ghadimi et al., 2016; Lan, 2020; Huang et al., 2021). As pointed out by Lan (2020, Lemma 6.3), when the size of $\mathcal{G}_t$ vanishes, $\mathbf{x}_{t+1}$ approaches to a stationary point of (17). For more comprehensive illustrations, the readers may refer to Lan (2020), and references therein. Below, we formally present the theoretical guarantee, whose proof is located in Appendix K.

**Theorem 5.1.** *Under Assumptions 2.5 and 3.3, let*

$$G := \sup \left\{ \alpha \in \mathbb{R}_+ | \alpha^2 \le 2\ell(2\alpha) \left( f(\mathbf{x}_0) - f^* + \phi(\mathbf{x}_0) \right) \right\},$$

$$L := \ell(2G), \ and \ 0 < \eta \le \frac{1}{L}.$$

*Then, CMD enjoys the following convergence rate:*

$$\frac{1}{T} \sum_{t=0}^{t-1} \|\mathcal{G}_t\|^2 \le \frac{F(\mathbf{x}_0) - F^*}{\eta T}. \tag{19}$$

The convergence in (19) matches the one derived under the standard smoothness conditions (Ghadimi et al., 2016; Lan, 2020). This result provides strong theoretical support for the practical application of (composite) mirror descent, as modern neural networks are non-convex and non-smooth. Our result may also shed light on the application of CMD in LLM pretraining (Xie et al., 2023) and preference alignment (Munos et al., 2024; Zhang et al., 2025e;d; Wu et al., 2026), which has been empirically observed to exhibit generalized smooth properties (Crawshaw et al., 2022; Liu et al., 2025b; Riabinin et al., 2025).

## 6. Conclusion

In this paper, we propose a refined smoothness notion called $\ell*$-smoothness, which characterizes the norm of the Hessian by leveraging a general norm and its dual. This extension encompasses the current generalized smooth conditions while highlighting non-Euclidean problem geometries. We establish new convergence results of mirror descent, accelerated mirror descent, optimistic mirror descent, mirror prox, and stochastic mirror descent algorithms under this setting. Beyond convex optimization, we further delve into non-convex composite mirror descent and achieve tight convergence rates. Our theory provides insights into the practical usage of mirror descent, especially in pretraining and preference alignment of LLMs (Xie et al., 2023; Zhang et al., 2025d;e).

## Acknowledgements

This work was partially supported by NSFC (U23A20382), the Fundamental and Interdisciplinary Disciplines Breakthrough Plan of the Ministry of Education of China (No. JYB2025XDXM118), and the Fundamental Research Funds for the Central Universities (2026300271).

## Impact Statement

This paper presents work whose goal is to advance the field of Machine Learning. There are many potential societal consequences of our work, none of which we feel must be specifically highlighted here.

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

# A. Related Work

In this section, we review some concepts of generalized smoothness from the existing literature.

## A.1. $(L_0, L_1)$-smoothness

$(L_0, L_1)$-**smooth: non-convex case**    The concept of $(L_0, L_1)$-smoothness is firstly proposed by Zhang et al. (2020b) based on empirical observations from LSTMs (Merity et al., 2018) and ResNets (He et al., 2016), which allows the function to have an affine-bounded Hessian norm. Under this new condition, they analyze gradient clipping (Mikolov et al., 2012; Pascanu et al., 2013) for deterministic non-convex optimization and derive a complexity of $O(\epsilon^{-2})$[4], which matches the lower bound (Carmon et al., 2020) up to constant factors. They further extend to stochastic settings and provide an $O(\epsilon^{-4})$ complexity bound under the uniformly bounded noise assumption. Gaash et al. (2025) investigate the high-probability convergence of SGD with clipping, and obtains a convergence rate that matches SGD up to polylogarithmic factors and additive terms. Yu et al. (2026) analyze both vector and matrix sign-based optimizers, namely SignSGD (Bernstein et al., 2018), Lion (Chen et al., 2023b), Muon (Jordan et al., 2024), and Muonlight (Liu et al., 2025a), under the generalized heavy-tailed noise conditions. They consider coordinate-wise $(L_0, L_1)$-smooth function class and obtain a complexity of $O(\epsilon^{-\frac{3p-2}{p-1}})$ for all methods, matching state-of-the-art.

$(L_0, L_1)$-**smooth: convex case**    Koloskova et al. (2023) show that gradient clipping has $O(\epsilon^{-1})$ complexity, matching the classic result in deterministic optimization. Takezawa et al. (2024) establish the same complexity bound for gradient descent with polyak stepsizes (Polyak, 1987). However, besides $(L_0, L_1)$-smoothness, these two works further impose an additional $L$-smooth assumption, where $L$ could be significantly larger than $L_0$ and $L_1$. Gorbunov et al. (2025) address the limitation and recover their results without the extra $L$-smoothness assumption. Moreover, they study a variant of adaptive gradient descent (Malitsky & Mishchenko, 2020) and provide an $O(\epsilon^{-1})$ complexity result, albeit with worse constant terms compared to the original one. For acceleration schemes in convex optimization, they modify the method of similar triangles (MST) (Gasnikov & Nesterov, 2018) and prove an optimal complexity of $O(\epsilon^{-0.5})$. Lobanov et al. (2024) provide a refined convergence analysis of gradient descent and achieve a linear speedup.

**Other explorations**    Ever since Zhang et al. (2020b) proposed $(L_0, L_1)$-smoothness condition, this generalized notion of smoothness has been flourishing in minimax optimization (Xian et al., 2024), bilevel optimization (Hao et al., 2024; Gong et al., 2024), multi-objective optimization (Zhang et al., 2025a), sign-based optimization (Crawshaw et al., 2022), zero-th order optimization (Lobanov & Gasnikov, 2025), distributionally robust optimization (Jin et al., 2021; Zhang et al., 2025b) and variational inequality (Vankov et al., 2024; 2025a). Numerous attempts have been made to refine existing algorithms under this weaker assumption, including variance reduction (Reisizadeh et al., 2025), clipping/normalized gradient (Zhang et al., 2020a; Qian et al., 2021; Zhao et al., 2021; Hübler et al., 2024; Yang et al., 2024), error feedback (Khirirat et al., 2025), and trust region methods (Xie et al., 2024a). Notably, Faw et al. (2022; 2023); Wang et al. (2023b); Li et al. (2023b); Wang et al. (2024a); Zhang et al. (2025c) explore the convergence of AdaGrad (Duchi et al., 2011), RMSprop (Hinton et al., 2012), and Adam (Kingma & Ba, 2015) under generalized smoothness. Beyond the optimization community, it has also garnered significant attention in federated learning (Khirirat et al., 2024; Demidovich et al., 2025), meta-learning (Chayti & Jaggi, 2024), and fine-tuning LLMs (Ma et al., 2026).

## A.2. $\alpha$-symmetric Generalized Smoothness

An important generalization of $(L_0, L_1)$-smoothness is $\alpha$-symmetric $(L_0, L_1)$-smoothness proposed by Chen et al. (2023c), which introduces symmetry into the original formulation and also allows the dependency on the gradient norm to be polynomial with degree of $\alpha$. The formal definition is given as follows:

$$\|\nabla f(\mathbf{x}) - \nabla f(\mathbf{y})\|_2 \leq \left( L_0 + L_1 \sup_{\theta \in [0,1]} \|\nabla f(\theta \mathbf{x} + (1-\theta)\mathbf{y})\|_2 \right) \|\mathbf{x} - \mathbf{y}\|_2, \ \forall \mathbf{x}, \mathbf{y} \in \mathcal{E}, \tag{20}$$

for some $L_0, L_1 \in \mathbb{R}_+$. For deterministic non-convex optimization, Chen et al. (2023c) establish the optimal complexity of $O(\epsilon^{-2})$ for a variant of normalized gradient descent (Nesterov, 1984; Cortés, 2006). They also show that the popular SPIDER algorithm (Fang et al., 2018) achieves the optimal $O(\epsilon^{-3})$ complexity in the stochastic setting. As pointed out

---

[4]In non-convex settings, the goal is to find $\epsilon$-stationary point $\mathbf{x}$ satisfying $\|\nabla f(\mathbf{x})\|_2 \leq \epsilon$. In convex settings, we aim at $\epsilon$-suboptimal point $\mathbf{x}$ satisfying $f(\mathbf{x}) - f^* \leq \epsilon$. The meaning of $\epsilon$ depends on the context.

by Chen et al. (2023c); Vankov et al. (2025b), any twice-differentiable function satisfying (20) is also $(L_0', L_1')$-smooth for different constant factors. As an extension, Jiang et al. (2024) study sign-based finite-sum non-convex optimization and obtain an improved complexity over the SignSVRG algorithm (Chzhen & Schechtman, 2023).

### A.3. $\ell$-smoothness

Recently, Li et al. (2023a) significantly generalize $(L_0, L_1)$-smoothness to $\ell$-smoothness, which assumes $\left\|\nabla^2 f(\mathbf{x})\right\|_2 \leq \ell(\|\nabla f(\mathbf{x})\|_2)$ for an arbitrary non-decreasing function $\ell$. The enhanced flexibility of $\ell$ accommodates a broader range of practical ML problems (Devlin et al., 2019; Caron et al., 2021; Radford et al., 2021), and can model certain real-world problems where $(L_0, L_1)$-smoothness fails (Cooper, 2024). Under this condition, they study gradient descent for strongly-convex, convex, and non-convex objectives, obtaining classic results of $O(\log(\epsilon^{-1}))$, $O(\epsilon^{-1})$, and $O(\epsilon^{-2})$, respectively. They also prove that Nesterov's accelerated gradient method (Nesterov, 1983) attains the optimal $O(\epsilon^{-0.5})$ complexity for convex objectives. Furthermore, they delve into stochastic non-convex optimization and show that stochastic gradient descent achieves the optimal complexity of $O(\epsilon^{-4})$ under finite variance conditions, matching the lower bound in Arjevani et al. (2023).

Tyurin (2025) improves the convergence rate from Li et al. (2023a) in the convex setting, and also enhances the rate in the non-convex setting when $\ell$ has a sub-quadratic and quadratic growth. However, for super-quadratic or even exponential $\ell$ in the non-convex regimes, Tyurin (2025) imposes an additional assumption of bounded gradients. Xie et al. (2024b) study online convex optimization problems by assuming each online function $f_t : \mathcal{X} \to \mathbb{R}$ is $\ell_t$-smooth and that $\ell_t(\cdot)$ can be queried arbitrarily by the learner. Based on $\ell_t$-smoothness, they derive gradient-variation regret bounds for convex and strongly convex functions, albeit with the limitation of assuming a globally constant upper bound on the adversary's smoothness.

### A.4. Other Generalizations of Classic Smoothness

It is known that a function $f$ is $L$-smooth if $Lh - f$ is convex (Nesterov et al., 2018) for $h(\cdot) = \|\cdot\|^2 / 2$. Advancements have been made to relax $h$ into an arbitrary Legendre function, leading to the development of a notion called relative smoothness (Birnbaum et al., 2011; Bauschke et al., 2017; Lu et al., 2018; Hanzely et al., 2021; Hanzely & Richtárik, 2021), which substitutes the canonical quadratic upper bound with $f(\mathbf{x}) \leq f(\mathbf{y}) + \langle \nabla f(\mathbf{y}), \mathbf{x} - \mathbf{y} \rangle + L B_h(\mathbf{x}, \mathbf{y})$, where $B_h$ is the Bregman divergence associated with $h$. Since relative smoothness is now standard in the analysis of mirror descent, we provide a detailed comparison with our $\ell*$-smoothness in Appendix A.5.

More recently, Mishkin et al. (2024) propose directional smoothness, a measure of local gradient variation that preserves the global smoothness along specific directions. To address the imbalanced Hessian spectrum distribution in practice (Sagun et al., 2016; Pan et al., 2022), Liu et al. (2025b); Jiang et al. (2025a) propose anisotropic smoothness and obtain improved bounds for adaptive gradient methods.

### A.5. Comparison with Relative Smoothness

According to Lu et al. (2018), a function $f$ is $L$-smooth relative to $h$ iff $Lh - f$ is convex, which is equivalent to $\nabla^2 f(x) \preceq L \nabla^2 h(x)$ when both functions are twice-differentiable on the interior of the domain. Notably, $h$ is not presumed to possess any special properties like strict or strong convexity. In the sequel, we compare this notion with $\ell*$-smoothness and highlight why ours is more practical.

**Practical infeasibility** Relative smoothness constrains the Hessian spectrum via a reference function $h$. However, for any $f$, there exist infinitely many valid $(L, h)$ pairs satisfying the definition. Therefore, finding the most appropriate pair is essential under this framework. Unfortunately, in most cases, $(L, h)$ can only be determined through delicate design (requiring prior knowledge of $f$). In contrast, $\ell*$-smoothness bounds the matrix primal-dual norm of the Hessian via a link function of the gradient dual norm, i.e., $\left\|\nabla^2 f(\mathbf{x})\right\|_{*,\cdot} := \sup_{\mathbf{h}} \left\{ \left\|\nabla^2 f(\mathbf{x})\mathbf{h}\right\|_* / \|\mathbf{h}\| \right\} \leq \ell(\|\nabla f(\mathbf{x})\|_*)$. This formulation does not involve Löwner partial order $\preceq$ and is generally much easier to verify, as it compresses the information of eigenvalues into a specific norm and connects it with gradients. Moreover, recent advances in ML suggest that $\ell*$-smoothness is satisfied by many practical objectives both theoretically and empirically (see references discussed earlier), whereas relative smoothness remains less prevalent in the ML community. To summarize, both notions extend the traditional smoothness from different perspectives: relative smoothness is more direct and strict, while generalized smoothness is more flexible and practical. Below, we provide concrete examples and further reasoning to support our claims.

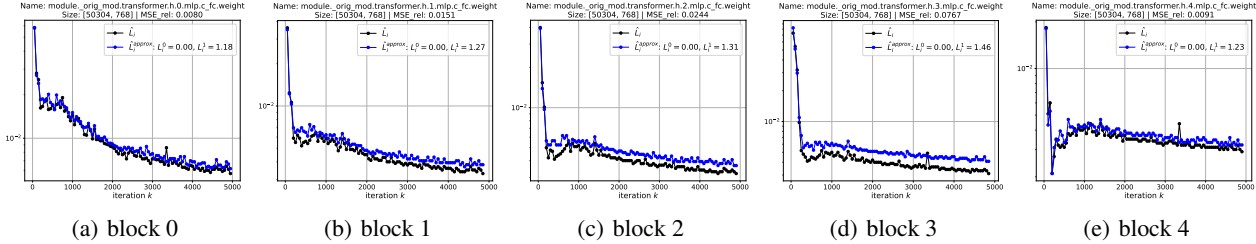

*Figure 3.* Validation of layer-wise $(L^0, L^1)$-smoothness for the group of parameters from the transformer blocks of `nanoGPT-124M` along unScion training trajectories. The group norms are non-Euclidean norms $\| \cdot \|_{(i)} = \sqrt{n_i/m_i} \| \cdot \|_{2 \to 2}$.

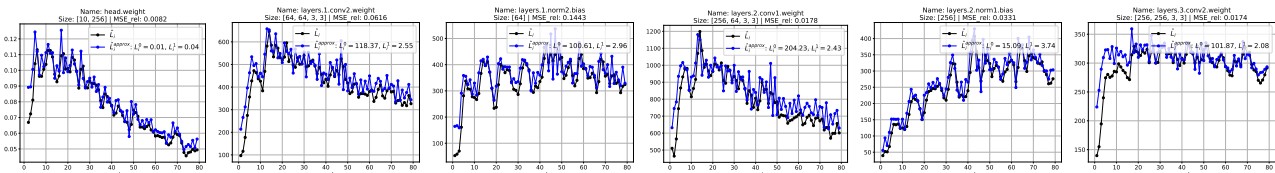

*Figure 4.* Empirical validation of $\ell*$-smoothness on computer vision tasks with CNN models in the *noiseless* setting.

**Examples** Consider the example in Lu et al. (2018, Proposition 2.1), where $\left\| \nabla^2 f(\mathbf{x}) \right\|_2 \leq p_r(\|\mathbf{x}\|_2)$ for some $r$-degree polynomial $p_r$. Then $f$ is $L$-smooth relative to $h$ with $L = \sup_{a>0} p_r(a)/(1 + a^r), h(\mathbf{x}) = \|\mathbf{x}\|_2^{r+2}/(r+2) + \|\mathbf{x}\|_2^2/2$. In this case, the subproblem has closed-form solutions only for $r = 1, 2, 3$, otherwise an extra root-finding oracle is required, which is unsatisfactory. More importantly, this framework only accommodates functions whose Hessian is bounded by polynomials of $\|\mathbf{x}\|_2$, which is restrictive. For instance, consider $g(\mathbf{x}) = \exp(\|\mathbf{x}\|_2^2/2)$. It holds that $\left\| \nabla^2 g(\mathbf{x}) \right\|_2 \leq (1 + \|\mathbf{x}\|_2^2) \exp(\|\mathbf{x}\|_2^2/2)$. Thus, the Hessian grows exponentially w.r.t. $\|\mathbf{x}\|_2$, rendering polynomial-based characterization inadequate. However, since $\|\nabla g(\mathbf{x})\|_2 = \|\mathbf{x}\|_2 \exp(\|\mathbf{x}\|_2^2/2)$, it follows that $g$ is $(0, \|\mathbf{x}\|_2 + 1)$-smooth. Evidently, generalized smoothness can model this function class better than relative smoothness. Furthermore, the rest of the examples in Lu et al. (2018) all conform to our $\ell*$-smoothness framework.

**Key limitations** In summary, relative smoothness has the following crucial limitations.

1. In practice, selecting a meaningful $(L, h)$ is challenging even with complete problem information. Finding solvable subproblems remains difficult.

2. For real-world applications where only a first-order oracle of $f$ is available (e.g., neural networks), relative smoothness lacks an effective solution. In contrast, generalized smoothness remains applicable by empirically estimating the link function $\ell$ (Zhang et al., 2020b).

## B. Experimental Details for LLMs

All experiments for the `nanoGPT` model (Karpathy, 2022)[5] are conducted using PyTorch[6] with Distributed Data Parallel (DDP)[7] across 8 NVIDIA 4090 GPUs (24GB each). The experiments are based on an open-source codebase which can be found at `https://github.com/artem-riabinin/Experiments-estimating-smoothness-for-NanoGPT-and-CNN/`. Theoretical formulations and details for the codebase can be found at Riabinin et al. (2025, Appendix E).

We mainly aimed at empirically validating our generalized smoothness model under non-Euclidean norms. In the procedure of training `nanoGPT` on the `Fineweb` dataset (Penedo et al., 2024) with `unScion` (Riabinin et al., 2025), we plot the

---

[5]We follow the nanoGPT (Karpathy, 2022) implementation of GPT-2 (Radford et al., 2019) model series.
[6]PyTorch Documentation. Available at: `https://pytorch.org/docs/stable/index.html`.
[7]Distributed Data Parallel (DDP) in PyTorch. Available at: `https://pytorch.org/docs/stable/notes/ddp.html`.

estimated *trajectory smoothness*

$$\hat{L}_i[k] := \frac{\|\nabla_i f_{\xi^{k+1}}(X^{k+1}) - \nabla_i f_{\xi^k}(X^k)\|_{(i)\star}}{\|X_i^{k+1} - X_i^k\|_{(i)}}, \tag{21}$$

and its approximation

$$\hat{L}_i^{\text{approx}}[k] := L_i^0 + L_i^1 \|\nabla_i f_{\xi^{k+1}}(X^{k+1})\|_{(i)\star}$$

as functions of the iteration index $k$. We observed similar findings to Riabinin et al. (2025, Appendix E.3.1), where there exhibits clear alignment between $\hat{L}_i[k]$ and $\hat{L}_i^{\text{approx}}[k]$, suggesting that our smoothness model holds approximately under non-Euclidean norms.

**Additional empirical results** In addition to the single block demonstration inFigures 1(a) to 1(c), Figure 3 shows more results for parameter groups at different transformer blocks of nanoGPT-124M. These complementary empirical findings further strengthens our theoretical $\ell*$-smoothness model. To isolate the potential influence of gradient noise, we also conduct *full-batch* trajectory smoothness verification in Figure 4, which consistently showcases a strong agreement between our $\ell*$-smoothness model and the model's curvature in practice.

**Practical guidelines** Riabinin et al. (2025, Appendix E.3.3) contains practical guidance for choosing the learning rate $\eta$, which is the ultimate goal of estimating $\ell$ and $G$. Their method is based on a previously recorded AdamW training trajectory. Since the smoothness model in Riabinin et al. (2025) is a special case of $\ell*$-smoothness, their methodology can be transferred to our setting.

## C. The Gain of Algorithmic Adaptivity

In this section, we present the omitted details for Section 2.2. The discussions underlying Appendices C.1.1 and C.1.2 are inspired by Gorbunov et al. (2025).

### C.1. Theoretical Justifications: Non-constant Link Functions

First, we present the omitted proof of Proposition 2.8.

*Proof of Proposition 2.8.* Let $s = \mathbf{1}_n^\top \mathbf{x}$ and $t = |s|$. Then

$$\nabla f(\mathbf{x}) = s^3 \mathbf{1}_n, \qquad \nabla^2 f(\mathbf{x}) = 3s^2 \mathbf{1}_n \mathbf{1}_n^\top.$$

Since $3t^2 \le 1 + 2t^3$ for all $t \ge 0$, we have

$$\|\nabla^2 f(\mathbf{x})\mathbf{h}\|_2 \le 3nt^2 \|\mathbf{h}\|_2 \le (n + 2nt^3) \|\mathbf{h}\|_2 = (n + 2\sqrt{n} \|\nabla f(\mathbf{x})\|_2) \|\mathbf{h}\|_2,$$

where $\|\nabla f(\mathbf{x})\|_2 = \sqrt{n}t^3$. Thus $f \in \mathcal{F}_{\hat{\ell}}(\|\cdot\|_2)$ with $\widehat{\ell}(\alpha) = n + 2\sqrt{n}\alpha$.

Similarly, since $\|\nabla f(\mathbf{x})\|_\infty = t^3$, for any $\mathbf{h} \in \mathbb{R}^n$,

$$\|\nabla^2 f(\mathbf{x})\mathbf{h}\|_\infty = 3t^2 |\mathbf{1}_n^\top \mathbf{h}| \le 3t^2 \|\mathbf{h}\|_1 \le (1 + 2 \|\nabla f(\mathbf{x})\|_\infty) \|\mathbf{h}\|_1.$$

Hence $f \in \mathcal{F}_{\widetilde{\ell}}(\|\cdot\|_1)$ with $\widetilde{\ell}(\alpha) = 1 + 2\alpha$.

**Important note.** The affine links above are chosen for simplicity and compatibility with the common $(L_0, L_1)$ form. The underlying operator-norm calculations are tight. In fact, the exact links are $\widehat{\ell}_{\text{exact}}(\alpha) = 3n^{2/3}\alpha^{2/3}$ and $\widetilde{\ell}_{\text{exact}}(\alpha) = 3\alpha^{2/3}$, yielding an even stronger dimension-dependent gap of order $n^{-2/3}$. □

Beyond the example function in (3), we additional examples that are widely encountered in practice. The unnormalized softmax logits, aka the logistic kernel function in (22), frequently appear in machine learning (Bishop & Nasrabadi, 2006), deep learning (Goodfellow et al., 2016), statistics (Keener, 2010; Wainwright, 2019), and statistical mechanics (Landau & Lifshitz, 2013). Similarly, the logistic regression function in (23) is a fundamental building block in the ML community.

### C.1.1. Unnormalized Softmax Logits

Consider the following logistic kernel function:

$$f(\mathbf{x}) := C \cdot \exp(\mathbf{w}^\top \mathbf{x} + b), \quad \mathbf{x}, \mathbf{w} \in \mathbb{R}^n, b \in \mathbb{R}, C \in \mathbb{R} \setminus \{0\}. \tag{22}$$

We focus on the unbounded domain like $\mathbb{R}^n$, which is often the case in reality. For bounded domains (satisfying Assumption J.2) considered in this paper, the discussions are deferred to Appendix C.1.3. We will first show that the Hessian $\nabla^2 f(\mathbf{x})$ is unbounded unless $\mathbf{w} = 0$, indicating $f$ is not $L$-smooth for any constant $L$. Then, we demonstrate that generalized smoothness can effectively handle this problem. Finally, we give characterizations of $\ell_*$-smoothness and $\ell$-smoothness, highlighting the edge of our formulation.

The gradient and Hessian of $f$ are given by

$$\nabla f(\mathbf{x}) = C \exp(\mathbf{w}^\top \mathbf{x} + b)\mathbf{w}, \quad \nabla^2 f(\mathbf{x}) = C \exp(\mathbf{w}^\top \mathbf{x} + b)\mathbf{w}\mathbf{w}^\top.$$

Then we have

$$\left\| \nabla^2 f(\mathbf{x}) \right\|_2 = |C| \exp(\mathbf{w}^\top \mathbf{x} + b) \left\| \mathbf{w} \right\|_2^2.$$

Since the domain of $\mathbf{x}$ is unbounded, we conclude that $\left\| \nabla^2 f(\mathbf{x}) \right\|_2$ is unbounded unless $\mathbf{w} = 0$. Hence, standard smoothness can not be applied here.

Next, we focus on $(L_0, L_1)$-smoothness. First, we argue that $(L_0, L_1)$-smoothness serves as a good remedy to the above failure case. Second, we reveal the gaps between $\ell$-smoothness and $\ell_*$-smoothness.

**$\ell$-smoothness**   We have

$$\left\| \nabla f(\mathbf{x}) \right\|_2 = |C| \exp(\mathbf{w}^\top \mathbf{x} + b) \left\| \mathbf{w} \right\|_2, \quad \left\| \nabla^2 f(\mathbf{x}) \right\|_2 = |C| \exp(\mathbf{w}^\top \mathbf{x} + b) \left\| \mathbf{w} \right\|_2^2.$$

Thus,

$$\left\| \nabla^2 f(\mathbf{x}) \right\|_2 = |C| \left\| \mathbf{w} \right\|_2 \cdot \left\| \nabla f(\mathbf{x}) \right\|_2,$$

which implies $f \in \mathcal{F}_{\widehat{\ell}}(\|\cdot\|_2)$ with $\widehat{\ell}(\alpha) = \|\mathbf{w}\|_2 \alpha$. Obviously, $(L_0, L_1)$-smoothness can model the curvature of this function more appropriately.

**$\ell_*$-smoothness**   Consider the norm pairs as $\|\cdot\| = \|\cdot\|_1$ and $\|\cdot\|_* = \|\cdot\|_\infty$. By Definition 2.1 in our manuscript, we have

$$\sup_{\mathbf{h} \in \mathbb{R}^n} \frac{\left\| \nabla^2 f(\mathbf{x})\mathbf{h} \right\|_\infty}{\|\mathbf{h}\|_1} = |C| \exp(\mathbf{w}^\top \mathbf{x} + b) \sup_{\mathbf{h} \in \mathbb{R}^n} \frac{\left\| \mathbf{w}\mathbf{w}^\top \mathbf{h} \right\|_\infty}{\|\mathbf{h}\|_1}$$

$$= |C| \exp(\mathbf{w}^\top \mathbf{x} + b) \|\mathbf{w}\|_\infty \sup_{\mathbf{h} \in \mathbb{R}^n} \frac{\left| \mathbf{w}^\top \mathbf{h} \right|}{\|\mathbf{h}\|_1}$$

$$= \left\| \nabla f(\mathbf{x}) \right\|_\infty \sup_{\mathbf{h} \in \Delta_d} \left| \mathbf{w}^\top \mathbf{h} \right|$$

$$= \left\| \nabla f(\mathbf{x}) \right\|_\infty \|\mathbf{w}\|_\infty.$$

Thus, $f \in \mathcal{F}_{\widetilde{\ell}}(\|\cdot\|_1)$ with $\widetilde{\ell}(\alpha) = \|\mathbf{w}\|_\infty \alpha$.

**Comparison**   Define the ratio between $\ell_*$-smoothness and $\ell$-smoothness as

$$\phi(\mathbf{w}) := \frac{\|\mathbf{w}\|_\infty}{\|\mathbf{w}\|_2} = \frac{\widetilde{\ell}(\alpha)}{\widehat{\ell}(\alpha)} \in \left[ \frac{1}{\sqrt{n}}, 1 \right].$$

From above, we can assert that our formulation is *strictly better* than that in Li et al. (2023a). Taking $\mathbf{w} = \mathbf{1}_n$ satisfies the left side of the inequality. In this case, $f \in \mathcal{F}_{\widehat{\ell}}(\|\cdot\|_2)$ with $\widehat{\ell}(\alpha) = \sqrt{n}\alpha$ and $f \in \mathcal{F}_{\widetilde{\ell}}(\|\cdot\|_1)$ with $\widetilde{\ell}(\alpha) = \alpha$. Clearly, our definition surpasses $\ell$-smoothness by a *dimension-dependent* factor.

**Even more practical case**  This time, we consider $\mathbf{w} \in \mathbb{R}^n$ as a random variable uniformly distributed on a unit sphere. Since the slopes of the generalized smooth functions $\widehat{\ell}, \widetilde{\ell}$ are now randomized, we measure the smoothness parameter in expectation. We clearly have $\|\mathbf{w}\|_2 = 1$ by definition. As for $\widetilde{\ell}$, it is well-known (Schechtman & Zinn, 2000; Boucheron et al., 2013) that

$$\mathbb{E}[\|\mathbf{w}\|_\infty] \preceq \sqrt{\frac{\log n}{n}}.$$

Hence, the (expected) ratio becomes

$$\phi_{\mathbb{E}}(\mathbf{w}) := \frac{\mathbb{E}[\|\mathbf{w}\|_\infty]}{\mathbb{E}[\|\mathbf{w}\|_2]} = \mathbb{E}[\|\mathbf{w}\|_\infty] \preceq \sqrt{\frac{\log n}{n}}.$$

Mathematically, we establish the bound $\widetilde{\ell}/\widehat{\ell} \preceq \sqrt{\log(n)/n}$ in expectation, demonstrating the quantitative advantage of $\ell*$-smoothness. This benefit becomes particularly pronounced when handling high-dimensional optimization problems, mirroring practical scenarios where our adaptive approach offers meaningful improvements over classical smoothness analysis.

### C.1.2. LOGISTIC REGRESSION

Consider the loss function of logistic regression:

$$f(\mathbf{x}) := C \cdot \log\left(1 + \exp(-\mathbf{w}^\top \mathbf{x})\right), \quad \mathbf{w}, \mathbf{x} \in \mathbb{R}^n, C \in \mathbb{R} \setminus \{0\}, \tag{23}$$

whose gradient and Hessian are given by

$$\nabla f(\mathbf{x}) = \frac{-C\mathbf{w}}{1 + \exp(\mathbf{w}^\top \mathbf{x})}, \quad \nabla^2 f(\mathbf{x}) = \frac{C\mathbf{w}\mathbf{w}^\top}{\exp(-\mathbf{w}^\top \mathbf{x})\left(1 + \exp(\mathbf{w}^\top \mathbf{x})\right)^2}.$$

Thus we have

$$\begin{aligned}
\left\|\nabla^2 f(\mathbf{x})\right\|_2 &= \frac{|C| \|\mathbf{w}\|_2^2}{\left(\exp(\frac{1}{2}\mathbf{w}^\top \mathbf{x}) + \exp(-\frac{1}{2}\mathbf{w}^\top \mathbf{x})\right)^2} \\
&\leq \frac{|C| \|\mathbf{w}\|_2^2}{\left(2\sqrt{\exp\left(\frac{1}{2}\mathbf{w}^\top \mathbf{x} - \frac{1}{2}\mathbf{w}^\top \mathbf{x}\right)}\right)^2} = \frac{|C| \|\mathbf{w}\|_2^2}{4},
\end{aligned}$$

implying that $f$ is $|C| \|\mathbf{w}\|_2^2 / 4$-smooth. This smoothness constant can become very large if $|C| \cdot \|\mathbf{w}\|_2 \gg 1$. Therefore, it's meaningful to seek an alternative. Let's consider the generalized smoothness notion as in Appendix C.1.1.

**$\ell$-smoothness**  We have

$$\|\nabla f(\mathbf{x})\|_2 = \frac{|C| \|\mathbf{w}\|_2}{1 + \exp(\mathbf{w}^\top \mathbf{x})}, \quad \left\|\nabla^2 f(\mathbf{x})\right\|_2 = \frac{|C| \|\mathbf{w}\|_2^2}{\exp(-\mathbf{w}^\top \mathbf{x})\left(1 + \exp(\mathbf{w}^\top \mathbf{x})\right)^2}.$$

Thus,

$$\begin{aligned}
\frac{\left\|\nabla^2 f(\mathbf{x})\right\|_2}{\|\nabla f(\mathbf{x})\|_2} &= \frac{(1 + \exp(\mathbf{w}^\top \mathbf{x})) \|\mathbf{w}\|_2}{\exp(-\mathbf{w}^\top \mathbf{x})\left(1 + \exp(\mathbf{w}^\top \mathbf{x})\right)^2} \\
&= \frac{\|\mathbf{w}\|_2}{1 + \exp(-\mathbf{w}^\top \mathbf{x})} \leq \|\mathbf{w}\|_2.
\end{aligned}$$

Since $\left\|\nabla^2 f(\mathbf{x})\right\|_2 \leq \|\mathbf{w}\|_2 \cdot \|\nabla f(\mathbf{x})\|_2, \forall \mathbf{x} \in \mathbb{R}^n$, we conclude that $f \in \mathcal{F}_{\widehat{\ell}}(\|\cdot\|_2)$ with $\widehat{\ell}(\alpha) = \|\mathbf{w}\|_2 \alpha$ (Chen et al., 2023c). Clearly, when $\|\mathbf{w}\|_2 \gg 1$, generalized smooth is more appropriate because it improves upon the standard smooth by a factor of $\|\mathbf{w}\|_2$.

$\ell*$**-smoothness** We follow a similar procedure as above:

$$\|\nabla f(\mathbf{x})\|_\infty = \frac{|C|\,\|\mathbf{w}\|_\infty}{1 + \exp(\mathbf{w}^\top \mathbf{x})},$$

and compute

$$
\begin{aligned}
\sup_{\mathbf{h}\in\mathbb{R}^n} \frac{\left\|\nabla^2 f(\mathbf{x})\mathbf{h}\right\|_\infty}{\|\mathbf{h}\|_1} &= \frac{|C|}{\exp(-\mathbf{w}^\top\mathbf{x})\left(1+\exp(\mathbf{w}^\top\mathbf{x})\right)^2} \sup_{\mathbf{h}\in\mathbb{R}^n} \frac{\left\|\mathbf{w}\mathbf{w}^\top\mathbf{h}\right\|_\infty}{\|\mathbf{h}\|_1} \\
&= \frac{|C|\,\|\mathbf{w}\|_\infty}{\exp(-\mathbf{w}^\top\mathbf{x})\left(1+\exp(\mathbf{w}^\top\mathbf{x})\right)^2} \sup_{\mathbf{h}\in\mathbb{R}^n} \frac{\left|\mathbf{w}^\top\mathbf{h}\right|}{\|\mathbf{h}\|_1} \\
&= \frac{\|\nabla f(\mathbf{x})\|_\infty}{1 + \exp(-\mathbf{w}^\top\mathbf{x})} \sup_{\mathbf{h}\in\Delta_d} \left|\mathbf{w}^\top\mathbf{h}\right| \\
&\leq \|\nabla f(\mathbf{x})\|_\infty \|\mathbf{w}\|_\infty.
\end{aligned}
$$

Therefore, we conclude that $f \in \mathcal{F}_{\widetilde{\ell}}(\|\cdot\|_1)$ with $\widetilde{\ell}(\alpha) = \|\mathbf{w}\|_\infty \alpha$.

**Comparison** Similarly, we can define the ratio function $\phi(\mathbf{w})$ as in Appendix C.1.1, and the same conclusions can be drawn: our definition surpasses $\ell$-smoothness by a *dimension-dependent* factor of $1/\sqrt{n}$.

**Logistic regression on a high-dimensional sphere** Similar to Appendix C.1.1, we now consider logistic regression on a high-dimensional sphere, with radius $R \gg 1$. In this case, $\|\mathbf{w}\|_2 = R \gg 1$, and $\mathbf{w}$ is uniformly distributed on the $R$-radius sphere. Note that $\mathbf{w}/R$ can be seen as a random variable uniformly distributed on the unit sphere, then we we also have $\phi_\mathbb{E}(\mathbf{w}) := \frac{\mathbb{E}[\|\mathbf{w}\|_\infty]}{\mathbb{E}[\|\mathbf{w}\|_2]} = \mathbb{E}[\|\mathbf{w}\|_\infty] \preceq \sqrt{\frac{\log n}{n}}$. Again, we verify the effectiveness and benefits of our $\ell*$-smoothness in this practical scenario.

### C.1.3. THE CASES FOR BOUNDED DOMAINS

The examples and justification in Appendices C.1.1 and C.1.2 targets unbounded domains like $\mathbb{R}^n$. Under Assumption J.2, the previous arguments are still valid and we emphasize that the advantage of $\ell*$-smoothness does not stem from unboundedness of the domain.

**Unnormalized Softmax Logits** For $f(\mathbf{x}) = C\exp(\mathbf{w}^\top\mathbf{x} + b), \mathbf{w}, \mathbf{x} \in Y := \{\mathbf{y} \in \mathbb{R}^n \mid \|\mathbf{y}\|_2 \leq R\}$. Then the standard smoothness constant is $L = \max_\mathbf{x}\|\nabla^2 f(\mathbf{x})\|_2 \leq |C|\exp(R^2 + b)R^2$. Unfortunately, it exhibits an exponential dependence on $R$, making traditional smoothness impractical. For instance, when $|C| = b = R = 5$, we get $L > 10^{15}$, which is clearly meaningless and necessitates a better formulation. One can verify that we still have $f \in \mathcal{F}_{\widehat{\ell}}(\|\cdot\|_2)$ with $\widehat{\ell}(\alpha) = \|\mathbf{w}\|_2\alpha$ and $f \in \mathcal{F}_{\widetilde{\ell}}(\|\cdot\|_1)$ with $\widetilde{\ell}(\alpha) = \|\mathbf{w}\|_\infty\alpha$.

**Logistic Regression** For $f(\mathbf{x}) = 4C\log(1 + \exp(-\mathbf{w}^\top\mathbf{x})), \mathbf{w}, \mathbf{x} \in Y$, we have $L = \max_\mathbf{x}\|\nabla^2 f(\mathbf{x})\|_2 \leq |C|\,\|\mathbf{w}\|_2^2$ and $f \in \mathcal{F}_{\widehat{\ell}}(\|\cdot\|_2), \widehat{\ell}(\alpha) = \|\mathbf{w}\|_2\alpha; f \in \mathcal{F}_{\widetilde{\ell}}(\|\cdot\|_1), \widetilde{\ell}(\alpha) = \|\mathbf{w}\|_\infty\alpha$. The example remains meaningful when the domain and $|C|$ is relatively large.

**Main takeaways** In general, it makes sense to consider $\ell*$-smoothness even if standard or global smoothness holds, and for some functions like unnormalized softmax logits or logistic regression, $\ell*$-smoothness provably improves upon $\ell$-smoothness with *dimension-dependent* gains, regardless of the domain being bounded or not.

### C.2. Theoretical Justifications: Constant Link Functions

Consider the function $h(\mathbf{x}) = \mathbf{x}^\top(\mathbf{1}_n - \mathbf{e}_1)(\mathbf{1}_n - \mathbf{e}_1)^\top\mathbf{x}/2, \mathbf{x} \in \Delta_n$, where $\Delta_n = \{\mathbf{x} \geq \mathbf{0}, \mathbf{1}_n^\top\mathbf{x} = 1 \mid \mathbf{x} \in \mathbb{R}^n\}$ is the $(n-1)$-dimensional simplex. We have the following proposition.

**Proposition C.1.** *(i)* $h \in \mathcal{F}_{\widehat{\ell}}(\|\cdot\|_2)$ *with* $\widehat{\ell}(\cdot) \equiv n - 1$; *(ii)* $h \in \mathcal{F}_{\widetilde{\ell}}(\|\cdot\|_1)$ *with* $\widetilde{\ell}(\cdot) \equiv 1$.

*Proof.* Denote $(\mathbf{1}_n - \mathbf{e}_1)(\mathbf{1}_n - \mathbf{e}_1)^\top$ by $A$. For the quadratic function $h(\mathbf{x}) = \frac{1}{2}\mathbf{x}^\top A\mathbf{x}$ where $A = A^\top$, we have

$$\nabla h(\mathbf{x}) = A\mathbf{x}, \quad \nabla^2 h(\mathbf{x}) = A. \tag{24}$$

Since $A = (\mathbf{1}_n - \mathbf{e}_1)(\mathbf{1}_n - \mathbf{e}_1)^\top$ is a rank-1 matrix, we have

$$\left\|\nabla^2 h(\mathbf{x})\right\|_2 = \|\mathbf{1}_n - \mathbf{e}_1\|_2 \|\mathbf{1}_n - \mathbf{e}_1\|_2 = \sqrt{n-1} \cdot \sqrt{n-1} = n-1. \tag{25}$$

Hence, for $h \in \mathcal{F}_{\widehat{\ell}}(\|\cdot\|_2)$, we have $\widehat{\ell}(\cdot) \equiv n-1$. By Definition 2.1, for $h \in \mathcal{F}_{\widetilde{\ell}}(\|\cdot\|)$, we aim to find the link function $\widetilde{\ell}(\cdot)$ such that

$$\forall \mathbf{u} \in \Delta_n, \left\|\nabla^2 h(\mathbf{x})\mathbf{u}\right\|_\infty \leq \widetilde{\ell}(\|\nabla h(\mathbf{x})\|_\infty) \|\mathbf{u}\|_1$$
$$\Longleftrightarrow \sup_{\mathbf{u} \in \Delta_n} \left\|\nabla^2 h(\mathbf{x})\mathbf{u}\right\|_\infty \leq \widetilde{\ell}(\|\nabla h(\mathbf{x})\|_\infty). \tag{26}$$

We proceed by computing

$$\sup_{\mathbf{u} \in \Delta_n} \|A\mathbf{u}\|_\infty = \sup_{\mathbf{u} \in \Delta_n} \left\|(\mathbf{1}_n - \mathbf{e}_1)[(\mathbf{1}_n - \mathbf{e}_1)^\top \mathbf{u}]\right\|_\infty$$
$$= \sup_{\mathbf{u} \in \Delta_n} (\mathbf{1}_n - \mathbf{e}_1)^\top \mathbf{u} = \sup_{\mathbf{u} \in \Delta_n} \sum_{i=2}^n u_i = 1, \tag{27}$$

where we make use of $\|\mathbf{1}_n - \mathbf{e}_1\|_\infty = 1$. Therefore, we conclude that $\widetilde{\ell}(\cdot) \equiv 1$. $\square$

Under $\ell$-smoothness, the function $h$ inevitably has a smoothness parameter that scales linearly to the dimension $n$. Whereas, under $\ell*$-smoothness, the dependence of the dimension no longer exists. The additional factor $n-1$ will inevitably incur a *dimension-dependent* convergence rate, which will be stated explicitly in Appendix C.5.

### C.3. Empirical Justifications

We focus on the following example function:

$$h(\mathbf{x}) = \frac{\left(\sum_{i=1}^n x_i^2\right)^2}{4} + \prod_{i=1}^n x_i^2 + \sum_{i=1}^n e^{5x_i}, \quad \forall \mathbf{x} \in \Delta_n. \tag{28}$$

For the Euclidean setup $h \in \mathcal{F}_{\widehat{\ell}}(\|\cdot\|_2)$ as well as the non-Euclidean setup $h \in \mathcal{F}_{\widetilde{\ell}}(\|\cdot\|_1)$, the smoothness can be modeled by

$$\left\|\nabla^2 h(\mathbf{x})\right\|_2 \leq \widehat{\ell}(\|\nabla h(\mathbf{x})\|_2), \quad \text{and} \quad \sup_{\mathbf{u} \in \Delta_n} \left\|\nabla^2 h(\mathbf{x})\mathbf{u}\right\|_\infty \leq \widetilde{\ell}(\|\nabla h(\mathbf{x})\|_\infty), \tag{29}$$

respectively. For various values of $n$, we conduct numerical experiments to examine the coarse forms of $\widehat{\ell}(\cdot)$ and $\widetilde{\ell}(\cdot)$. Specifically, we compute and plot $\left\|\nabla^2 h(\mathbf{x})\right\|_2$ against $\|\nabla h(\mathbf{x})\|_2$ for $h \in \mathcal{F}_{\widehat{\ell}}(\|\cdot\|_2)$, leveraging PyTorch's auto-grad functionality (Paszke et al., 2019) to calculate gradients and Hessians. Similarly, for $h \in \mathcal{F}_{\widetilde{\ell}}(\|\cdot\|_1)$, we compute $\sup_{\mathbf{u} \in \Delta_n} \left\|\nabla^2 h(\mathbf{x})\mathbf{u}\right\|_\infty$ and $\|\nabla h(\mathbf{x})\|_\infty$. The main challenge lies in evaluating $\sup_{\mathbf{u} \in \Delta_n} \left\|\nabla^2 h(\mathbf{x})\mathbf{u}\right\|_\infty$, which, as a special case of the $p \to q$ matrix norm problem (Bhaskara & Vijayaraghavan, 2011), can be approximated using constant-factor algorithms for $p \geq 2 \geq q$ (Khot & Naor, 2012; Pisier, 2012; Bhattiprolu et al., 2019). In our $\infty \to 1$ case, it suffices to model it as a linear programming problem with constraints $\mathbf{u} \in \Delta_n$ and then solve it numerically. Consequently, we are able to plot $\sup_{\mathbf{u} \in \Delta_n} \left\|\nabla^2 h(\mathbf{x})\mathbf{u}\right\|_\infty$ against $\|\nabla h(\mathbf{x})\|_\infty$, as well. The results, shown in Figure 5, suggest that the generalized smooth relation in both cases can be approximated by the affine functions shown below:

$$\widehat{\ell}(\alpha) = \widehat{L}_0 + \widehat{L}_1\alpha, \quad \widetilde{\ell}(\alpha) = \widetilde{L}_0 + \widetilde{L}_1\alpha. \tag{30}$$

We find this characterization reasonable based on observations from Figure 5: when $n > 20$, the scattered points align into a straight line. To this end, we fit the values of $\widehat{L}_0, \widehat{L}_1, \widetilde{L}_0,$ and $\widetilde{L}_1$ using 500 randomly sampled points from the simplex $\Delta_n$.

Based on the modeling in (30), we can see that the slopes $\widehat{L}_1$ and $\widetilde{L}_1$ determine the magnitude of smoothness along the optimization trajectory (e.g., for some constant $0 < G < \infty$, $\widetilde{\ell}(G) = \widetilde{L}_0 + \widetilde{L}_1 G$), to a great extent. To clarify, note that the definition of $G$ in (7) has a closed-form solution:

$$G = \sup\left\{\alpha \in \mathbb{R}_+ \,\middle|\, \alpha^2 \leq 2F\left(\widetilde{L}_0 + \widetilde{L}_1\alpha\right)\right\}$$
$$= F\widetilde{L}_1 + \sqrt{2F\widetilde{L}_0 + F^2\widetilde{L}_1^2} \leq 2F\widetilde{L}_1 + \sqrt{2F\widetilde{L}_0}, \tag{31}$$

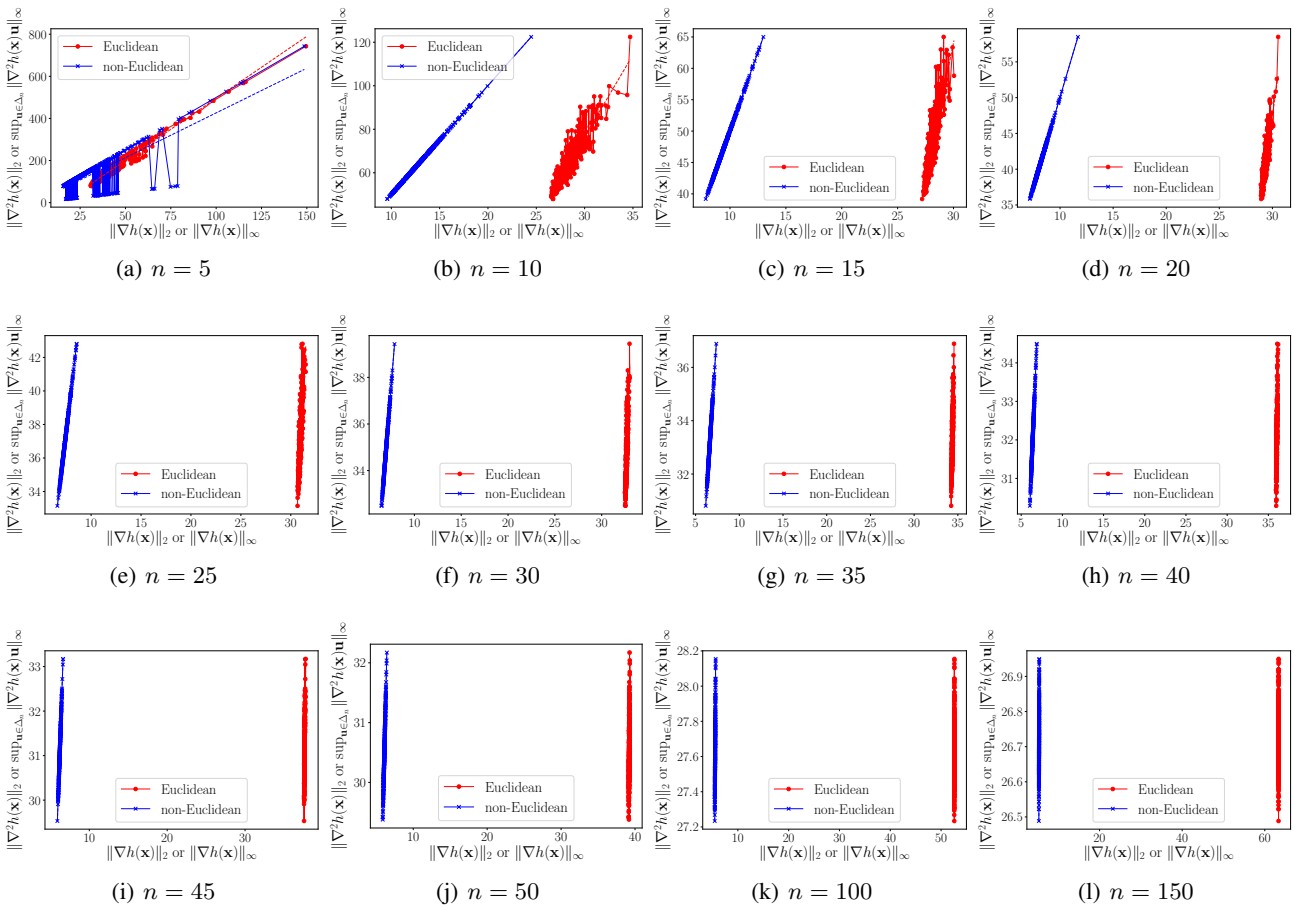

*Figure 5.* Comparison of $\mathcal{F}_{\widehat{\ell}}(\|\cdot\|_2)$ and $h \in \mathcal{F}_{\widetilde{\ell}}(\|\cdot\|_1)$ generalized smooth functions class for different values of $n$.

where $f(\mathbf{x}_0) - f^*$ is denoted by $F$. The above relation implies that $G$ grows linearly w.r.t. $\widetilde{L}_1$ and $\sqrt{\widetilde{L}_0}$. As a result, by (7), we compute the following effective smoothness parameter:

$$L = \widetilde{\ell}(2G) = \widehat{L}_0 + 2\widehat{L}_1 G \leq \widehat{L}_0 + 4F\widehat{L}_1^2 + 2F\widehat{L}_1\sqrt{2F\widetilde{L}_0}. \tag{32}$$

Therefore, we conclude that $\widetilde{L}_1$ is the dominant term affecting the smoothness parameter $L$.

Due to the reasoning above, we concentrate on the slopes $\widetilde{L}_1, \widehat{L}_1$ to analyze and compare the characteristics of the link functions $\widehat{\ell}$ and $\widetilde{\ell}$. To further explore this, we examine the ratio $\widetilde{L}_1/\widehat{L}_1$ for various values of $n$. Specifically, we vary $n$ from 6 to 198 in increments of 3 and fit a function of the form $g(n) = a \cdot n^{-b}, a, b > 0$. The results, shown in Figure 6, reveal that the gap between $\widetilde{\ell}$ and $\widehat{\ell}$ is approximately at the order of $O(n^{-0.4})$.

### C.4. Relationship with Block Diagonal Hessians

Zhang et al. (2024c) argue that Hessians in modern neural networks often exhibit block-wise structural properties, and that Transformers further display strong *block heterogeneity*, meaning that Hessian spectra can vary significantly across parameter blocks. Motivated by this observation, An et al. (2025) propose a block diagonal smoothness model for structured optimization. Below, we clarify that this type of smoothness is a **strict special case** of our $\ell*$-smoothness.

Let $\mathbf{x}, \mathbf{h} \in \mathbb{R}^d$, $\nabla f(\mathbf{x}) \in \mathbb{R}^d$, $\nabla^2 f(\mathbf{x}) \in \mathbb{R}^{d \times d}$, and let $\mathbf{L} \in \mathbb{R}^{d \times d}$ be positive definite. Define

$$\|\mathbf{h}\|_{\mathbf{L}} := \left\|\mathbf{L}^{1/2}\mathbf{h}\right\|_2, \qquad \|\mathbf{g}\|_{\mathbf{L}^{-1}} := \left\|\mathbf{L}^{-1/2}\mathbf{g}\right\|_2.$$

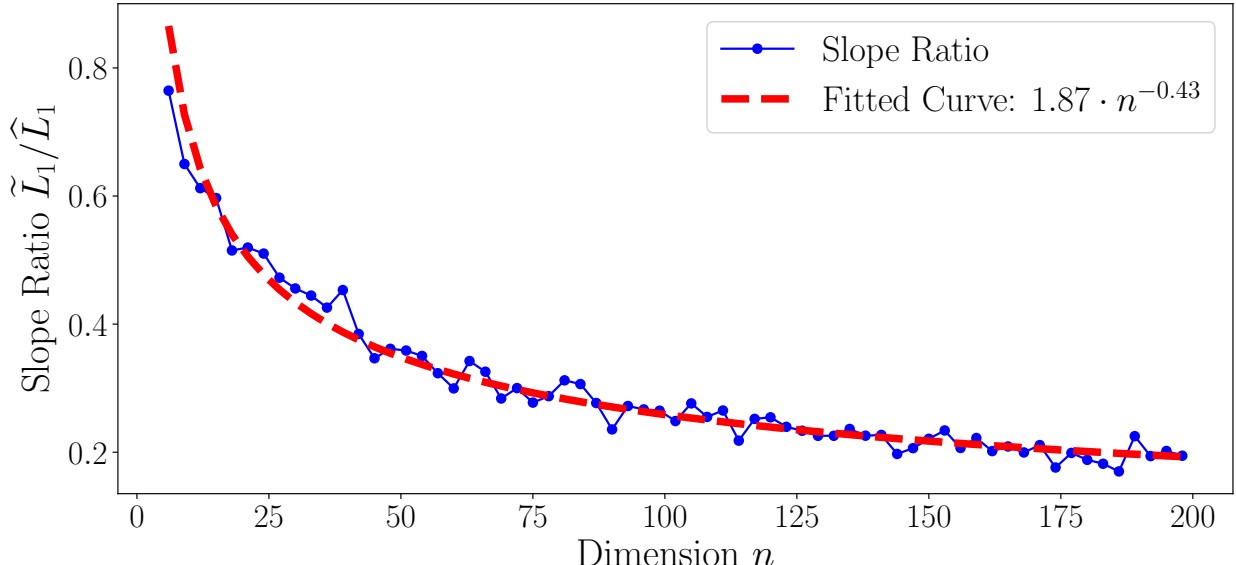

Figure 6. The slope ratio $\widetilde{L}_1/\widehat{L}_1$ w.r.t. dimension $n$ and the fitted curve.

The 1-smoothness assumption w.r.t. $\|\cdot\|_{\mathbf{L}}$ is equivalent to

$$-\mathbf{L} \preceq \nabla^2 f(\mathbf{x}) \preceq \mathbf{L}, \qquad \forall \mathbf{x} \in \mathbb{R}^d.$$

Therefore, for any $\mathbf{h} \in \mathbb{R}^d$,

$$\begin{aligned}
\left\|\nabla^2 f(\mathbf{x})\mathbf{h}\right\|_{\mathbf{L}^{-1}} &= \left\|\mathbf{L}^{-1/2}\nabla^2 f(\mathbf{x})\mathbf{h}\right\|_2 \\
&\leq \left\|\mathbf{L}^{-1/2}\nabla^2 f(\mathbf{x})\mathbf{L}^{-1/2}\right\|_{\mathrm{op}} \left\|\mathbf{L}^{1/2}\mathbf{h}\right\|_2 \leq \|\mathbf{h}\|_{\mathbf{L}}.
\end{aligned}$$

Hence, $f \in \mathcal{F}_\ell(\|\cdot\|_{\mathbf{L}})$ with the constant link function $\ell(\alpha) \equiv 1$.

When $\mathbf{L}$ is block diagonal, e.g.,

$$\mathbf{L} = \mathrm{diag}(L_1 I_{d_1}, \ldots, L_m I_{d_m}), \qquad L_i > 0, \quad \sum_{i=1}^{m} d_i = d,$$

the induced norm $\|\cdot\|_{\mathbf{L}}$ naturally assigns different curvature scales to different parameter blocks, thereby capturing the block heterogeneity emphasized by Zhang et al. (2024c). Our framework is more general in the following aspects:

(i) it allows arbitrary norm geometries beyond quadratic matrix-induced norms such as $\|\cdot\|_{\mathbf{L}}$;

(ii) it allows non-constant link functions $\ell(\cdot)$, so the local curvature scale can grow with the gradient magnitude;

(iii) it treats the primal norm and the dual norm explicitly, which is essential for non-Euclidean algorithms such as mirror descent.

Therefore, block diagonal smoothness can be viewed as a constant-link, matrix-induced special case of $\ell_*$-smoothness.

**Coverage of existing (generalized) smoothness models**   More broadly, our $\ell_*$-smoothness provides a unified language for many (non-Euclidean) smoothness models used in modern optimization. For example, diagonal or coordinate-wise smoothness can be expressed through weighted $\ell_2$ or $\ell_\infty/\ell_1$ geometries; layer-wise and operator-norm smoothness models

can be expressed through block-specific norms; and anisotropic or directional smoothness can be interpreted as choosing norms that reflect non-uniform curvature across directions or parameter groups (Crawshaw et al., 2022; Riabinin et al., 2025; Mishkin et al., 2024; Liu et al., 2025b; Jiang et al., 2025a;b; Yu et al., 2026; Tao et al., 2026). In this sense, as the first non-Euclidean generalized smoothness model, $\ell*$-smoothness is expressive enough to cover a wide range of existing smoothness conditions while also allowing the curvature scale to adapt to the gradient norm.

### C.5. Comparison on the Convergence Rates

Finally, we explicitly discuss the improvement in convergence rates. The similar discussions traces back to Nemirovski et al. (2009, (2.59) and (2.60)) and is further developed in Lan (2020, Section 3.2). Consider a generalized smooth function $f$ defined on the simplex $\Delta_n$, where $f$ satisfies $f \in \mathcal{F}_{\widehat{\ell}}(\|\cdot\|_2)$ and $f \in \mathcal{F}_{\widetilde{\ell}}(\|\cdot\|)$. According to Li et al. (2023a, Theorem 4.2), gradient descent achieves a convergence rate of

$$\frac{\widehat{\ell}(\|\nabla f(\mathbf{x}_0)\|_2)\|\mathbf{x}_0 - \mathbf{x}_*\|_2^2}{2T} \leq \frac{\widehat{\ell}(\|\nabla f(\mathbf{x}_0)\|_2)}{2T} \triangleq R_1(T), \tag{33}$$

where we use $\|\mathbf{x}_0 - \mathbf{x}_*\|_2^2 \leq \sup_{\mathbf{x},\mathbf{y}\in\Delta_n}\|\mathbf{x} - \mathbf{y}\|_2^2 = 1$. For mirror descent, as established in Theorem 3.5, the convergence rate is given by

$$\frac{\widetilde{\ell}(G)B(\mathbf{x}_*, \mathbf{x}_0)}{T} \leq \frac{\widetilde{\ell}(G)\ln n}{T} \triangleq R_2(T), \tag{34}$$

where the diameter of the simplex is specified as $\sqrt{\ln n}$ (Nemirovski et al., 2009). First, we consider the example function $h(\mathbf{x}) = \mathbf{x}^\top(\mathbf{1}_n - \mathbf{e}_1)(\mathbf{1}_n - \mathbf{e}_1)^\top\mathbf{x}/2, \mathbf{x} \in \Delta_n$ in Appendix C.2. By Proposition C.1, we have $h \in \mathcal{F}_{\widehat{\ell}}(\|\cdot\|_2)$ and $h \in \mathcal{F}_{\widetilde{\ell}}(\|\cdot\|)$ with $\widehat{\ell}(\cdot) \equiv n - 1$ and $\widetilde{\ell}(\cdot) \equiv 1$. Hence, the above convergence rates are valid and the ratio of the convergence rates is

$$\frac{R_2(T)}{R_1(T)} = \frac{\widetilde{\ell}(G)\ln n}{\widehat{\ell}(\|\nabla f(\mathbf{x}_0)\|_2)} = \frac{\ln n}{n - 1}, \tag{35}$$

suggesting that mirror descent defined in (8) provably improves the convergence rate by a factor of $n/\ln n$. The effect will become prominent when the dimension is large.

Next, our focal point shifts to the example function $h(\cdot)$ defined in (28). In view of discussions made in Appendix C.3, we conclude that $h \in \mathcal{F}_{\widehat{\ell}}(\|\cdot\|_2)$ and $h \in \mathcal{F}_{\widetilde{\ell}}(\|\cdot\|)$ with $\widehat{\ell}$ and $\widetilde{\ell}$ satisfying (30). Moreover, we have $\widetilde{L}_1/\widehat{L}_1 \simeq n^{-0.4}$. Then, the ratio of the convergence rates can be written as

$$\frac{R_2(T)}{R_1(T)} = \frac{\widetilde{\ell}(G)\ln n}{\widehat{\ell}(\|\nabla f(\mathbf{x}_0)\|_2)} \simeq \frac{\widetilde{\ell}(\|\nabla f(\mathbf{x}_0)\|_\infty)\ln n}{\widehat{\ell}(\|\nabla f(\mathbf{x}_0)\|_2)} \simeq \frac{\widetilde{L}_1\ln n}{\widehat{L}_1} \simeq \frac{\ln n}{n^{0.4}}, \tag{36}$$

underlining the advantage of mirror descent over gradient descent by reducing the dependency on the dimensionality factor. Obviously, the justifications for non-constant link functions in Appendix C.1 also follow from the above discussions, which we omit for simplicity.

## D. Properties of Generalized Smooth Function Classes

In this section, we demonstrate some basic properties of the function class $\mathcal{F}_\ell(\|\cdot\|)$ as well as $\mathcal{F}_{\ell,r}(\|\cdot\|)$. Lemma D.1 is a generalized version of Lemma A.4 of Li et al. (2023a), which only supports $f \in \mathcal{F}_\ell(\|\cdot\|_2)$. Since the original derivations did not utilize the inherent property of the Euclidean norm, i.e., $\langle\cdot,\cdot\rangle = \|\cdot\|_2^2$, we can adapt its proof to accommodate an arbitrary norm $\|\cdot\|$.

**Lemma D.1.** *Let $f \in \mathcal{F}_\ell(\|\cdot\|)$. For any $\widetilde{\mathbf{x}}, \mathbf{x} \in \mathcal{X}$, denote $\mathbf{x}(s) := s\widetilde{\mathbf{x}} + (1 - s)\mathbf{x}$. If $\mathbf{x}(s) \in \mathcal{X}$ holds for all $0 \leq s \leq 1$ and $\|\widetilde{\mathbf{x}} - \mathbf{x}\| \leq \frac{G}{\ell(\|\nabla f(\mathbf{x})\|_* + G)}$ for any $G \in \mathbb{R}_{++}$, then we have*

$$\max_{0 \leq s \leq 1}\|\nabla f(\mathbf{x}(s))\|_* \leq \|\nabla f(\mathbf{x})\|_* + G. \tag{37}$$

*Proof.* First, define $g(s) := \|\nabla f(\mathbf{x}(s))\|_*$ for $0 \leq s \leq 1$. Since $\mathbf{x}(s) \in \mathcal{X}$, $\mathbf{g}(s)$ is differentiable almost everywhere by Definition 2.1. Hence, the following relation holds almost everywhere over $(0, 1)$:

$$
\begin{aligned}
g'(s) = \lim_{t \to s} \frac{g(t) - g(s)}{t - s} &\leq \lim_{t \to s} \frac{\|\nabla f(\mathbf{x}(t)) - \nabla f(\mathbf{x}(s))\|_*}{t - s} \\
&= \left\| \lim_{t \to s} \frac{\nabla f(\mathbf{x}(t)) - \nabla f(\mathbf{x}(s))}{t - s} \right\|_* = \left\| \nabla^2 f(\mathbf{x}(s))(\widetilde{\mathbf{x}} - \mathbf{x}) \right\|_* \leq \ell(g(s)) \|\widetilde{\mathbf{x}} - \mathbf{x}\|,
\end{aligned}
\tag{38}
$$

where the first inequality uses the triangle inequality and the last inequality uses Definition 2.1. Next, we define $h(u) := \ell(u) \|\widetilde{\mathbf{x}} - \mathbf{x}\|$ and $k(u) := \int_0^u 1/h(v)\mathrm{d}v$. We immediately conclude that $g'(s) \leq h(g(s))$ holds almost everywhere over $(0, 1)$, which satisfies the condition of a generalized version of Grönwall's inequality (Li et al., 2023a, Lemma A.3). Hence, we have

$$
k(\|\nabla f(\widetilde{\mathbf{x}})\|_*) = k(g(1)) \leq k(g(0)) + 1 = k(\|\nabla f(\mathbf{x}\|_*)) + 1.
\tag{39}
$$

Based on this result, we can derive

$$
\begin{aligned}
k(\|\nabla f(\widetilde{\mathbf{x}})\|_*) \|\widetilde{\mathbf{x}} - \mathbf{x}\| &\leq k(g(0)) \|\widetilde{\mathbf{x}} - \mathbf{x}\| + \|\widetilde{\mathbf{x}} - \mathbf{x}\| \\
&\leq \int_0^{g(0)} \frac{\|\widetilde{\mathbf{x}} - \mathbf{x}\|}{\ell(u) \|\widetilde{\mathbf{x}} - \mathbf{x}\|} \mathrm{d}u + \frac{G}{\ell(\|\nabla f(\mathbf{x})\|_* + G)} \\
&\leq \int_0^{\|\nabla f(\mathbf{x})\|_*} \frac{1}{\ell(u)} \mathrm{d}u + \int_{\|\nabla f(\mathbf{x})\|_*}^{\|\nabla f(\mathbf{x})\|_* + G} \frac{1}{\ell(u)} \mathrm{d}u \\
&= k(\|\nabla f(\mathbf{x})\|_* + G) \|\widetilde{\mathbf{x}} - \mathbf{x}\|,
\end{aligned}
\tag{40}
$$

where the second inequality uses the bound for $\|\widetilde{\mathbf{x}} - \mathbf{x}\|$ and the last inequality use the non-decreasing property of $\ell$. By the definition of $k$, it is obvious to see that $k$ is increasing. Then we can conclude that

$$
\|\nabla f(\widetilde{\mathbf{x}})\|_* \leq \|\nabla f(\mathbf{x})\|_* + G.
\tag{41}
$$

Note that RHS of (41) involves the anchor point $\mathbf{x}$. Under the given conditions, the closed line segment between $\widetilde{\mathbf{x}}$ and $\mathbf{x}$ is contained in $\mathcal{X}$. Then, for any $0 \leq \tilde{s} \leq 1$, we also have $\{\mathbf{x}(s) | 0 \leq s \leq \tilde{s}\} \subseteq \mathcal{X}$ and $\|\mathbf{x}(\tilde{s}) - \mathbf{x}\| \leq \frac{G}{\ell(\|\nabla f(\mathbf{x})\|_* + G)}$. Therefore, we can invoke (41) to show that

$$
\|\nabla f(\mathbf{x}(\tilde{s}))\|_* \leq \|\nabla f(\mathbf{x})\|_* + G, \quad \forall \tilde{s} \in [0, 1].
\tag{42}
$$

Maximizing over $0 \leq \tilde{s} \leq 1$ yields the desired result. $\qquad\square$

In the following, we provide the omitted proof of Proposition 2.6.

*Proof of Proposition 2.6.* The proof is very similar to that of Li et al. (2023a, Proposition 3.2), except that the derivations based on $\|\cdot\|_2$ should be replaced by an arbitrary norm $\|\cdot\|$.

**Part 1.** $\mathcal{F}_{\ell,r}(\|\cdot\|) \subseteq \mathcal{F}_\ell(\|\cdot\|)$.

Given an $\mathbf{x} \in \mathcal{X}$ where $\nabla^2 f(\mathbf{x})$ exists, according to Definition 2.4, the following relation holds

$$
\|\nabla f(\mathbf{x} + s\mathbf{h}) - \nabla f(\mathbf{x})\|_* \leq \ell(\|\nabla f(\mathbf{x})\|_*) \cdot s \|\mathbf{h}\|
\tag{43}
$$

for any $\mathbf{h} \in \mathcal{X}$ and any $s \in \mathbb{R}_{++}$ satisfying $s \|\mathbf{h}\| \leq r(\|\nabla f(\mathbf{x})\|_*)$. Then, based on the continuity of the dual norm, we deduce that

$$
\begin{aligned}
\left\| \nabla^2 f(\mathbf{x}) \mathbf{h} \right\|_* &= \left\| \lim_{s \to 0} \frac{\nabla f(\mathbf{x} + s\mathbf{h}) - \nabla f(\mathbf{x})}{s} \right\|_* \\
&= \lim_{s \to 0} \left\| \frac{\nabla f(\mathbf{x} + s\mathbf{h}) - \nabla f(\mathbf{x})}{s} \right\|_* \leq \ell(\|\nabla f(\mathbf{x})\|_*) \cdot \|\mathbf{h}\|.
\end{aligned}
\tag{44}
$$

Next, we prove that $\nabla^2 f(\mathbf{x})$ exists almost everywhere. We briefly list the following two key steps; the detailed derivations can be found in Li et al. (2023a, Proposition 3.2).

1. Invoke Rademacher's Theorem (Evans, 2018) to show that $f$ is twice differentiable almost everywhere within the ball $\mathcal{B}(\mathbf{x}, r(\|\nabla f(\mathbf{x})\|_*))$.

2. Leverage the covering technique to show that $f$ is twice differentiable almost everywhere within the domain $\mathcal{X}$.

**Part 2. Under Assumption 2.5, $\mathcal{F}_\ell(\|\cdot\|) \subseteq \mathcal{F}_{\widetilde{\ell},\widetilde{r}}(\|\cdot\|)$ where $\widetilde{\ell}(\alpha) = \ell(\alpha + G), \widetilde{r}(\alpha) = \frac{G}{\widetilde{\ell}(\alpha)}$ for any $G \in \mathbb{R}_{++}$.**

First, we show that $\mathcal{B}(\mathbf{x}, \widetilde{r}(\|\nabla f(\mathbf{x})\|_*)) \subseteq \mathcal{X}$ for any $\mathbf{x} \in \mathcal{X}$ by contradiction[8]. For any $\widetilde{\mathbf{x}} \in \mathcal{B}(\mathbf{x}, \widetilde{r}(\|\nabla f(\mathbf{x})\|_*))$, suppose $\widetilde{\mathbf{x}} \notin \mathcal{X}$. Define

$$\mathbf{x}(s) := s\widetilde{\mathbf{x}} + (1 - s)\mathbf{x}, \ 0 \leq s \leq 1 \text{ and } s_b := \inf \{s | \mathbf{x}(s) \notin \mathcal{X}\}. \tag{45}$$

By the definition of $s_b$ and Assumption 2.5, we have $\lim_{s \to s_b} f(\mathbf{x}(s)) = \infty$ and $\mathbf{x}(s) \in \mathcal{X}$ for $0 \leq s < s_b$. Then we have

$$\begin{aligned}
\forall s \in [0, s_b) : f(\mathbf{x}(s)) &= f(\mathbf{x}) + \int_0^s \langle \nabla f(\mathbf{x}(u)), \widetilde{\mathbf{x}} - \mathbf{x} \rangle \mathrm{d}u \\
&\leq f(\mathbf{x}) + s \cdot \max_{0 \leq u \leq s} \|\nabla f(\mathbf{x}(u))\|_* \cdot \|\widetilde{\mathbf{x}} - \mathbf{x}\| \\
&\leq f(\mathbf{x}) + (\|\nabla f(\mathbf{x})\|_* + G) \|\widetilde{\mathbf{x}} - \mathbf{x}\| < \infty,
\end{aligned} \tag{46}$$

where the last inequality uses Lemma D.1 because the following condition is satisfied

$$\|\widetilde{\mathbf{x}} - \mathbf{x}\| \leq \widetilde{r}(\|\nabla f(\mathbf{x})\|_*) = G/\ell(\|\nabla f(\mathbf{x})\|_* + G) \tag{47}$$

by definition. This contradicts $\lim_{s \to s_b} f(\mathbf{x}(s)) = \infty$. Thus we have $\widetilde{\mathbf{x}} \in \mathcal{X}$ and consequently, $\mathcal{B}(\mathbf{x}, \widetilde{r}(\|\nabla f(\mathbf{x})\|_*)) \subseteq \mathcal{X}$. Next, we prove (2) in Definition 2.4. For any $\mathbf{x}_1, \mathbf{x}_2 \in \mathcal{B}(\mathbf{x}, \widetilde{r}(\|\nabla f(\mathbf{x})\|_*))$, define $\mathbf{x}_{1,2}(s) := s\mathbf{x}_1 + (1 - s)\mathbf{x}_2$ for $0 \leq s \leq 1$. Since $\mathbf{x}_{1,2}(s) \in \mathcal{B}(\mathbf{x}, \widetilde{r}(\|\nabla f(\mathbf{x})\|_*))$, we obtain

$$\begin{aligned}
\|\nabla f(\mathbf{x}_1) - \nabla f(\mathbf{x}_2)\|_* &= \left\|\int_0^1 \nabla^2 f(\mathbf{x}_{1,2}(s)) (\mathbf{x}_1 - \mathbf{x}_2)\mathrm{d}s\right\|_* \\
&\leq \int_0^1 \ell(\|\nabla f(\mathbf{x}_{1,2}(s))\|_*) \cdot \|\mathbf{x}_1 - \mathbf{x}_2\| \, \mathrm{d}s \\
&\leq \max_{0 \leq s \leq 1} \ell(\|\nabla f(\mathbf{x}_{1,2}(s))\|_*) \cdot \|\mathbf{x}_1 - \mathbf{x}_2\| \\
&\leq \ell(\|\nabla f(\mathbf{x})\|_* + G) \cdot \|\mathbf{x}_1 - \mathbf{x}_2\|,
\end{aligned} \tag{48}$$

where the first inequality uses Definition 2.1 and the last inequality uses Lemma D.1 and the non-decreasing property of $\ell$. $\square$

We present the following effective or local smoothness property derived under generalized smoothness. Lemma 2.7 is extremely useful as it closes the gap between generalized smoothness and classical smoothness.

*Proof of Lemma 2.7.* To prove this lemma, we adapt the existing result for $f \in \mathcal{F}_{\ell,r}(\|\cdot\|_2)$ (Li et al., 2023a, Lemma 3.3). Specifically, we establish three analogous statements for $f \in \mathcal{F}_{\ell,r}(\|\cdot\|)$ and then leverage the equivalence between generalized smooth function classes to conclude that these statements also hold for $f \in \mathcal{F}_\ell(\|\cdot\|)$. By Definition 2.4, we have $\ell(\|\nabla f(\mathbf{x})\|_*) \leq \ell(G)$ and $r(G) \leq r(\|\nabla f(\mathbf{x})\|_*)$. Hence, it follows that (i) $\mathcal{B}(\mathbf{x}, r(G)) \subseteq \mathcal{B}(\mathbf{x}, r(\|\nabla f(\mathbf{x})\|_*)) \subseteq \mathcal{X}$; and (ii) $\forall \mathbf{x}_1, \mathbf{x}_2 \in \mathcal{B}(\mathbf{x}, r(G))$:

$$\|\nabla f(\mathbf{x}_1) - \nabla f(\mathbf{x}_2)\|_* \leq \ell(\|\nabla f(\mathbf{x})\|_*) \|\mathbf{x}_1 - \mathbf{x}_2\| \leq \ell(G) \|\mathbf{x}_1 - \mathbf{x}_2\|. \tag{49}$$

Statements (i) and (ii) are similar to statements 1 and 2 in the lemma. Now we proceed to the quadratic upper bound. Define $\mathbf{x}(s) = s\mathbf{x}_1 + (1 - s)\mathbf{x}_2$ for $0 \leq s \leq 1$. It's clear that $\mathbf{x}(s), \mathbf{x}_2 \in \mathcal{B}(\mathbf{x}, r(G))$, so according to (49), we have

$$\|\nabla f(\mathbf{x}(s)) - \nabla f(\mathbf{x}_2)\|_* \leq \ell(G) \|\mathbf{x}_1 - \mathbf{x}_2\|. \tag{50}$$

---

[8]If $\mathcal{X} = \mathcal{E}$, this step (together with the analysis in the sequel) is no longer needed.

Then we can derive a local quadratic upper bound (denoted by (iii)) as follows

$$
\begin{aligned}
f(\mathbf{x}_1) - f(\mathbf{x}_2) &= \int_0^1 \langle \nabla f(\mathbf{x}(s)), \mathbf{x}_1 - \mathbf{x}_2 \rangle \mathrm{d}s \\
&= \langle \nabla f(\mathbf{x}_2), \mathbf{x}_1 - \mathbf{x}_2 \rangle + \int_0^1 \langle \nabla f(\mathbf{x}(s)) - \nabla f(\mathbf{x}_2), \mathbf{x}_1 - \mathbf{x}_2 \rangle \mathrm{d}s \\
&\leq \langle \nabla f(\mathbf{x}_2), \mathbf{x}_1 - \mathbf{x}_2 \rangle + \int_0^1 \|\nabla f(\mathbf{x}(s)) - \nabla f(\mathbf{x}_2)\|_* \|\mathbf{x}_1 - \mathbf{x}_2\| \, \mathrm{d}s \\
&\leq \langle \nabla f(\mathbf{x}_2), \mathbf{x}_1 - \mathbf{x}_2 \rangle + \frac{\ell(G)}{2} \|\mathbf{x}_1 - \mathbf{x}_2\|^2 .
\end{aligned}
\tag{51}
$$

At last, we apply the equivalence of generalized smooth function classes to show that the same derivations also hold for $f \in \mathcal{F}_\ell(\|\cdot\|)$. In this case, it suffices to set $L = \ell(2G)$ and $r(G) = G/L$ according to Proposition 2.6. Then the statements (i), (ii), and (iii) can be easily transformed into the ones in Lemma 2.7, which concludes the proof. $\qquad\square$

The following lemma is introduced out of convenience, for the sake of frequent usages in the next few sections.

**Lemma D.2.** *Under Assumption 3.3, let $\|\nabla f(\mathbf{x})\|_* \leq G \in \mathbb{R}_+$ for any given $\mathbf{x} \in \mathcal{X}$. If $0 < \eta \leq \gamma/\ell(2G)$ for some $\gamma \in (0,1)$, then for any $\mathbf{x}_1, \mathbf{x}_2 \in \mathcal{B}(\mathbf{x}, G/\ell(2G))$ and any $\mathbf{x}_3, \mathbf{x}_4 \in \mathcal{X}$,*

$$
\eta \langle \nabla f(\mathbf{x}_1) - \nabla f(\mathbf{x}_2), \mathbf{x}_3 - \mathbf{x}_4 \rangle \leq \frac{\gamma^2}{2\beta} \|\mathbf{x}_1 - \mathbf{x}_2\|^2 + \frac{\beta}{2} \|\mathbf{x}_3 - \mathbf{x}_4\|^2 ,
\tag{52}
$$

*holds for any $\beta \in \mathbb{R}_{++}$.*

*Proof.* Since $\|\nabla f(\mathbf{x})\|_* \leq G$ and $\mathbf{x}_1, \mathbf{x}_2 \in \mathcal{B}(\mathbf{x}, G/\ell(2G))$, by Lemma 2.7, we have

$$
\|\nabla f(\mathbf{x}_1) - \nabla f(\mathbf{x}_2)\|_* \leq \ell(2G) \|\mathbf{x}_1 - \mathbf{x}_2\| .
\tag{53}
$$

According to Hölder's Inequality, we have

$$
\forall \beta \in \mathbb{R}_{++} : \eta \langle \nabla f(\mathbf{x}_1) - \nabla f(\mathbf{x}_2), \mathbf{x}_3 - \mathbf{x}_4 \rangle \leq \frac{\eta^2}{2\beta} \|\nabla f(\mathbf{x}_1) - \nabla f(\mathbf{x}_2)\|_*^2 + \frac{\beta}{2} \|\mathbf{x}_3 - \mathbf{x}_4\|^2 .
\tag{54}
$$

Combining these two inequalities with the condition that $0 < \eta \leq \gamma/\ell(2G)$ completes the proof. $\qquad\square$

Lastly, we provide the omitted proof of Lemma 3.4 in Section 3.2. Lemma 3.4 is also a generalized version of the reversed Polyak-Łojasiewicz Inequality in Li et al. (2023a, Lemma 3.5). Since their derivations rely on the inherent properties of the Euclidean norm, we need to take a different approach to prove this lemma.

*Proof of Lemma 3.4.* For any $\mathbf{x} \in \mathcal{X}$ and any $\mathbf{h}$ satisfying $\|\mathbf{h}\| \leq \|\nabla f(\mathbf{x})\|_* / \ell(2 \|\nabla f(\mathbf{x})\|_*)$, by Lemma 2.7, the following relation holds:

$$
\langle \nabla f(\mathbf{x}), \mathbf{h} \rangle - \frac{\ell(2 \|\nabla f(\mathbf{x})\|_*)}{2} \|\mathbf{h}\|^2 \leq f(\mathbf{x}) - f(\mathbf{x} - \mathbf{h}) \leq f(\mathbf{x}) - f^* .
\tag{55}
$$

In the unconstrained setting, we can leverage the conjugate of the square norm, i.e., $\sup_{\mathbf{h}} \langle \mathbf{x}, \mathbf{h} \rangle - \|\mathbf{h}\|^2 / 2 = \|\mathbf{x}\|_*^2 / 2$ (Orabona, 2019, Example 5.11) to cope with LHS of (55). Now that we are dealing with the vector lying within a small ball centered at $\mathbf{x}$, we should apply the clipping technique to lower-bound the inner product. Let $\mathbf{u}$ be any unit vector with $\langle \nabla f(\mathbf{x}), \mathbf{u} \rangle = \|\nabla f(\mathbf{x})\|_*$. Then we scale $\mathbf{u}$ to fit the radius of the ball $\mathcal{B}(\mathbf{x}, \|\nabla f(\mathbf{x})\|_* / \ell(2 \|\nabla f(\mathbf{x})\|_*))$, i.e., set $\mathbf{v} = \mathbf{u} \cdot \|\nabla f(\mathbf{x})\|_* / \ell(2 \|\nabla f(\mathbf{x})\|_*)$ so that $\|\mathbf{v}\| = \|\nabla f(\mathbf{x})\|_* / \ell(2 \|\nabla f(\mathbf{x})\|_*)$. Since (55) holds for any $\mathbf{h}$ satisfying $\|\mathbf{h}\| \leq \|\nabla f(\mathbf{x})\|_* / \ell(2 \|\nabla f(\mathbf{x})\|_*)$, we take $\mathbf{h} = \mathbf{v}$ and obtain

$$
\begin{aligned}
&\frac{\|\nabla f(\mathbf{x})\|_*}{\ell(2 \|\nabla f(\mathbf{x})\|_*)} \langle \nabla f(\mathbf{x}), \mathbf{u} \rangle - \frac{\ell(2 \|\nabla f(\mathbf{x})\|_*)}{2} \left[ \frac{\|\nabla f(\mathbf{x})\|_*}{\ell(2 \|\nabla f(\mathbf{x})\|_*)} \right]^2 \\
&= \frac{\|\nabla f(\mathbf{x})\|_*^2}{2\ell(2 \|\nabla f(\mathbf{x})\|_*)} \leq f(\mathbf{x}) - f^* .
\end{aligned}
\tag{56}
$$

$\qquad\square$

The following lemma goes one step further to give a quantitative characterization of the relationship between the gradient's dual norm and the suboptimality gap.

**Lemma D.3.** *Under Assumption 3.3, for a given $\mathbf{x} \in \mathcal{X}$ satisfying $f(\mathbf{x}) - f^* \leq F$, we have $\|\nabla f(\mathbf{x})\|_* \leq G < \infty$ and $G^2 = 2\ell(2G)F$, where $G := \sup \{\alpha \in \mathbb{R}_+ | \alpha^2 \leq 2F\ell(2\alpha)\}$.*

*Proof.* Based on Lemma 3.4, the proof is analogous to that of Li et al. (2023a, Corollary 3.6), which involves merely an $f \in \mathcal{F}_\ell(\|\cdot\|_2)$. Since we have a sub-quadratic $\ell$, we clearly have $\lim_{\alpha \to \infty} \frac{\alpha^2}{2\ell(2\alpha)} = \infty$. So for $F \in \mathbb{R}_+, \exists \alpha_F \in \mathbb{R}_{++}$, such that $\forall \alpha > \alpha_F, \frac{\alpha^2}{2\ell(2\alpha)} > F$. Therefore, if $\frac{\alpha^2}{2\ell(2\alpha)} \leq F$ holds for some $\alpha$, then it must be that $\alpha \leq \alpha_F$. Hence, our construction of $G$ implies that $G \leq \alpha_F < \infty$. We conclude that $\|\nabla f(\mathbf{x})\|_* \leq G$ by Lemma 3.4. Lastly, $G^2 = 2\ell(2G)F$ is directly implied by the compactness of the set $\{\alpha \in \mathbb{R}_+ | \alpha^2 \leq 2F\ell(2\alpha)\}$. $\qquad\square$

# E. Analysis for Mirror Descent

We present the following standard lemma for mirror descent, which can be found in Bubeck et al. (2015) and Lan (2020).

**Lemma E.1.** *For any $\mathbf{g} \in \mathcal{E}^*, \mathbf{x}_0 \in \mathcal{X}$, define $\mathbf{x}^+ = \mathcal{P}_{\mathbf{x}_0}(\mathbf{g})$, then we have $\|\mathbf{x}^+ - \mathbf{x}_0\| \leq \|\mathbf{g}\|_*$.*

*Proof.* Using the first-order optimality condition for mirror descent as well as the three-point equation for Bregman functions on the update (Nemirovski, 2004), we have

$$\langle \mathbf{g}, \mathbf{x}^+ - \mathbf{x} \rangle \leq B(\mathbf{x}, \mathbf{x}_0) - B(\mathbf{x}, \mathbf{x}^+) - B(\mathbf{x}^+, \mathbf{x}_0), \quad \forall \mathbf{x} \in \mathcal{X}. \tag{57}$$

Take $\mathbf{x} = \mathbf{x}_0$ and rearrange it,

$$\langle \mathbf{g}, \mathbf{x}_0 - \mathbf{x}^+ \rangle \geq B(\mathbf{x}_0, \mathbf{x}^+) + B(\mathbf{x}^+, \mathbf{x}_0). \tag{58}$$

Using Cauchy-Schwarz Inequality and the property of Bregman divergence, we obtain

$$\begin{aligned}
\|\mathbf{g}\|_* \|\mathbf{x}_0 - \mathbf{x}^+\| &\geq \langle \mathbf{g}, \mathbf{x}_0 - \mathbf{x}^+ \rangle \\
&\geq B(\mathbf{x}_0, \mathbf{x}^+) + B(\mathbf{x}^+, \mathbf{x}_0) \geq \frac{1}{2} \|\mathbf{x}_0 - \mathbf{x}^+\|^2 + \frac{1}{2} \|\mathbf{x}^+ - \mathbf{x}_0\|^2
\end{aligned} \tag{59}$$

If $\|\mathbf{x}_0 - \mathbf{x}^+\| = 0$, Lemma E.1 is naturally true. Otherwise, dividing both sides of the inequality by $\|\mathbf{x}_0 - \mathbf{x}^+\|$ yields the result. $\qquad\square$

**Lemma E.2.** *Under Assumptions 2.5 and 3.3, for any $\mathbf{x}_0 \in \mathcal{X}$ satisfying $f(\mathbf{x}_0) - f^* \leq F$, define $G := \sup \{\alpha \in \mathbb{R}_+ | \alpha^2 \leq 2F\ell(2\alpha)\}$ and $\mathbf{x}^+ = \mathcal{P}_{\mathbf{x}_0}(\eta \nabla f(\mathbf{x}_0))$. If $0 < \eta \leq 1/\ell(2G)$, we have $f(\mathbf{x}^+) \leq f(\mathbf{x}_0)$ and $\|\nabla f(\mathbf{x}^+)\|_* \leq G < \infty$.*

*Proof.* Choose $\mathbf{g} = \eta \nabla f(\mathbf{x}_0)$ and $\mathbf{x} = \mathbf{x}_0$ in (57), we obtain

$$\langle \eta \nabla f(\mathbf{x}_0), \mathbf{x}^+ - \mathbf{x}_0 \rangle \leq -B(\mathbf{x}_0, \mathbf{x}^+) - B(\mathbf{x}^+, \mathbf{x}_0). \tag{60}$$

Under Assumption 2.5, we know that $f^* > -\infty$, which gives $f(\mathbf{x}_0) - f^* < \infty$. By Lemma D.3, we immediately deduce that $\|\nabla f(\mathbf{x}_0)\|_* \leq G < \infty$. Then we invoke Lemma E.1 to control the distance between two iterates:

$$\|\mathbf{x}^+ - \mathbf{x}_0\| \leq \eta \|\nabla f(\mathbf{x}_0)\|_* \leq \frac{\|\nabla f(\mathbf{x}_0)\|_*}{\ell(2G)} \leq \frac{G}{\ell(2G)}, \tag{61}$$

where we use $\eta \leq 1/\ell(2G)$. The conditions of Lemma 2.7 are now satisfied, so we use the effective smoothness property to obtain

$$f(\mathbf{x}^+) \leq f(\mathbf{x}_0) + \langle \nabla f(\mathbf{x}_0), \mathbf{x}^+ - \mathbf{x}_0 \rangle + \frac{\ell(2G)}{2} \|\mathbf{x}^+ - \mathbf{x}_0\|^2. \tag{62}$$

Multiplying by $\eta$ and adding this to (60), we have

$$\begin{aligned}
\eta f(\mathbf{x}^+) &\leq \eta f(\mathbf{x}_0) + \frac{\eta \ell(2G)}{2} \|\mathbf{x}^+ - \mathbf{x}_0\|^2 - B(\mathbf{x}_0, \mathbf{x}^+) - B(\mathbf{x}^+, \mathbf{x}_0) \\
&\leq \eta f(\mathbf{x}_0) + \frac{\eta \ell(2G) - 1}{2} \|\mathbf{x}^+ - \mathbf{x}_0\|^2 - B(\mathbf{x}_0, \mathbf{x}^+) \\
&\leq \eta f(\mathbf{x}_0),
\end{aligned} \tag{63}$$

where the second inequality uses $B(\mathbf{x}^+, \mathbf{x}_0) \geq \frac{1}{2} \|\mathbf{x}^+ - \mathbf{x}_0\|^2$ and the last inequality uses $\eta \leq 1/\ell(2G)$ and the non-negativity of Bregman divergence. Finally, we obtain the following descent property:

$$f(\mathbf{x}^+) - f^* \leq f(\mathbf{x}_0) - f^*. \tag{64}$$

Invoking Lemma D.3 again finishes the proof. $\qquad\square$

*Proof of Theorem 3.5.* According to the descent property in (64), we have $f(\mathbf{x}_{t+1}) - f^* \leq f(\mathbf{x}_t) - f^* \leq \cdots \leq f(\mathbf{x}_0) - f^*$. By Lemma E.2, we deduce that for $\|\nabla f(\mathbf{x}_t)\|_* \leq G < \infty$ holds for all $t \in \mathbb{N}$, where $G := \sup\left\{\alpha \in \mathbb{R}_+ | \alpha^2 \leq 2(f(\mathbf{x}_0) - f^*)\ell(2\alpha)\right\}$. After establishing such an upper bound for the dual norm of the gradients along the trajectory, we can cope with generalized smooth functions in a similar way as the global smooth functions. Similar to (57), we consider the update of mirror descent in (8):

$$\forall \mathbf{x} \in \mathcal{X}, \ \langle \eta \nabla f(\mathbf{x}_t), \mathbf{x}_{t+1} - \mathbf{x} \rangle \leq B(\mathbf{x}, \mathbf{x}_t) - B(\mathbf{x}, \mathbf{x}_{t+1}) - B(\mathbf{x}_{t+1}, \mathbf{x}_t). \tag{65}$$

Note that the local Lipschitz constant $\ell(2 \|\nabla f(\mathbf{x}_t)\|_*)$ is bounded by $L = \ell(2G)$. By Lemma E.1 and $\eta \leq \frac{1}{L}$, we have

$$\|\mathbf{x}_{t+1} - \mathbf{x}_t\| \leq \eta \|\nabla f(\mathbf{x}_t)\|_* \leq \frac{G}{L}. \tag{66}$$

Therefore, the distance between $\mathbf{x}_{t+1}$ and $\mathbf{x}_t$ is small enough to satisfy the conditions in Lemma 2.7. Using the effective smoothness characterization, we obtain

$$f(\mathbf{x}_{t+1}) \leq f(\mathbf{x}_t) + \langle \nabla f(\mathbf{x}_t), \mathbf{x}_{t+1} - \mathbf{x}_t \rangle + \frac{L}{2} \|\mathbf{x}_{t+1} - \mathbf{x}_t\|^2. \tag{67}$$

Next, we divide the inner product term $\langle \eta \nabla f(\mathbf{x}_t), \mathbf{x}_{t+1} - \mathbf{x} \rangle$ into two parts, and bound them separately:

$$\begin{aligned}
\langle \nabla f(\mathbf{x}_t), \mathbf{x}_{t+1} - \mathbf{x} \rangle &= \langle \nabla f(\mathbf{x}_t), \mathbf{x}_t - \mathbf{x} \rangle + \langle \nabla f(\mathbf{x}_t), \mathbf{x}_{t+1} - \mathbf{x}_t \rangle \\
&\geq f(\mathbf{x}_t) - f(\mathbf{x}) + f(\mathbf{x}_{t+1}) - f(\mathbf{x}_t) - \frac{L}{2} \|\mathbf{x}_{t+1} - \mathbf{x}_t\|^2 \\
&= f(\mathbf{x}_{t+1}) - f(\mathbf{x}) - \frac{L}{2} \|\mathbf{x}_{t+1} - \mathbf{x}_t\|^2,
\end{aligned} \tag{68}$$

where the inequality uses the convexity of $f$ and (67). Combining (65) with (68), we derive

$$\eta \left[f(\mathbf{x}_{t+1}) - f(\mathbf{x})\right] - \frac{\eta L}{2} \|\mathbf{x}_{t+1} - \mathbf{x}_t\|^2 \leq B(\mathbf{x}, \mathbf{x}_t) - B(\mathbf{x}, \mathbf{x}_{t+1}) - B(\mathbf{x}_{t+1}, \mathbf{x}_t) \tag{69}$$

After rearrangement and due to the fact that $B(\mathbf{x}_{t+1}, \mathbf{x}_t) \geq \frac{1}{2} \|\mathbf{x}_{t+1} - \mathbf{x}_t\|^2$, we have

$$\begin{aligned}
\eta \left[f(\mathbf{x}_{t+1}) - f(\mathbf{x})\right] &\leq B(\mathbf{x}, \mathbf{x}_t) - B(\mathbf{x}, \mathbf{x}_{t+1}) + \frac{\eta L - 1}{2} \|\mathbf{x}_{t+1} - \mathbf{x}_t\|^2 \\
&\leq B(\mathbf{x}, \mathbf{x}_t) - B(\mathbf{x}, \mathbf{x}_{t+1})
\end{aligned} \tag{70}$$

Telescoping it and using Jensen's Inequality,

$$\begin{aligned}
f(\overline{\mathbf{x}}_T) - f(\mathbf{x}) &\leq \frac{\sum_{t=0}^{T-1} \eta \left[f(\mathbf{x}_{t+1}) - f(\mathbf{x})\right]}{\eta T} \\
&\leq \frac{\sum_{t=0}^{T-1} B(\mathbf{x}, \mathbf{x}_t) - B(\mathbf{x}, \mathbf{x}_{t+1})}{\eta T} \leq \frac{B(\mathbf{x}, \mathbf{x}_0)}{\eta T},
\end{aligned} \tag{71}$$

Taking $\mathbf{x} = \mathbf{x}_*$ gives the desired convergence for the average-iterate. As for the last iterate, recall the descent property in (64), we have $\sum_{t=0}^{T-1} f(\mathbf{x}_{t+1}) - f(\mathbf{x}) \geq T[f(\mathbf{x}_T) - f(\mathbf{x})]$. Thus, the convergence rate is exactly the same as that of $\overline{\mathbf{x}}_T$. $\qquad\square$

# F. Analysis for Accelerated Mirror Descent

The lemmas for this section are stated under the conditions in Theorem 3.7. For simplicity, we do not specify this point below.

**Lemma F.1.** *Define* $e_t := \|\mathbf{y}_t - \mathbf{x}_{t-1}\|$ *for any* $t \in \mathbb{N}_+$, *then we have* $e_1 = e_2 = 0$. *If*

$$e_{t-1} \leq G/L, \quad \|\nabla f(\mathbf{x}_{t-2})\|_* \leq G \tag{72}$$

*hold for any* $t \geq 2$, *we further have*

$$e_t = \frac{2}{t+1} \|\mathbf{x}_{t-1} - \mathbf{z}_{t-1}\| \leq \frac{(t-2)(\eta(t-1)+t)}{t(t+1)} e_{t-1} + \frac{(t-2)(t-1)\eta G}{t(t+1)L}, \quad \forall t \geq 2. \tag{73}$$

*Proof.* By (10), the following holds for any $t \geq 3$:

$$\mathbf{y}_t - \mathbf{x}_{t-1} = \alpha_t(\mathbf{z}_{t-1} - \mathbf{x}_{t-1}) = \alpha_t(1 - \alpha_{t-1})(\mathbf{z}_{t-1} - \mathbf{x}_{t-2}). \tag{74}$$

We consider the last term above:

$$\|\mathbf{z}_{t-1} - \mathbf{x}_{t-2}\| \leq \|\mathbf{z}_{t-1} - \mathbf{z}_{t-2}\| + \|\mathbf{z}_{t-2} - \mathbf{x}_{t-2}\| \leq \eta_{t-1} \|\nabla f(\mathbf{y}_{t-1})\|_* + \frac{e_{t-1}}{\alpha_{t-1}}, \tag{75}$$

where the last inequality is due to (10) and Lemma E.1. Then we use a similar idea as in Lemma D.1 to bound $\|\nabla f(\mathbf{y}_{t-1})\|_*$:

$$\|\nabla f(\mathbf{y}_{t-1})\|_* \leq \|\nabla f(\mathbf{y}_{t-1}) - \nabla f(\mathbf{x}_{t-2})\|_* + \|\nabla f(\mathbf{x}_{t-2})\|_* \leq Le_{t-1} + G, \tag{76}$$

where we use (72) and Lemma 2.7. Combine the previous results, we obtain

$$
\begin{aligned}
e_t &\leq \alpha_t(1 - \alpha_{t-1}) \left[ \eta_{t-1}(Le_{t-1} + G) + \frac{e_{t-1}}{\alpha_{t-1}} \right] \\
&\leq \frac{2}{t+1} \cdot \frac{t-2}{t} \left[ \frac{\eta(t-1)G}{2L} + \frac{\eta(t-1)e_{t-1}}{2} + \frac{te_{t-1}}{2} \right] \\
&\leq \frac{(t-2)(t-1)}{t(t+1)} \cdot \frac{\eta G}{L} + \frac{(t-2)(\eta(t-1)+t)}{t(t+1)} \cdot e_{t-1}.
\end{aligned}
\tag{77}
$$

For $e_2$, we also have $e_2 = 2/(2+1) \|\mathbf{x}_1 - \mathbf{z}_1\| = 0$ based on (10). $e_1 = 0$ is a direct consequence of our initialization. $\square$

**Lemma F.2.** *If* $\mathbf{x}_t, \mathbf{y}_t$ *satisfies* $\|\nabla f(\mathbf{y}_t)\|_* \leq 2G$ *and* $\|\mathbf{x}_t - \mathbf{y}_t\| \leq 2G/L$, *then for any* $\mathbf{x} \in \mathcal{X}$:

$$\frac{\eta t(t+1)}{4L} [f(\mathbf{x}_t) - f(\mathbf{x})] \leq \frac{\eta t(t-1)}{4L} [f(\mathbf{x}_{t-1}) - f(\mathbf{x})] + B(\mathbf{x}, \mathbf{z}_{t-1}) - B(\mathbf{x}, \mathbf{z}_t), \forall t \in \mathbb{N}_+. \tag{78}$$

*Proof.* In view of Lemma 2.7 as well as the given condition, we have

$$
\begin{aligned}
f(\mathbf{x}_t) &\leq f(\mathbf{y}_t) + \langle \nabla f(\mathbf{y}_t), \mathbf{x}_t - \mathbf{y}_t \rangle + \frac{L}{2} \|\mathbf{x}_t - \mathbf{y}_t\|^2 \\
&= (1 - \alpha_t) [f(\mathbf{y}_t) + \langle \nabla f(\mathbf{y}_t), \mathbf{x}_{t-1} - \mathbf{y}_t \rangle] \\
&\quad + \alpha_t [f(\mathbf{y}_t) + \langle \nabla f(\mathbf{y}_t), \mathbf{z}_t - \mathbf{y}_t \rangle] + \frac{\alpha_t^2 L}{2} \|\mathbf{z}_t - \mathbf{z}_{t-1}\|^2 \\
&\leq (1 - \alpha_t)f(\mathbf{x}_{t-1}) + \alpha_t \left[ f(\mathbf{y}_t) + \langle \nabla f(\mathbf{y}_t), \mathbf{z}_t - \mathbf{y}_t \rangle + \frac{1}{\eta_t} B(\mathbf{z}_t, \mathbf{z}_{t-1}) \right],
\end{aligned}
\tag{79}
$$

where the first equality is due to $\mathbf{x}_t = (1 - \alpha_t)\mathbf{x}_{t-1} + \alpha_t\mathbf{z}_t$ and $\mathbf{x}_t - \mathbf{y}_t = \alpha_t(\mathbf{z}_t - \mathbf{z}_{t-1})$; the first inequality uses the convexity of $f$, the property of Bregman divergence and the fact $\alpha_t L \leq \eta_t$ (since $\eta \leq 1$). Similar to (57), the following holds for any $\mathbf{x} \in \mathcal{X}$:

$$\langle \nabla f(\mathbf{y}_t), \mathbf{z}_t - \mathbf{x} \rangle \leq \frac{B(\mathbf{x}, \mathbf{z}_{t-1}) - B(\mathbf{x}, \mathbf{z}_t) - B(\mathbf{z}_t, \mathbf{z}_{t-1})}{\eta_t}. \tag{80}$$

Plugging the above relation into the previous derivations, we obtain

$$f(\mathbf{x}_t) \leq (1 - \alpha_t) f(\mathbf{x}_{t-1}) + \alpha_t \left[ f(\mathbf{y}_t) + \langle \nabla f(\mathbf{y}_t), \mathbf{x} - \mathbf{y}_t \rangle + \frac{B(\mathbf{x}, \mathbf{z}_{t-1}) - B(\mathbf{x}, \mathbf{z}_t)}{\eta_t} \right]$$

$$\leq (1 - \alpha_t) f(\mathbf{x}_{t-1}) + \alpha_t f(\mathbf{x}) + \frac{\alpha_t}{\eta_t} \left[ B(\mathbf{x}, \mathbf{z}_{t-1}) - B(\mathbf{x}, \mathbf{z}_t) \right], \tag{81}$$

where the second step uses the convexity of $f$. A simple rearrangement yields

$$\frac{\eta_t}{\alpha_t} \left[ f(\mathbf{x}_t) - f(\mathbf{x}) \right] \leq \frac{(1 - \alpha_t)\eta_t}{\alpha_t} \left[ f(\mathbf{x}_{t-1}) - f(\mathbf{x}) \right] + B(\mathbf{x}, \mathbf{z}_{t-1}) - B(\mathbf{x}, \mathbf{z}_t). \tag{82}$$

Specifying the value of $\alpha_t$ and $\eta_t$ completes the proof. $\qquad\square$

**Lemma F.3.** *The trajectory generated by* (10) *satisfies the following relations for all* $t \in \mathbb{N}_+$:

1. *(bounded suboptimality gap)* $\quad f(\mathbf{x}_t) - f^* \leq f(\mathbf{x}_0) - f^*$.

2. *(bounded gradients)* $\quad \|\nabla f(\mathbf{x}_t)\|_* \leq G < \infty, \|\nabla f(\mathbf{y}_t)\|_* \leq 2G < \infty$.

3. *(bounded sequence)* $\quad e_t \leq G/L$.

4. *(bounded trajectory)* $\quad \mathbf{x}_t, \mathbf{z}_t \in \mathcal{B}\left(\mathbf{x}_*, \sqrt{2B(\mathbf{x}_*, \mathbf{x}_0)}\right)$.

*Proof.* First, we prove the bounded trajectory property. Define the Lyapunov function $E_t$ as

$$E_t := \frac{\eta t(t+1)}{4L} \left[ f(\mathbf{x}_t) - f^* \right] + B(\mathbf{x}_*, \mathbf{z}_t) \geq B(\mathbf{x}_*, \mathbf{z}_t) \geq \frac{1}{2} \|\mathbf{x}_* - \mathbf{z}_t\|^2. \tag{83}$$

Lemma F.2 states that $E_t \leq E_{t-1} \leq \cdots E_1 \leq E_0 = B(\mathbf{x}, \mathbf{x}_0)$. Combining with (83), we deduce that $\|\mathbf{z}_t - \mathbf{x}_*\| \leq \sqrt{2B(\mathbf{x}_*, \mathbf{x}_0)}$, implying $\mathbf{z}_t \in \mathcal{B}\left(\mathbf{x}_*, \sqrt{2B(\mathbf{x}_*, \mathbf{x}_0)}\right)$. For $\mathbf{x}_t$, recall the update rule of $\mathbf{x}_t = (1 - \alpha_t)\mathbf{x}_{t-1} + \alpha_t \mathbf{z}_t$ and that $\mathbf{x}_0 = \mathbf{z}_0$. We deduce that $\mathbf{x}_t$ is a convex combination of $\{\mathbf{z}_s\}_{s=0}^t$. By convexity of $\mathcal{B}\left(\mathbf{x}_*, \sqrt{2B(\mathbf{x}_*, \mathbf{x}_0)}\right)$, it holds that $\mathbf{x}_t \in \mathcal{B}\left(\mathbf{x}_*, \sqrt{2B(\mathbf{x}_*, \mathbf{x}_0)}\right)$. For conditions 1 to 3, we proceed by induction, as specified below.

**Part 1. Base Case**
For $t = 1$, we clearly have $\mathbf{y}_1 = \mathbf{z}_0 = \mathbf{x}_0$ and thus $\|\nabla f(\mathbf{y}_1)\|_* \leq G$. Also, we have $\|\mathbf{x}_1 - \mathbf{y}_1\| = \|\mathbf{z}_1 - \mathbf{z}_0\| \leq \eta_1 \|\nabla f(\mathbf{z}_0)\|_* \leq G/(2L)$. In this scenario, the conditions of Lemma F.2 are satisfied, we immediately obtain

$$\frac{\eta}{2L} \left[ f(\mathbf{x}_1) - f(\mathbf{x}_0) \right] \leq -B(\mathbf{z}_0, \mathbf{z}_1) \leq 0, \tag{84}$$

by choosing $\mathbf{x} = \mathbf{x}_0 = \mathbf{z}_0$. Then we conclude that $f(\mathbf{x}_1) - f^* \leq f(\mathbf{x}_0) - f^*$. By Lemma D.3, we have $\|\nabla f(\mathbf{x}_t)\|_* \leq G < \infty$. By definition, we have $e_1 = 0$, which completes the proof for this part.

**Part 2. Induction Step**
Suppose the statements hold for all $t \leq s - 1$ where $s \geq 2$. Our goal is to show that they also hold for $t = s$. The main idea is to show the conditions of Lemma F.2 hold via Lemma F.1. Then we establish the descent property under Lemma F.2, which is the key to the bounded suboptimality gap as well as the bounded gradients. First, we consider the third statement by splitting the timeline.

**Case (i).** $s \geq \tau \geq 3$

Recall the definition of $\tau$ in Theorem 3.7:

$$\tau = \left\lceil \frac{4\sqrt{2B(\mathbf{x}_*, \mathbf{x}_0)}L}{G} \right\rceil - 1 = \left\lceil \frac{8\sqrt{2B(\mathbf{x}_*, \mathbf{x}_0)}}{\frac{2G}{L}} \right\rceil - 1 \geq \left\lceil \frac{8\sqrt{2B(\mathbf{x}_*, \mathbf{x}_0)}}{2\sqrt{2B(\mathbf{x}_*, \mathbf{x}_0)}} \right\rceil - 1 \geq 3. \tag{85}$$

where we use $2G/L \le 2\sqrt{2B(\mathbf{x}_*, \mathbf{x}_0)}$ (Yu et al., 2024, Proposition A.6). Then we have

$$
\begin{aligned}
e_s =& \frac{2}{s+1} \|\mathbf{x}_{t-1} - \mathbf{z}_{t-1}\| \le \frac{2}{\tau+1} \left( \|\mathbf{x}_{t-1} - \mathbf{x}_*\| + \|\mathbf{z}_{t-1} - \mathbf{x}_*\| \right) \\
\le& \frac{2}{\tau+1} \cdot 2\sqrt{2B(\mathbf{x}_*, \mathbf{x}_0)} \le \frac{4\sqrt{2B(\mathbf{x}_*, \mathbf{x}_0)}}{\frac{4\sqrt{2B(\mathbf{x}_*, \mathbf{x}_0)}L}{G}} \le \frac{G}{L}.
\end{aligned}
\tag{86}
$$

**Case (ii).** $2 \le s \le \tau - 1$

From the induction basis, we see that the conditions of Lemma F.1 are satisfied. Then we have

$$
\begin{aligned}
e_s \le& \frac{(s-2)(\eta(s-1)+s)}{s(s+1)} e_{s-1} + \frac{(s-2)(s-1)\eta G}{s(s+1)L} \\
\le& \underbrace{\frac{(\tau-3)(\eta(\tau-2)+\tau-1)}{\tau(\tau-1)}}_{q_\tau} \cdot e_{s-1} + \underbrace{\frac{(\tau-3)(\tau-2)\eta}{\tau(\tau-1)}}_{a_\tau} \cdot \frac{G}{L}.
\end{aligned}
\tag{87}
$$

We aim to bound $e_s$ as a contraction mapping. For the ratio parameter, we have

$$
q_\tau = \frac{\tau-3}{\tau} + \frac{(\tau-3)(\tau-2)\eta}{\tau(\tau-1)} \le 1 - \frac{3}{\tau} + \frac{3}{2\tau} < 1,
\tag{88}
$$

where we use the definition of $\eta$. Then we obtain

$$
e_s - \frac{a_\tau}{1-q_\tau} \le q_\tau \left( e_{s-1} - \frac{a_\tau}{1-q_\tau} \right) \le \cdots \le q_\tau^{s-2} \left( e_2 - \frac{a_\tau}{1-q_\tau} \right) \overset{e_2=0}{\le} 0.
\tag{89}
$$

Thus,

$$
e_s \le \frac{a_\tau}{1-q_\tau} \le \frac{\frac{3}{2\tau}}{\frac{3}{\tau} - \frac{3}{2\tau}} \cdot \frac{G}{L} \le \frac{G}{L}.
\tag{90}
$$

Therefore, we have proven the third statement.

Next, we show that the conditions of Lemma F.2 are satisfied.

$$
\|\nabla f(\mathbf{y}_s)\|_* \le \|\nabla f(\mathbf{y}_s) - \nabla f(\mathbf{x}_{s-1})\|_* + \|\nabla f(\mathbf{x}_{s-1})\|_* \le L e_{s-1} + G \le 2G,
\tag{91}
$$

where we use the induction basis $\|\nabla f(\mathbf{x}_{s-1})\|_* \le G, e_s \le G/L$ and Lemma 2.7. We also have

$$
\|\mathbf{x}_t - \mathbf{y}_t\| = \alpha_t \|\mathbf{z}_t - \mathbf{z}_{t-1}\| \le \alpha_t \eta_t \|\nabla f(\mathbf{y}_t)\|_* \le \frac{t\eta G}{(t+1)L} \le \frac{G}{L}.
\tag{92}
$$

Then we invoke Lemma F.2 to get the following

$$
\frac{\eta s(s+1)}{4L} [f(\mathbf{x}_s) - f(\mathbf{x}_0)] \le \frac{\eta s(s-1)}{4L} [f(\mathbf{x}_{s-1}) - f(\mathbf{x}_0)] + B(\mathbf{x}_0, \mathbf{z}_{s-1}) - B(\mathbf{x}_0, \mathbf{z}_s),
\tag{93}
$$

which also holds for all $1 \le t \le s-1$ by induction. Summing from $t = 1$ to $t = s$, we obtain

$$
\frac{\eta s(s+1)}{4L} [f(\mathbf{x}_s) - f(\mathbf{x}_0)] \le -B(\mathbf{x}_0, \mathbf{z}_s) \le 0,
\tag{94}
$$

which yields $f(\mathbf{x}_t) - f^* \le f(\mathbf{x}_0) - f^*$. $\|\nabla f(\mathbf{x}_t)\|_* \le G < \infty$ is immediately derived using Lemma D.3. Finally, the three statements are all proven, which completes the proof. $\square$

*Proof of Theorem 3.7.* By Lemma F.3, we immediately conclude that

$$
\max\{\|\nabla f(\mathbf{y}_t)\|_*, 2\|\nabla f(\mathbf{x}_t)\|_*\} \le 2G < \infty, \quad \forall t \in \mathbb{N}_+.
\tag{95}
$$

Similar to (92), we have $\|\mathbf{x}_t - \mathbf{y}_t\| \leq 2G/L$. Then, we invoke Lemma F.2 to derive the following:

$$\frac{\eta t(t+1)}{4L}\left[f(\mathbf{x}_t) - f(\mathbf{x})\right] \leq \frac{\eta t(t-1)}{4L}\left[f(\mathbf{x}_{t-1}) - f(\mathbf{x})\right] + B(\mathbf{x}, \mathbf{z}_{t-1}) - B(\mathbf{x}, \mathbf{z}_t), \tag{96}$$

which holds for all $t \in \mathbb{N}_+$ and $\mathbf{x} \in \mathcal{X}$. Summing from $1$ to $T$ and choosing $\mathbf{x} = \mathbf{x}_*$, we obtain

$$\frac{\eta T(T+1)}{4L}\left[f(\mathbf{x}_T) - f^*\right] \leq B(\mathbf{x}_*, \mathbf{z}_0) - B(\mathbf{x}_*, \mathbf{z}_T) \leq B(\mathbf{x}_*, \mathbf{x}_0). \tag{97}$$

A simple rearrangement yields the desired convergence result. $\qquad\square$

## G. Analysis for Optimistic Mirror Descent

**Lemma G.1.** *Consider the $t$-th iteration of the optimistic mirror descent algorithm defined in* (12). *The following relations hold:*

$$\begin{aligned}
\eta(f(\mathbf{y}_t) - f(\mathbf{x}_{t-1})) \leq &\eta\langle\nabla f(\mathbf{y}_t) - \nabla f(\mathbf{y}_{t-1}), \mathbf{y}_t - \mathbf{x}_{t-1}\rangle \\
&- \frac{1}{2}\|\mathbf{x}_t - \mathbf{y}_t\|^2 - \frac{1}{2}\|\mathbf{y}_t - \mathbf{x}_{t-1}\|^2 - \frac{1}{2}\|\mathbf{x}_t - \mathbf{x}_{t-1}\|^2,
\end{aligned} \tag{98}$$

$$\begin{aligned}
\eta(f(\mathbf{x}_t) - f(\mathbf{x}_{t-1})) \leq &\eta\langle\nabla f(\mathbf{x}_t) - \nabla f(\mathbf{y}_{t-1}), \mathbf{y}_t - \mathbf{x}_{t-1}\rangle \\
&+ \eta\langle\nabla f(\mathbf{x}_t) - \nabla f(\mathbf{y}_t), \mathbf{x}_t - \mathbf{y}_t\rangle \\
&- \frac{1}{2}\|\mathbf{x}_t - \mathbf{y}_t\|^2 - \frac{1}{2}\|\mathbf{y}_t - \mathbf{x}_{t-1}\|^2 - \frac{1}{2}\|\mathbf{x}_t - \mathbf{x}_{t-1}\|^2.
\end{aligned} \tag{99}$$

*Proof.* By Lemma H.1, we have

$$\forall \mathbf{x} \in \mathcal{X}, \langle\eta\nabla f(\mathbf{y}_{t-1}), \mathbf{y}_t - \mathbf{x}\rangle \leq B(\mathbf{x}, \mathbf{x}_{t-1}) - B(\mathbf{x}, \mathbf{y}_t) - B(\mathbf{y}_t, \mathbf{x}_{t-1}); \tag{100}$$
$$\forall \mathbf{x} \in \mathcal{X}, \langle\eta\nabla f(\mathbf{y}_t), \mathbf{x}_t - \mathbf{x}\rangle \leq B(\mathbf{x}, \mathbf{x}_{t-1}) - B(\mathbf{x}, \mathbf{x}_t) - B(\mathbf{x}_t, \mathbf{x}_{t-1}). \tag{101}$$

Choose $\mathbf{x} = \mathbf{x}_t$ in (100), $\mathbf{x} = \mathbf{x}_{t-1}$ in (101) and then add them together:

$$\begin{aligned}
&\langle\eta\nabla f(\mathbf{y}_{t-1}), \mathbf{y}_t - \mathbf{x}_t\rangle + \langle\eta\nabla f(\mathbf{y}_t), \mathbf{x}_t - \mathbf{x}_{t-1}\rangle \\
&\leq -B(\mathbf{x}_t, \mathbf{y}_t) - B(\mathbf{y}_t, \mathbf{x}_{t-1}) - B(\mathbf{x}_{t-1}, \mathbf{x}_t).
\end{aligned} \tag{102}$$

By convexity of $f$ and the property of Bregman function, we obtain

$$\begin{aligned}
&\eta(f(\mathbf{y}_t) - f(\mathbf{x}_{t-1})) \\
\leq &\langle\eta\nabla f(\mathbf{y}_t), \mathbf{y}_t - \mathbf{x}_{t-1}\rangle \\
= &\eta\langle\nabla f(\mathbf{y}_t) - \nabla f(\mathbf{y}_{t-1}), \mathbf{y}_t - \mathbf{x}_t\rangle + \langle\eta f(\mathbf{y}_{t-1}), \mathbf{y}_t - \mathbf{x}_t\rangle + \langle\eta f(\mathbf{y}_t), \mathbf{x}_t - \mathbf{x}_{t-1}\rangle \\
\leq &\eta\langle\nabla f(\mathbf{y}_t) - \nabla f(\mathbf{y}_{t-1}), \mathbf{y}_t - \mathbf{x}_t\rangle - \frac{1}{2}\|\mathbf{x}_t - \mathbf{y}_t\|^2 - \frac{1}{2}\|\mathbf{y}_t - \mathbf{x}_{t-1}\|^2 - \frac{1}{2}\|\mathbf{x}_t - \mathbf{x}_{t-1}\|^2.
\end{aligned} \tag{103}$$

To prove the second bound, we first notice that

$$\begin{aligned}
\langle\nabla f(\mathbf{x}_t), \mathbf{x}_t - \mathbf{x}_{t-1}\rangle = &\underbrace{\langle\nabla f(\mathbf{y}_{t-1}), \mathbf{y}_t - \mathbf{x}_{t-1}\rangle}_{A_t} + \underbrace{\langle\nabla f(\mathbf{x}_t) - \nabla f(\mathbf{y}_{t-1}), \mathbf{y}_t - \mathbf{x}_{t-1}\rangle}_{B_t} \\
&+ \underbrace{\langle\nabla f(\mathbf{y}_t), \mathbf{x}_t - \mathbf{y}_{t-1}\rangle}_{C_t} + \underbrace{\langle\nabla f(\mathbf{x}_t) - \nabla f(\mathbf{y}_t), \mathbf{x}_t - \mathbf{y}_t\rangle}_{D_t}.
\end{aligned} \tag{104}$$

Similar to the proof of Lemma H.3, we bound the four terms separately. To bound $A_t$ and $C_t$, we choose $\mathbf{x} = \mathbf{x}_{t-1}$ in (100), $\mathbf{x} = \mathbf{y}_t$ in (101). Adding them up, we have

$$\begin{aligned}
\eta(A_t + C_t) = &\langle\eta\nabla f(\mathbf{y}_{t-1}), \mathbf{y}_t - \mathbf{x}_{t-1}\rangle + \langle\eta\nabla f(\mathbf{y}_t), \mathbf{x}_t - \mathbf{y}_t\rangle \\
\leq &-B(\mathbf{x}_{t-1}, \mathbf{y}_t) - B(\mathbf{y}_t, \mathbf{x}_t) - B(\mathbf{x}_t, \mathbf{x}_{t-1}).
\end{aligned} \tag{105}$$

By convexity of $f$ and the property of Bregman function, we obtain

$$
\begin{aligned}
\eta(f(\mathbf{x}_t) - f(\mathbf{x}_{t-1})) \leq & \eta \langle \nabla f(\mathbf{x}_t), \mathbf{x}_t - \mathbf{x}_{t-1} \rangle \\
\leq & \eta(B_t + D_t) - \frac{1}{2} \|\mathbf{x}_{t-1} - \mathbf{y}_t\|^2 - \frac{1}{2} \|\mathbf{y}_t - \mathbf{x}_t\|^2 - \frac{1}{2} \|\mathbf{x}_t - \mathbf{x}_{t-1}\|^2,
\end{aligned}
\tag{106}
$$

which completes the proof by plugging in the definitions of $B_t$ and $D_t$. $\qquad\square$

**Lemma G.2.** *Under Assumptions 2.5 and 3.3, consider the updates in* (12) *and define* $G := \sup\{\alpha \in \mathbb{R}_+ | \alpha^2 \leq 2(f(\mathbf{x}_0) - f^*)\ell(2\alpha)\}$. *If* $0 < \eta \leq \gamma/\ell(2G)$ *for some* $\gamma \in (0, 1/3]$, *the following statements holds for all* $t \in \mathbb{N}_+$:

1. *(bounded suboptimality gap)* $\quad f(\mathbf{y}_t) - f^* \leq f(\mathbf{x}_0) - f^*, f(\mathbf{x}_t) - f^* \leq f(\mathbf{x}_0) - f^*;$

2. *(bounded gradients)* $\quad \|\nabla f(\mathbf{y}_t)\|_* \leq G < \infty, \|\nabla f(\mathbf{x}_t)\|_* \leq G < \infty.$

3. *(tracking $\{\mathbf{y}_t\}_{t\in\mathbb{N}}$ sequence)* $\quad \|\mathbf{y}_t - \mathbf{y}_{t-1}\| \leq \frac{G}{L} \sum_{s=1}^{t} \gamma^s \leq \frac{\gamma G}{(1-\gamma)L} < \frac{G}{L}.$

*Proof.* The proof resembles that of Lemma E.2, which involves induction. For convenience, we denote $L := \ell(2G)$.

**Part 1. Base Case**
Consider the first round of (12). Similar to the proof of Lemma E.2, we argue that $f(\mathbf{y}_0) - f^* = f(\mathbf{x}_0) - f^* < \infty$. By Lemma 3.4, we have $\|\nabla f(\mathbf{y}_0)\|_* = \|\nabla f(\mathbf{y}_0)\|_* \leq G < \infty$. According to Lemma G.1, we have

$$
\begin{aligned}
\eta(f(\mathbf{y}_1) - f(\mathbf{x}_0)) \leq & \eta \langle \nabla f(\mathbf{y}_1) - \nabla f(\mathbf{y}_0), \mathbf{y}_1 - \mathbf{x}_0 \rangle \\
& - \frac{1}{2} \|\mathbf{x}_1 - \mathbf{y}_1\|^2 - \frac{1}{2} \|\mathbf{y}_1 - \mathbf{x}_0\|^2 - \frac{1}{2} \|\mathbf{x}_1 - \mathbf{x}_0\|^2.
\end{aligned}
\tag{107}
$$

It suffices to control the inner product term. By Lemma E.1 as well as our initialization $\mathbf{x}_0 = \mathbf{y}_0$, we have

$$
\|\mathbf{y}_1 - \mathbf{y}_0\| = \|\mathbf{y}_1 - \mathbf{x}_0\| \leq \eta \|\nabla f(\mathbf{x}_0)\|_* \leq \frac{\gamma G}{L} \leq \frac{G}{L}.
\tag{108}
$$

Then we use Lemma D.2 to establish the following upper bound:

$$
\eta \langle \nabla f(\mathbf{y}_1) - \nabla f(\mathbf{y}_0), \mathbf{y}_1 - \mathbf{x}_1 \rangle \leq \frac{\gamma^2}{2} \|\mathbf{y}_1 - \mathbf{y}_0\|^2 + \frac{1}{2} \|\mathbf{y}_1 - \mathbf{x}_1\|^2.
\tag{109}
$$

Noticing $\mathbf{x}_0 = \mathbf{y}_0$ and $\gamma^2 < 1$, we obtain

$$
\eta(f(\mathbf{y}_1) - f(\mathbf{x}_0)) \leq -\frac{1}{2} \|\mathbf{x}_1 - \mathbf{x}_0\|^2 \leq 0.
\tag{110}
$$

Now we clearly have $f(\mathbf{y}_1) - f^* \leq f(\mathbf{x}_0) - f^*$ as $\eta > 0$. By Lemma D.3, we conclude that $\|\nabla f(\mathbf{y}_1)\|_* \leq G$. Next, we prove the other half of the statements. By Lemma G.1, we have

$$
\begin{aligned}
\eta(f(\mathbf{x}_1) - f(\mathbf{x}_0)) \leq & \eta \langle \nabla f(\mathbf{x}_1) - \nabla f(\mathbf{y}_0), \mathbf{y}_1 - \mathbf{x}_0 \rangle + \eta \langle \nabla f(\mathbf{x}_1) - \nabla f(\mathbf{y}_1), \mathbf{x}_1 - \mathbf{y}_1 \rangle \\
& - \frac{1}{2} \|\mathbf{x}_0 - \mathbf{y}_1\|^2 - \frac{1}{2} \|\mathbf{y}_1 - \mathbf{x}_1\|^2 - \frac{1}{2} \|\mathbf{x}_1 - \mathbf{x}_0\|^2.
\end{aligned}
\tag{111}
$$

In the language of Lemma G.1, we shall establish an upper bound for the terms $\eta B_1$ and $\eta D_1$, i.e., the inner product terms in the above inequality. By Lemma E.1, we have

$$
\|\mathbf{x}_1 - \mathbf{y}_0\| = \|\mathbf{x}_1 - \mathbf{x}_0\| \leq \eta \|\nabla f(\mathbf{y}_1)\|_* \leq \frac{\gamma G}{L} \leq \frac{G}{L},
\tag{112}
$$

and by Lemma H.1,

$$
\|\mathbf{x}_1 - \mathbf{y}_1\| \leq \eta \|\nabla f(\mathbf{y}_1) - \nabla f(\mathbf{y}_0)\|_* \leq \eta L \|\mathbf{y}_1 - \mathbf{y}_0\| \leq \frac{\gamma^2 G}{L} \leq \frac{G}{L},
\tag{113}
$$

where the second inequality uses $\|\mathbf{y}_1 - \mathbf{y}_0\| \leq \frac{G}{L}$ and Lemma 2.7. Then we invoke Lemma D.2 to derive the following bounds for $\eta B_1$ (set $\beta = 1$) as well as $\eta D_1$ (set $\beta = \frac{1}{2}$):

$$\eta B_1 = \eta \langle \nabla f(\mathbf{x}_1) - \nabla f(\mathbf{y}_0), \mathbf{y}_1 - \mathbf{x}_0 \rangle \leq \frac{\gamma^2}{2} \|\mathbf{x}_1 - \mathbf{y}_0\|^2 + \frac{1}{2} \|\mathbf{y}_1 - \mathbf{x}_0\|^2 . \tag{114}$$

$$\eta D_1 = \eta \langle \nabla f(\mathbf{x}_1) - \nabla f(\mathbf{y}_1), \mathbf{x}_1 - \mathbf{y}_1 \rangle \leq \frac{1}{2} \|\mathbf{x}_1 - \mathbf{y}_1\|^2 . \tag{115}$$

Combine (111) with the above inequalities, we obtain

$$\begin{aligned}
\eta(f(\mathbf{x}_1) - f(\mathbf{x}_0)) \leq & \frac{\gamma^2}{2} \|\mathbf{x}_1 - \mathbf{y}_0\|^2 + \frac{1}{2} \|\mathbf{y}_1 - \mathbf{x}_0\|^2 + \frac{1}{2} \|\mathbf{x}_1 - \mathbf{y}_1\|^2 \\
& - \frac{1}{2} \|\mathbf{x}_0 - \mathbf{y}_1\|^2 - \frac{1}{2} \|\mathbf{y}_1 - \mathbf{x}_1\|^2 - \frac{1}{2} \|\mathbf{x}_1 - \mathbf{x}_0\|^2 \\
\leq & - \frac{1 - \gamma^2}{2} \|\mathbf{x}_1 - \mathbf{x}_0\|^2 \overset{\gamma^2 < 1}{\leq} 0,
\end{aligned} \tag{116}$$

where we use the initialization setup of $\mathbf{x}_0 = \mathbf{y}_0$. Consequently, we conclude that $f(\mathbf{x}_1) - f^* \leq f(\mathbf{x}_0) - f^*$. $\|\nabla f(\mathbf{y}_1)\|_* \leq G$ is straightforward after a standard application of Lemma D.3.

**Part 2. Induction Step**
Suppose the function values and the gradients are bounded before the $t$-th iteration. Thus we have $\|\nabla f(\mathbf{y}_{t-1})\|_* \leq G$, $\|\nabla f(\mathbf{x}_{t-1})\|_* \leq G$ and $\|\mathbf{y}_{t-1} - \mathbf{y}_{t-2}\| \leq \frac{G}{L} \sum_{s=1}^{t-1} \gamma^s \leq \frac{\gamma G}{(1-\gamma)L}$. We prove **statement 3** first, which tracks the amount of change of the sequence $\{\mathbf{y}_t\}_{t \in \mathbb{N}}$.

$$\begin{aligned}
\|\mathbf{y}_t - \mathbf{y}_{t-1}\| \leq & \|\mathbf{y}_t - \mathbf{x}_{t-1}\| + \|\mathbf{x}_{t-1} - \mathbf{y}_{t-1}\| \\
\leq & \eta \|\nabla f(\mathbf{y}_{t-1})\|_* + \eta \|\nabla f(\mathbf{y}_{t-1}) - \nabla f(\mathbf{y}_{t-2})\|_* \\
\leq & \eta G + \eta L \|\mathbf{y}_{t-1} - \mathbf{y}_{t-2}\| \\
\leq & \frac{\gamma G}{L} + \frac{\gamma G}{L} \sum_{s=1}^{t-1} \gamma^s = \frac{G}{L} \sum_{s=1}^{t} \gamma^s \leq \frac{\gamma G}{(1-\gamma)L},
\end{aligned} \tag{117}$$

where the second inequality uses Lemma H.1 and the third one use Lemma 2.7. Now we examine the rest statements for the $t$-th round of the algorithm. By Lemma G.1, we have

$$\begin{aligned}
& \eta(f(\mathbf{x}_t) - f(\mathbf{x}_{t-1})) \\
\leq & \underbrace{\eta \langle \nabla f(\mathbf{x}_t) - \nabla f(\mathbf{y}_{t-1}), \mathbf{y}_t - \mathbf{x}_{t-1} \rangle}_{B_t} + \underbrace{\eta \langle \nabla f(\mathbf{x}_t) - \nabla f(\mathbf{y}_t), \mathbf{x}_t - \mathbf{y}_t \rangle}_{D_t} \\
& - \frac{1}{2} \|\mathbf{x}_t - \mathbf{y}_t\|^2 - \frac{1}{2} \|\mathbf{y}_t - \mathbf{x}_{t-1}\|^2 - \frac{1}{2} \|\mathbf{x}_t - \mathbf{x}_{t-1}\|^2 .
\end{aligned} \tag{118}$$

To leverage the effective smoothness property, we need to carefully control the distance quantity, namely $\|\mathbf{x}_t - \mathbf{y}_t\|$ and $\|\mathbf{x}_t - \mathbf{y}_{t-1}\|$. Based on the bound of $\|\mathbf{y}_t - \mathbf{y}_{t-1}\|$, by Lemmas 2.7 and H.1 we have

$$\begin{aligned}
\|\mathbf{x}_t - \mathbf{y}_t\| \leq & \eta \|\nabla f(\mathbf{y}_t) - \nabla f(\mathbf{y}_{t-1})\|_* \leq \eta L \|\mathbf{y}_t - \mathbf{y}_{t-1}\| \\
\leq & \frac{G}{L} \sum_{s=2}^{t+1} \gamma^s \leq \frac{\gamma^2 G}{(1-\gamma)L} < \frac{G}{L},
\end{aligned} \tag{119}$$

and

$$\|\mathbf{x}_t - \mathbf{y}_{t-1}\| \leq \|\mathbf{x}_t - \mathbf{y}_t\| + \|\mathbf{y}_t - \mathbf{y}_{t-1}\| \leq \frac{(\gamma^2 + \gamma)G}{(1-\gamma)L} < \frac{G}{L}. \tag{120}$$

With these prepared, we invoke Lemma D.2 and choose $\beta = 2\gamma^2/(1-2\gamma)$ therein to derive

$$\begin{aligned}
B_t \leq & \frac{1 - 2\gamma}{4} \|\mathbf{x}_t - \mathbf{y}_{t-1}\|^2 + \frac{\gamma^2}{1-2\gamma} \|\mathbf{y}_t - \mathbf{x}_{t-1}\|^2 \\
\leq & \frac{1 - 2\gamma}{2} \|\mathbf{x}_t - \mathbf{x}_{t-1}\|^2 + \frac{1-2\gamma}{2} \|\mathbf{x}_{t-1} - \mathbf{y}_{t-1}\|^2 + \frac{1}{2} \|\mathbf{y}_t - \mathbf{x}_{t-1}\|^2 ,
\end{aligned} \tag{121}$$

where we use the value range requirement of $\gamma \in (0, 1/3]$. Similarly, choosing $\beta = \gamma$ in Lemma D.2 gives

$$D_t = \eta \langle \nabla f(\mathbf{x}_t) - \nabla f(\mathbf{y}_t), \mathbf{x}_t - \mathbf{y}_t \rangle \leq \gamma \|\mathbf{x}_t - \mathbf{y}_t\|^2. \tag{122}$$

Adding (121) and (122) to (118), we obtain

$$
\begin{aligned}
&\eta(f(\mathbf{x}_t) - f(\mathbf{x}_{t-1})) \\
\leq & \frac{1-2\gamma}{2} \|\mathbf{x}_t - \mathbf{x}_{t-1}\|^2 + \frac{1-2\gamma}{2} \|\mathbf{x}_{t-1} - \mathbf{y}_{t-1}\|^2 + \frac{1}{2} \|\mathbf{y}_t - \mathbf{x}_{t-1}\|^2 + \gamma \|\mathbf{x}_t - \mathbf{y}_t\|^2 \\
& - \frac{1}{2} \|\mathbf{x}_t - \mathbf{y}_t\|^2 - \frac{1}{2} \|\mathbf{y}_t - \mathbf{x}_{t-1}\|^2 - \frac{1}{2} \|\mathbf{x}_t - \mathbf{x}_{t-1}\|^2 \\
\leq & \frac{1-2\gamma}{2} \|\mathbf{x}_{t-1} - \mathbf{y}_{t-1}\|^2 - \frac{1-2\gamma}{2} \|\mathbf{x}_t - \mathbf{y}_t\|^2.
\end{aligned}
\tag{123}
$$

By induction, it's easy to verify that the above relation holds for all iterations before the current one. Summing them up, we have

$$
\begin{aligned}
\eta(f(\mathbf{x}_{t-1}) - f(\mathbf{x}_0)) &= \sum_{s=1}^{t-1} \eta(f(\mathbf{x}_s) - f(\mathbf{x}_{s-1})) \\
&\leq \sum_{s=1}^{t-1} \frac{1-2\gamma}{2} \|\mathbf{x}_{s-1} - \mathbf{y}_{s-1}\|^2 - \frac{1-2\gamma}{2} \|\mathbf{x}_s - \mathbf{y}_s\|^2 \\
&\overset{\mathbf{x}_0 = \mathbf{y}_0}{=} - \frac{1-2\gamma}{2} \|\mathbf{x}_{t-1} - \mathbf{y}_{t-1}\|^2.
\end{aligned}
\tag{124}
$$

Combining (123) with (124), we obtain the following bound for the suboptimality gap of $\mathbf{x}_t$:

$$\eta(f(\mathbf{x}_t) - f(\mathbf{x}_0)) \leq -\frac{1-2\gamma}{2} \|\mathbf{x}_t - \mathbf{y}_t\|^2 \leq 0. \tag{125}$$

Obviously, we have $f(\mathbf{x}_t) - f^* \leq f(\mathbf{x}_0) - f^*$ as $\eta > 0$. Applying Lemma D.3 yields $\|\nabla f(\mathbf{x}_t)\|_* \leq G$. It remains to show the suboptimality as well as the gradient bound for $\mathbf{y}_t$. By Lemma G.1, we have

$$
\begin{aligned}
&\eta(f(\mathbf{y}_t) - f(\mathbf{x}_{t-1})) \\
\leq & \underbrace{\eta\langle \nabla f(\mathbf{y}_t) - \nabla f(\mathbf{x}_{t-1}), \mathbf{y}_t - \mathbf{x}_{t-1} \rangle}_{E_t} + \underbrace{\eta\langle \nabla f(\mathbf{x}_{t-1}) - \nabla f(\mathbf{y}_{t-1}), \mathbf{y}_t - \mathbf{x}_{t-1} \rangle}_{F_t} \\
& - \frac{1}{2} \|\mathbf{x}_t - \mathbf{y}_t\|^2 - \frac{1}{2} \|\mathbf{y}_t - \mathbf{x}_{t-1}\|^2 - \frac{1}{2} \|\mathbf{x}_t - \mathbf{x}_{t-1}\|^2.
\end{aligned}
\tag{126}
$$

Before using Lemma D.2 to bound the terms $E_t$ and $F_t$, we need to verify the applicability of Lemma 2.7. By Lemma E.1, we have

$$\|\mathbf{y}_t - \mathbf{x}_{t-1}\| \leq \eta \|\nabla f(\mathbf{y}_{t-1})\|_* \leq \frac{\gamma G}{L} \leq \frac{G}{L}. \tag{127}$$

Choosing $\beta = \gamma$ in Lemma D.2, $E_t$ can be bounded as follows:

$$E_t = \eta\langle \nabla f(\mathbf{y}_t) - \nabla f(\mathbf{x}_{t-1}), \mathbf{y}_t - \mathbf{x}_{t-1} \rangle \leq \gamma \|\mathbf{y}_t - \mathbf{x}_{t-1}\|^2. \tag{128}$$

Similar routine should be done for $F_t$. By Lemmas 2.7 and H.1, we have

$$
\begin{aligned}
\|\mathbf{x}_{t-1} - \mathbf{y}_{t-1}\| &\leq \eta \|\nabla f(\mathbf{y}_{t-1}) - \nabla f(\mathbf{y}_{t-2})\|_* \leq \eta L \|\mathbf{y}_{t-1} - \mathbf{y}_{t-2}\| \\
&\leq \frac{G}{L} \sum_{s=1}^{t-1} \gamma^{s+1} \leq \frac{\gamma^2 G}{(1-\gamma)L} < \frac{G}{L}.
\end{aligned}
\tag{129}
$$

Choosing $\beta = \gamma^2/(1-2\gamma)$ in Lemma D.2, $F_t$ can be bounded as follows:

$$F_t \leq \frac{1-2\gamma}{2} \|\mathbf{x}_{t-1} - \mathbf{y}_{t-1}\|^2 + \frac{\gamma^2}{2(1-2\gamma)} \|\mathbf{y}_t - \mathbf{x}_{t-1}\|^2. \tag{130}$$

Adding (128) and (130) to (126), we obtain

$$
\begin{aligned}
&\eta(f(\mathbf{y}_t) - f(\mathbf{x}_{t-1})) \\
&\leq \gamma \|\mathbf{y}_t - \mathbf{x}_{t-1}\|^2 + \frac{1-2\gamma}{2} \|\mathbf{x}_{t-1} - \mathbf{y}_{t-1}\|^2 + \frac{\gamma^2}{2(1-2\gamma)} \|\mathbf{y}_t - \mathbf{x}_{t-1}\|^2 \\
&\quad - \frac{1}{2} \|\mathbf{x}_t - \mathbf{y}_t\|^2 - \frac{1}{2} \|\mathbf{y}_t - \mathbf{x}_{t-1}\|^2 - \frac{1}{2} \|\mathbf{x}_t - \mathbf{x}_{t-1}\|^2 \\
&\leq \frac{1-2\gamma}{2} \|\mathbf{x}_{t-1} - \mathbf{y}_{t-1}\|^2 - \frac{1-2\gamma}{2} \|\mathbf{x}_t - \mathbf{y}_t\|^2,
\end{aligned}
\tag{131}
$$

where the last inequality is due to $\gamma^2/(1-2\gamma) + 2\gamma \leq 1$ for any $\gamma \in (0, 1/3]$. Adding (124) to the above inequality, we have

$$
\eta(f(\mathbf{y}_t) - f(\mathbf{x}_0)) \leq -\frac{1-2\gamma}{2} \|\mathbf{x}_t - \mathbf{y}_t\|^2 \leq 0.
\tag{132}
$$

We immediately see that $f(\mathbf{y}_t) - f^* \leq f(\mathbf{x}_0) - f^*$. Then $\|\nabla f(\mathbf{x}_t)\|_* \leq G$ is a direct result based on Lemma D.3. Combining **Part 1** with **Part 2** finishes the proof. $\qquad\square$

*Proof of Theorem 3.9.* Applying Lemma H.1 to the updates defined in (12), we have

$$
\begin{aligned}
\forall \mathbf{x} \in \mathcal{X}, \eta \langle \nabla f(\mathbf{y}_t), \mathbf{y}_t - \mathbf{x} \rangle \leq{}& B(\mathbf{x}, \mathbf{x}_{t-1}) - B(\mathbf{x}, \mathbf{x}_t) - B(\mathbf{x}_t, \mathbf{y}_t) - B(\mathbf{y}_t, \mathbf{x}_{t-1}) \\
&+ \eta \langle \nabla f(\mathbf{y}_t) - \nabla f(\mathbf{y}_{t-1}), \mathbf{y}_t - \mathbf{x}_t \rangle.
\end{aligned}
\tag{133}
$$

By Lemma G.2, we have $\|\mathbf{y}_t - \mathbf{y}_{t-1}\| \leq \frac{G}{L} \sum_{s=1}^t 1/3^s \leq \frac{G}{2L} < \frac{G}{L}$ as we specify $\gamma = 1/3$, which satisfies the condition in Lemma D.2. Hence, the following result holds:

$$
\eta \langle \nabla f(\mathbf{y}_t) - \nabla f(\mathbf{y}_{t-1}), \mathbf{y}_t - \mathbf{x}_t \rangle \leq \frac{1}{12} \|\mathbf{y}_t - \mathbf{y}_{t-1}\|^2 + \frac{1}{3} \|\mathbf{y}_t - \mathbf{x}_t\|^2,
\tag{134}
$$

where we set $\beta = 2/3$ in Lemma D.2. Plugging this into (133) and recalling the property of Bregman divergence, we have

$$
\begin{aligned}
\eta \langle \nabla f(\mathbf{y}_t), \mathbf{y}_t - \mathbf{x} \rangle \leq{}& B(\mathbf{x}, \mathbf{x}_{t-1}) - B(\mathbf{x}, \mathbf{x}_t) - \frac{1}{2} \|\mathbf{x}_t - \mathbf{y}_t\|^2 - \frac{1}{2} \|\mathbf{y}_t - \mathbf{x}_{t-1}\|^2 \\
&+ \frac{1}{6} \|\mathbf{y}_t - \mathbf{x}_{t-1}\|^2 + \frac{1}{6} \|\mathbf{x}_{t-1} - \mathbf{y}_{t-1}\|^2 + \frac{1}{3} \|\mathbf{y}_t - \mathbf{x}_t\|^2 \\
\leq{}& B(\mathbf{x}, \mathbf{x}_{t-1}) - B(\mathbf{x}, \mathbf{x}_t) + \frac{1}{6} \|\mathbf{x}_{t-1} - \mathbf{y}_{t-1}\|^2 - \frac{1}{6} \|\mathbf{x}_t - \mathbf{y}_t\|^2,
\end{aligned}
\tag{135}
$$

which holds for any $\mathbf{x} \in \mathcal{X}$. Telescoping sum, we obtain

$$
\sum_{t=1}^T \langle \eta \nabla f(\mathbf{y}_t), \mathbf{y}_t - \mathbf{x} \rangle \leq B(\mathbf{x}, \mathbf{x}_0) - B(\mathbf{x}, \mathbf{x}_T) - \frac{1}{6} \|\mathbf{x}_T - \mathbf{y}_T\|^2 \leq B(\mathbf{x}, \mathbf{x}_0).
\tag{136}
$$

By Jensen's Inequality, we derive the following convergence rate:

$$
f(\overline{\mathbf{y}}_T) - f^* \leq \frac{1}{\eta T} \sum_{t=1}^T \langle \eta \nabla f(\mathbf{y}_t), \mathbf{y}_t - \mathbf{x}_* \rangle \leq \frac{B(\mathbf{x}_*, \mathbf{x}_0)}{\eta T},
\tag{137}
$$

where $\overline{\mathbf{y}}_T = \sum_{t=1}^T \mathbf{y}_t / T$. $\qquad\square$

# H. Analysis for Mirror Prox

The following lemma describes some basics of the main building blocks of the algorithms that require two prox-mapping per iteration (Korpelevich, 1976; Popov, 1980). The techniques primarily stem from the seminal work of Nemirovski (2004). For completeness, we provide the proof here.

**Lemma H.1.** *For any $\mathbf{g}_1, \mathbf{g}_2 \in \mathcal{E}^*$ and any $\mathbf{x}_0 \in \mathcal{X}$ satisfying the following algorithmic dynamics:*

$$\mathbf{x}_1 = \mathcal{P}_{\mathbf{x}_0}(\mathbf{g}_1), \quad \mathbf{x}_2 = \mathcal{P}_{\mathbf{x}_0}(\mathbf{g}_2), \tag{138}$$

*we have*

$$\forall \mathbf{x} \in \mathcal{X}, \langle \mathbf{g}_1, \mathbf{x}_1 - \mathbf{x} \rangle \leq B(\mathbf{x}, \mathbf{x}_0) - B(\mathbf{x}, \mathbf{x}_1) - B(\mathbf{x}_1, \mathbf{x}_0), \tag{139}$$

$$\forall \mathbf{x} \in \mathcal{X}, \langle \mathbf{g}_2, \mathbf{x}_2 - \mathbf{x} \rangle \leq B(\mathbf{x}, \mathbf{x}_0) - B(\mathbf{x}, \mathbf{x}_2) - B(\mathbf{x}_2, \mathbf{x}_0), \tag{140}$$

$$\forall \mathbf{x} \in \mathcal{X}, \langle \nabla f(\mathbf{x}_1), \mathbf{x}_1 - \mathbf{x} \rangle \leq B(\mathbf{x}, \mathbf{x}_0) - B(\mathbf{x}, \mathbf{x}_2) - B(\mathbf{x}_2, \mathbf{x}_1) - B(\mathbf{x}_1, \mathbf{x}_0)$$
$$- \langle \mathbf{g}_1 - \mathbf{g}_2, \mathbf{x}_1 - \mathbf{x}_2 \rangle + \langle \nabla f(\mathbf{x}_1) - \mathbf{g}_2, \mathbf{x}_1 - \mathbf{x} \rangle, \tag{141}$$

*and the continuity of the prox-mapping is characterized by*

$$\|\mathbf{x}_1 - \mathbf{x}_2\| = \|\mathcal{P}_{\mathbf{x}_0}(\mathbf{g}_1) - \mathcal{P}_{\mathbf{x}_0}(\mathbf{g}_2)\| \leq \|\mathbf{g}_1 - \mathbf{g}_2\|_*. \tag{142}$$

*Proof.* Since $\psi(\cdot)$ is 1-strongly convex, we can leverage an existing tool (Nemirovski, 2004, Lemma 2.1), which exploits the continuity of the operator $\mathcal{P}_{\mathbf{x}}(\cdot)$, to show (142). (139) and (140) is immediately proven after using the first-order optimality condition for projection under Bregman divergence (Nemirovski, 2004, Lemma 3.1) on both updates. Taking $\mathbf{x} = \mathbf{x}_2$ in (139) and adding (140), we obtain

$$\langle \mathbf{g}_2, \mathbf{x}_2 - \mathbf{x} \rangle + \langle \mathbf{g}_1, \mathbf{x}_1 - \mathbf{x}_2 \rangle \leq B(\mathbf{x}, \mathbf{x}_0) - B(\mathbf{x}, \mathbf{x}_2) - B(\mathbf{x}_2, \mathbf{x}_1) - B(\mathbf{x}_1, \mathbf{x}_0), \quad \forall \mathbf{x} \in \mathcal{X}. \tag{143}$$

Rearrange the terms as below:

$$\forall \mathbf{x} \in \mathcal{X}, \langle \mathbf{g}_2, \mathbf{x}_1 - \mathbf{x} \rangle \leq B(\mathbf{x}, \mathbf{x}_0) - B(\mathbf{x}, \mathbf{x}_2) - B(\mathbf{x}_2, \mathbf{x}_1) - B(\mathbf{x}_1, \mathbf{x}_0)$$
$$- \langle \mathbf{g}_1 - \mathbf{g}_2, \mathbf{x}_1 - \mathbf{x}_2 \rangle. \tag{144}$$

Adding $\langle \nabla f(\mathbf{x}_1) - \mathbf{g}_2, \mathbf{x}_1 - \mathbf{x} \rangle$ to both sides of this inequality completes the proof. $\square$

**Lemma H.2.** *Under Assumptions 2.5 and 3.3, consider the updates in (14). If $0 < \eta \leq 1/2\ell(2G)$ where $G := \sup\left\{\alpha \in \mathbb{R}_+ | \alpha^2 \leq 2(f(\mathbf{x}_0) - f^*)\ell(2\alpha)\right\}$, the following statements holds for all $t \in \mathbb{N}_+$:*

1. *(descent property)*      $f(\mathbf{x}_t) \leq f(\mathbf{x}_{t-1}), f(\mathbf{y}_t) \leq f(\mathbf{x}_{t-1})$;

2. *(bounded gradients)*    $\|\nabla f(\mathbf{y}_t)\|_* \leq G < \infty, \|\nabla f(\mathbf{x}_t)\|_* \leq G < \infty$.

*Proof.* We prove this lemma by induction.

**Part 1. Base Case**
We consider the case where $t = 1$. Similar to the proof of Lemma E.2, we argue that $f(\mathbf{y}_0) - f^* = f(\mathbf{x}_0) - f^* < \infty$. By Lemma 3.4, we have $\|\nabla f(\mathbf{y}_0)\|_* = \|\nabla f(\mathbf{y}_0)\|_* \leq G < \infty$. Now we look at the sequence $\{\mathbf{y}_t\}_{t \in \mathbb{N}}$ that mirror prox maintains, whose dynamics are exactly like mirror descent. For convenience, we denote $\ell(2G)$ by $L$. As $\eta \leq 1/(2L) \leq 1/L$, we can apply Lemma E.2 to conclude that $f(\mathbf{y}_1) \leq f(\mathbf{x}_0) = f(\mathbf{y}_0)$ and $\|\nabla f(\mathbf{y}_1)\|_* \leq G$. Next, we apply Lemma H.3 to the first round of the mirror prox algorithm. It's straightforward to verify that the conditions are satisfied for the case where $s = 1$. Hence, we have $f(\mathbf{x}_1) \leq f(\mathbf{x}_0)$. By Lemma D.3, we conclude that $\|\nabla f(\mathbf{x}_1)\|_* \leq G$.

**Part 2. Induction Step**
Suppose the statements hold for all $t \leq s - 1$ where $s > 2$. So we have $\|\nabla f(\mathbf{y}_{s-1})\|_* \leq G$ and $\|\nabla f(\mathbf{x}_{s-1})\|_* \leq G$. We also have $f(\mathbf{x}_{s-1}) \leq f(\mathbf{x}_{s-2}) \leq f(\mathbf{x}_0)$. Now we consider the case where $t = s$. Similar to the base case, we invoke Lemma E.2 to immediately conclude that $f(\mathbf{y}_s) \leq f(\mathbf{x}_{s-1}) \leq f(\mathbf{x}_0)$ and $\|\nabla f(\mathbf{y}_s)\|_* \leq G$. Next, we prove the other half of the statements. Note that the conditions in Lemma H.3 are all satisfied by the previous derivations. So we invoke Lemma H.3 to deduce that $f(\mathbf{x}_s) \leq f(\mathbf{x}_{s-1})$, which immediately implies $f(\mathbf{x}_s) \leq f(\mathbf{x}_0)$. Then $\|\nabla f(\mathbf{x}_s)\|_* \leq G$ is a direct result based on Lemma D.3. Combining **Part 1** with **Part 2** finishes the proof. $\square$

**Lemma H.3.** *Under Assumptions 2.5 and 3.3, consider the following updates for any $s \in \mathbb{N}_+$:*

$$\mathbf{y}_s = \mathcal{P}_{\mathbf{x}_{s-1}}(\eta \nabla f(\mathbf{x}_{s-1})), \quad \mathbf{x}_s = \mathcal{P}_{\mathbf{x}_{s-1}}(\eta \nabla f(\mathbf{y}_s)), \tag{145}$$

*which satisfies $0 < \eta \le 1/2\ell(2G)$ for some $G \in \mathbb{R}_+$. If $\|\nabla f(\mathbf{x}_{s-1})\|_* \le G$ and $\|\nabla f(\mathbf{y}_s)\|_* \le G$ hold, then we have $f(\mathbf{x}_s) \le f(\mathbf{x}_{s-1})$.*

*Proof.* Notice that

$$\langle \nabla f(\mathbf{x}_s), \mathbf{x}_s - \mathbf{x}_{s-1} \rangle = \underbrace{\langle \nabla f(\mathbf{y}_s), \mathbf{x}_s - \mathbf{y}_s \rangle}_{A_s} + \underbrace{\langle \nabla f(\mathbf{x}_s) - \nabla f(\mathbf{y}_s), \mathbf{x}_s - \mathbf{y}_s \rangle}_{B_s}$$
$$+ \underbrace{\langle \nabla f(\mathbf{x}_{s-1}), \mathbf{y}_s - \mathbf{x}_{s-1} \rangle}_{C_s} \tag{146}$$
$$+ \underbrace{\langle \nabla f(\mathbf{x}_s) - \nabla f(\mathbf{x}_{s-1}), \mathbf{y}_s - \mathbf{x}_{s-1} \rangle}_{D_s}.$$

We bound the four terms separately. $A_s$ and $C_s$ can be bounded by the property of prox-mapping (Nemirovski, 2004, Lemma 3.1). Choosing $\mathbf{x}_1 = \mathbf{y}_s, \mathbf{x}_0 = \mathbf{x}_{s-1}, \mathbf{g}_1 = \eta \nabla f(\mathbf{x}_{s-1}), \mathbf{x} = \mathbf{x}_{s-1}$ in (139), we obtain

$$\eta C_s = \langle \eta \nabla f(\mathbf{x}_{s-1}), \mathbf{y}_s - \mathbf{x}_{s-1} \rangle \le -B(\mathbf{x}_{s-1}, \mathbf{y}_s) - B(\mathbf{y}_s, \mathbf{x}_{s-1}). \tag{147}$$

Choosing $\mathbf{x}_2 = \mathbf{x}_s, \mathbf{x}_0 = \mathbf{x}_{s-1}, \mathbf{g}_2 = \eta \nabla f(\mathbf{y}_s), \mathbf{x} = \mathbf{y}_s$ in (140), we obtain

$$\eta A_s = \langle \eta \nabla f(\mathbf{y}_s), \mathbf{x}_s - \mathbf{y}_s \rangle \le B(\mathbf{y}_s, \mathbf{x}_{s-1}) - B(\mathbf{y}_s, \mathbf{x}_s) - B(\mathbf{x}_s, \mathbf{x}_{s-1}). \tag{148}$$

Adding them together, we have

$$\eta A_s + \eta C_s \le - B(\mathbf{x}_{s-1}, \mathbf{y}_s) - B(\mathbf{y}_s, \mathbf{x}_s) - B(\mathbf{x}_s, \mathbf{x}_{s-1})$$
$$\le -\frac{1}{2} \|\mathbf{x}_{s-1} - \mathbf{y}_s\|^2 - \frac{1}{2} \|\mathbf{y}_s - \mathbf{x}_s\|^2 - \frac{1}{2} \|\mathbf{x}_s - \mathbf{x}_{s-1}\|^2. \tag{149}$$

For $B_s$ and $D_s$, we shall use the effective smoothness property. For convenience, denote by $L := \ell(2G)$ as we did in Lemma 2.7. Observe that

$$\|\mathbf{x}_s - \mathbf{x}_{s-1}\| \le \eta \|\nabla f(\mathbf{y}_s)\|_* \le \frac{G}{2L} \le \frac{G}{L} \tag{150}$$

by Lemma E.1 and the given condition. Then we call Lemma D.2 to bound $\eta D_s$ as follows:

$$\eta D_s = \eta \langle \nabla f(\mathbf{x}_s) - \nabla f(\mathbf{x}_{s-1}), \mathbf{y}_s - \mathbf{x}_{s-1} \rangle \le \frac{1}{8} \|\mathbf{x}_s - \mathbf{x}_{s-1}\|^2 + \frac{1}{2} \|\mathbf{y}_s - \mathbf{x}_{s-1}\|^2 \tag{151}$$

As for $B_s$, we still need to verify that the distance between $\mathbf{x}_s$ and $\mathbf{y}_s$ is small enough in order to apply Lemma 2.7. We utilize (142) to show that

$$\|\mathbf{x}_s - \mathbf{y}_s\| = \|\mathcal{P}_{\mathbf{x}_{s-1}}(\eta \nabla f(\mathbf{y}_s)) - \mathcal{P}_{\mathbf{x}_{s-1}}(\eta \nabla f(\mathbf{x}_{s-1}))\| \le \eta \|\nabla f(\mathbf{y}_s) - \nabla f(\mathbf{x}_{s-1})\|_*. \tag{152}$$

Then by Lemma E.1 and $\|\nabla f(\mathbf{x}_{s-1})\|_* \le G$, it follows

$$\|\mathbf{y}_s - \mathbf{x}_{s-1}\| \le \eta \|\nabla f(\mathbf{x}_{s-1})\|_* \le \frac{G}{2L} \le \frac{G}{L}. \tag{153}$$

Using Lemma 2.7, we deduce that

$$\|\mathbf{x}_s - \mathbf{y}_s\| \le \eta \|\nabla f(\mathbf{y}_s) - \nabla f(\mathbf{x}_{s-1})\|_* \le \eta L \|\mathbf{y}_s - \mathbf{x}_{s-1}\| \le \frac{G}{4L} \le \frac{G}{L}. \tag{154}$$

Utilizing Lemma 2.7 again along with Cauchy-Schwarz Inequality, we have

$$\eta B_s = \eta \langle \nabla f(\mathbf{x}_s) - \nabla f(\mathbf{y}_s), \mathbf{x}_s - \mathbf{y}_s \rangle$$
$$\le \eta \|\nabla f(\mathbf{x}_s) - \nabla f(\mathbf{y}_s)\|_* \|\mathbf{x}_s - \mathbf{y}_s\| \le \eta L \|\mathbf{x}_s - \mathbf{y}_s\|^2 \le \frac{1}{2} \|\mathbf{x}_s - \mathbf{y}_s\|^2. \tag{155}$$

Adding up (149), (151) and (155), we obtain

$$\eta(A_s + B_s + C_s + D_s) \leq -\frac{1}{2}\|\mathbf{x}_{s-1} - \mathbf{y}_s\|^2 - \frac{1}{2}\|\mathbf{y}_s - \mathbf{x}_s\|^2 - \frac{1}{2}\|\mathbf{x}_s - \mathbf{x}_{s-1}\|^2$$
$$+ \frac{1}{8}\|\mathbf{x}_s - \mathbf{x}_{s-1}\|^2 + \frac{1}{2}\|\mathbf{y}_s - \mathbf{x}_{s-1}\|^2 + \frac{1}{2}\|\mathbf{x}_s - \mathbf{y}_s\|^2 \quad (156)$$
$$= -\frac{3}{8}\|\mathbf{x}_s - \mathbf{x}_{s-1}\|^2 \leq 0.$$

Since $\eta > 0$, it directly yields

$$f(\mathbf{x}_s) - f(\mathbf{x}_{s-1}) \leq \langle \nabla f(\mathbf{x}_s), \mathbf{x}_s - \mathbf{x}_{s-1} \rangle = A_s + B_s + C_s + D_s \leq 0. \quad (157)$$

$\square$

*Proof of Theorem 3.10.* By Lemma H.2, we have $\|\nabla f(\mathbf{y}_t)\|_* \leq G < \infty, \|\nabla f(\mathbf{x}_t)\|_* \leq G < \infty$ for all $t \in \mathbb{N}_+$. Since the bound for $\nabla f(\mathbf{x}_0) = \nabla f(\mathbf{y}_0)$ naturally holds, so we obtain

$$\max\{\|\nabla f(\mathbf{y}_t)\|_*, \|\nabla f(\mathbf{x}_t)\|_*\} \leq G < \infty, \quad \forall t \in \mathbb{N}. \quad (158)$$

Now it suffices to show the convergence rate is at the order of $O(1/T)$. Choosing $\mathbf{x}_0 = \mathbf{x}_{t-1}, \mathbf{x}_1 = y_t, \mathbf{g}_1 = \eta \nabla f(\mathbf{x}_{t-1}), \mathbf{x}_2 = \mathbf{x}_t, \mathbf{g}_2 = \eta \nabla f(\mathbf{y}_t)$ in Lemma H.1, we have

$$\forall \mathbf{x} \in \mathcal{X}, \langle \eta \nabla f(\mathbf{y}_t), \mathbf{y}_t - \mathbf{x} \rangle \leq B(\mathbf{x}, \mathbf{x}_{t-1}) - B(\mathbf{x}, \mathbf{x}_t) - B(\mathbf{x}_t, \mathbf{y}_t) - B(\mathbf{y}_t, \mathbf{x}_{t-1})$$
$$- \eta \langle \nabla f(\mathbf{x}_{t-1}) - \nabla f(\mathbf{y}_t), \mathbf{y}_t - \mathbf{x}_t \rangle. \quad (159)$$

By Lemma E.1 and (158), we have

$$\|\mathbf{y}_t - \mathbf{x}_{t-1}\| \leq \eta \|\nabla f(\mathbf{x}_{t-1})\|_* \leq \frac{G}{2L} \leq \frac{G}{L}. \quad (160)$$

As $\eta \leq 1/(2L)$, we set $\beta = 1$ in Lemma D.2 to derive

$$\eta \langle \nabla f(\mathbf{y}_t) - \nabla f(\mathbf{x}_{t-1}), \mathbf{y}_t - \mathbf{x}_t \rangle \leq \frac{1}{8}\|\mathbf{y}_t - \mathbf{x}_{t-1}\|^2 + \frac{1}{2}\|\mathbf{y}_t - \mathbf{x}_t\|^2. \quad (161)$$

Combine (159) with (161):

$$\forall \mathbf{x} \in \mathcal{X}, \langle \eta \nabla f(\mathbf{y}_t), \mathbf{y}_t - \mathbf{x} \rangle \leq B(\mathbf{x}, \mathbf{x}_{t-1}) - B(\mathbf{x}, \mathbf{x}_t) - \frac{1}{2}\|\mathbf{x}_t - \mathbf{y}_t\|^2 - \frac{1}{2}\|\mathbf{y}_t - \mathbf{x}_{t-1}\|^2$$
$$+ \frac{1}{8}\|\mathbf{y}_t - \mathbf{x}_{t-1}\|^2 + \frac{1}{2}\|\mathbf{y}_t - \mathbf{x}_t\|^2 \quad (162)$$
$$\leq B(\mathbf{x}, \mathbf{x}_{t-1}) - B(\mathbf{x}, \mathbf{x}_t) - \frac{3}{8}\|\mathbf{y}_t - \mathbf{x}_{t-1}\|^2,$$

where we also use the property of Bregman divergence. By convexity of $f$, we have

$$\forall \mathbf{x} \in \mathcal{X}, f(\mathbf{y}_t) - f(\mathbf{x}) \leq \langle \nabla f(\mathbf{y}_t), \mathbf{y}_t - \mathbf{x} \rangle \leq \frac{B(\mathbf{x}, \mathbf{x}_{t-1}) - B(\mathbf{x}, \mathbf{x}_t)}{\eta}. \quad (163)$$

Telescoping and taking $\mathbf{x} = \mathbf{x}_*$ on both sides of the inequality, we obtain

$$\sum_{t=1}^T [f(\mathbf{y}_t) - f^*] \leq \frac{B(\mathbf{x}_*, \mathbf{x}_0) - B(\mathbf{x}_*, \mathbf{x}_T)}{\eta} \leq \frac{B(\mathbf{x}_*, \mathbf{x}_0)}{\eta}. \quad (164)$$

Since the output is the average of iterates, i.e., $\overline{\mathbf{y}}_T = \sum_{t=1}^T \mathbf{y}_t/T$, the proof is finished by Jensen's inequality and the convexity of $f$. $\square$

# I. Analysis of Stochastic Mirror Descent

The lemmas in this section hold under the conditions of Theorem 4.2. For brevity, we do not restate this point.

**Lemma I.1.** *Suppose $\eta_s \le \frac{1}{2L}$ and that $\|\nabla f(\mathbf{x}_s)\|_* \le G, \|\mathbf{g}_s\|_* \le \frac{G}{\eta_{s+1}L}$ for all $0 \le s \le t-1$. For any $0 < \delta < 1$, with probability at least $1 - \delta$, for each $t \in [T]$, it holds that*

$$f(\mathbf{x}_t) - f(\mathbf{x}_0) \le 16\sigma^2 \log\left(\frac{2}{\delta}\right) \sum_{s=1}^{t} \frac{\eta_s^2}{\sum_{k=s}^{t} \eta_k}. \tag{165}$$

*Proof.* This lemma is built upon Liu & Zhou (2025a, Lemma 4.3). Under the conditions of $\|\nabla f(\mathbf{x}_s)\|_* \le G, \|\mathbf{g}_s\|_* \le \frac{G}{\eta L}$, in each step $t$ where we utilize $\ell*$-smoothness, it degenerates to the standard $L$-smoothness in Liu & Zhou (2025a). Thus, the proof of Lemma 4.3 in Liu & Zhou (2025a) remains valid. Substituting arbitrary $\mathbf{x} = \mathbf{x}_0$, noting that $M = 0$ and $1 \le 2\log\left(\frac{2}{\delta}\right)$ finishes the proof. $\square$

**Lemma I.2** (Bounded suboptimality gaps). *With probability at least $1 - \frac{\delta}{2}$, for all $t \in [T]$, we have*

$$f(\mathbf{x}_t) - f^* \le f(\mathbf{x}_0) - f^* + \frac{64\sigma}{\sqrt{\delta}}. \tag{166}$$

*In the meantime, it holds that*

$$\|\nabla f(\mathbf{x}_t)\|_* \le G, \ \|\mathbf{g}_t\|_* \le \frac{G}{\eta_{t+1}L}, \ \|\boldsymbol{\epsilon}_t\|_* \le \frac{G}{2\eta_{s+1}L} \quad \forall t - 1 \in [T]. \tag{167}$$

*Proof.* The essence underlying this lemma is to enforce the happening of events $\{A_t\}_{t=0}^{T-1}$ and $\{B_t\}_{t=0}^{T-1}$. For the latter, which governs the gradient estimates, we invoke Chebyshev's inequality under Assumption 4.1:

$$\mathbb{P}\left(\bigcup_{t=0}^{T-1} B_t\right) \le \sum_{t=0}^{T-1} \mathbb{P}\left(\|\boldsymbol{\epsilon}_t\|_* > \frac{G}{2\eta_{s+1}L}\right) \le \frac{4\eta^2 L^2 \sigma^2}{G^2} \le \frac{4\eta^2 G^4 \delta}{128^2 G^2} \le \frac{\delta}{4}, \tag{168}$$

where the last inequality is due to $\eta \le 32/G$ together with the following relation induced by Lemma D.3:

$$G^2 = 2LF = 2L\left(f(\mathbf{x}_0) - f^* + \frac{64\sigma}{\sqrt{\delta}}\right) \ge \frac{128L\sigma}{\sqrt{\delta}}. \tag{169}$$

On the other hand, for the former $\{A_t\}_{t=0}^{T-1}$, which governs the suboptimality gaps, we seek help from Lemma I.1. We proceed by induction to bound the failure probability of events $\{A_t\}_{t=0}^{T-1}$. For $t = 0$, $A_0$ trivially holds. By Lemma D.3, we have $\|\nabla f(\mathbf{x}_0)\|_* \le G$. Since $B_0$ holds with probability at least $1 - \frac{\delta}{4T}$ according to (168), we have

$$\eta_1 \|\mathbf{g}_0\|_* \le \eta_1 \|\nabla f(\mathbf{x}_0)\|_* + \eta_1 \|\boldsymbol{\epsilon}_0\|_* \le \frac{G}{2L} + \frac{\eta_1 G}{2\eta_1 L} = \frac{G}{L}.$$

Then, we invoke Lemma I.1 to derive

$$f(\mathbf{x}_1) - f(\mathbf{x}_0) \le \frac{16\eta\sigma^2}{\sqrt{T}}\log\left(\frac{8T}{\delta}\right) = \frac{16\sigma}{\sqrt{T}}\sqrt{\log\left(\frac{8T}{\delta}\right)} \quad \text{w.p.} \ge 1 - \frac{\delta}{4T},$$

which implies $A_0, A_1, B_0, B_1$ holds simultaneously w.p.$\ge 1 - \frac{3\delta}{4T} \ge 1 - \frac{\delta}{T}$. Now suppose $\cup_{s=0}^{t-1} A_s$ and $\cup_{s=0}^{t-1} B_s$ happens simultaneously w.p.$\ge 1 - \frac{t\delta}{2T}$. Analogously, the conditions of Lemma I.1 are satisfied since the suboptimality gaps as well as noise norms remain under control. By Lemma I.1, with probability at least $1 - \frac{\delta}{4T}$, we have

$$f(\mathbf{x}_t) - f(\mathbf{x}_0) \le \frac{16\sigma}{\sqrt{T}}\sqrt{\log\left(\frac{8T}{\delta}\right)} \sum_{s=1}^{t} \frac{\eta}{\sqrt{T}(t-s+1)} \le 16\sigma\left(1 + \log(t)\right)\sqrt{\frac{\log\left(\frac{8T}{\delta}\right)}{T}}$$

$$\le 32\sigma\log(T)\sqrt{\frac{\log\left(\frac{8T}{\delta}\right)}{T}} \le \frac{64\sigma}{e}\sqrt{\log\left(\frac{8}{\delta}\right) + 3} \le 37\sigma\sqrt{\log\left(\frac{8}{\delta}\right)} \le \frac{64\sigma}{\sqrt{\delta}},$$

which means $A_t$ happens w.p.$\geq 1 - \frac{\delta}{4T}$. Recall (168) and inductive basis $\cup_{s=0}^{t-1} A_s, \cup_{s=0}^{t-1} B_s$ happening w.p.$\geq 1 - \frac{t\delta}{4T}$. Taking union bounds across $\cup_{s=0}^{t-1} A_s, \cup_{s=0}^{t-1} B_s$ and $A_t, B_t$ gives $\cup_{s=0}^{t} A_s, \cup_{s=0}^{t} B_s$ happening w.p.$\geq 1 - \frac{(t+1)\delta}{2T}$. Hence, we have completed the proof. □

With the above preparations, we are ready to prove the main theorem.

*Proof of Theorem 4.2.* By (167) in Lemma I.2, we deduce that throughout the trajectory of (16), we can view the behavior of $\ell*$-smooth functions as the standard smooth functions with effective smoothness parameter $L$. In this way, the conditions for Lemma I.1 are satisfied. According to Eq.(30) in Liu & Zhou (2025a), with probability at least $1 - \frac{\delta}{2}$, we have

$$f(\mathbf{x}_T) - f^* \leq \frac{4B(\mathbf{x}_*, \mathbf{x}_0)}{\sum_{t=1}^{T} \eta_t} + 4\sigma^2 \left(1 + 2\log\left(\frac{4}{\delta}\right)\right) \sum_{t=1}^{T} \frac{\eta_t^2}{\sum_{s=t}^{T} \eta_s}.$$

Note that the above relation must build upon the premise of Lemma I.2, i.e., the occurrence of events $\cup_{t=0}^{T-1} A_t$ and $\cup_{t=0}^{T-1} B_t$. We take a union bound among these events to obtain

$$f(\mathbf{x}_T) - f^* \leq \frac{4B(\mathbf{x}_*, \mathbf{x}_0)}{\sum_{t=1}^{T} \eta_t} + 16\sigma^2 \log\left(\frac{4}{\delta}\right) \sum_{t=1}^{T} \frac{\eta_t^2}{\sum_{s=t}^{T} \eta_s}, \quad \text{w.p.} \geq 1 - \delta.$$

Noting that $\eta_t = \min\left\{\frac{1}{2L}, \frac{64}{G}, \sqrt{\frac{B(\mathbf{x}_*, \mathbf{x}_0)}{\sigma^2 \log\left(\frac{1}{\delta}\right) \log(T)}}\right\}$, we finally arrive at

$$f(\mathbf{x}_T) - f^* \leq O\left(\frac{LB(\mathbf{x}_*, \mathbf{x}_0)}{T} + \sigma\sqrt{\frac{B(\mathbf{x}_*, \mathbf{x}_0) \log\left(\frac{1}{\delta}\right) \log(T)}{T}} + \frac{GB(\mathbf{x}_*, \mathbf{x}_0)}{\sqrt{T}}\right)$$

$$\overset{(169)}{\leq} O\left(\frac{LB(\mathbf{x}_*, \mathbf{x}_0)}{T} + \sigma\sqrt{\frac{B(\mathbf{x}_*, \mathbf{x}_0) \log\left(\frac{1}{\delta}\right) \log(T)}{T}}\right), \quad \text{w.p.} \geq 1 - \delta,$$

which holds sufficiently large $T$ and small $\delta$. □

## J. Extension for Stochastic Convex Optimization Under New Noise Model

In this section, we provide new convergence results for SCO under $\ell*$-smoothness and a newly proposed noise model (Assumption J.1). Appendix J.1 introduces the new noise condition. Appendix J.2 gives the main result, providing a high-probability anytime convergence guarantee. Appendix J.3 compares it with other popular noise assumptions, where one can see why this new formulation can be meaningful and more expressive. Appendix J.4 provides the omitted proof.

### J.1. Generalized Bounded Noise Model

We formally introduce the generalized bounded noise assumption as below.

**Assumption J.1.** For all $t \in \mathbb{N}$, the stochastic gradients are unbiased:

$$\mathbb{E}_{t-1}[\boldsymbol{\epsilon}_t] = 0, \tag{170}$$

where the expectation $\mathbb{E}_{t-1}$ is conditioned on the past stochasticity $\{\boldsymbol{\epsilon}_s\}_{s=0}^{t-1}$. The noise satisfies the generalized bounded condition w.r.t. a non-decreasing polynomial function $\sigma : \mathbb{R}_+ \to \mathbb{R}_{++}$:

$$\|\boldsymbol{\epsilon}_t\|_* \leq \sigma(\|\nabla f(\mathbf{x}_t)\|_*), \quad \text{a.s.} \tag{171}$$

Moreover, the degree of the polynomial $\sigma$ is finite: $\deg(\sigma) < +\infty$.

This generalized bounded noise assumption[9] is inspired by the affine variance condition initially proposed by Bottou et al. (2018), where the noise level depends on the norm of the true gradient and is captured via an affine function. Assumption J.1

---

[9]The noise is bounded when the gradient is bounded, but can become unbounded if the gradient explodes.

encompasses the uniformly bounded noise (Zhang et al., 2020b) and affine noise (Liu et al., 2023b; Hong & Lin, 2024) condition with $\sigma(\alpha) \equiv \sigma$ and $\sigma(\alpha) \equiv \sigma_0 + \sigma_1\alpha$, respectively. Moreover, we underline that Assumption J.1 is never stronger than the commonly assumed finite moment or sub-Gaussian noise condition, with thorough discussions deferred to Appendix J.3.

For theoretical analysis, we also need the following bounded domain assumption, which is also widely adopted in the context of mirror descent (Zhang et al., 2023; 2024b; Yu et al., 2024; Bai et al., 2025).

**Assumption J.2.** The domain $\mathcal{X}$ is bounded in the sense that $\max_{\mathbf{x}\in\mathcal{X}} \psi(\mathbf{x}) - \min_{\mathbf{x}\in\mathcal{X}} \psi(\mathbf{x}) \leq D^2$.

Note that Assumption J.2 does not imply $\sup_{\mathbf{x},\mathbf{y}} B(\mathbf{x},\mathbf{y}) \leq D^2$, which is used in Eldowa & Paudice (2024, Lemma 9) to simplify the analysis. This stronger assumption is rarely used and can not be satisfied by the simplest instance of Bregman divergence induced by negative entropy functions.

## J.2. Theoretical Guarantees

Based on our new noise model, establishing a theoretical guarantee is highly challenging for the sake of the following two quantities that need to be controlled.

**Suboptimality gap** Due to the inherent stochasticity, the sequence $\{f(\mathbf{x}_t) - f^*\}_{t\in\mathbb{N}}$ may become unbounded. Recall from deterministic optimization, we directly link suboptimality gaps to the last-iterate convergence. However, stochastic gradient descent (SGD) (Robbins & Monro, 1951) primarily focuses on the average-iterate, and therefore, such a link does not hold. A fruitful line of work (Shamir & Zhang, 2013; Harvey et al., 2019; Orabona, 2020; Jain et al., 2021) explore the last-iterate convergence of SGD but often remains confined to Euclidean settings, compact domains, or uniformly bounded noise. Our solution is motivated by Orabona (2020), who elegantly converts average-iterate convergence of SGD to last-iterate in-expectation convergence for non-smooth objectives within Euclidean domains. Through a careful analysis of martingales and a refinement of their techniques, we extend the framework to $\ell_*$-smooth objectives in non-Euclidean settings. Leveraging this conversion framework, we prove that suboptimality gaps are bounded for all iterations simultaneously with high probability, indicating that the gradients are also bounded for all $t \in \mathbb{N}$. Such "anytime" characteristic is vital for our analysis, since Lemma 2.7 needs to be applied to every round of SMD.

**Dual norm of the stochastic gradient** In the context of SCO, we still need to exploit the local smoothness property (Lemma 2.7), which requires the distance between consecutive iterates to be bounded. In the deterministic case, we have $\|\mathbf{x}_{t+1} - \mathbf{x}_t\| \leq \|\nabla f(\mathbf{x}_t)\|_*$, and by Lemma 3.4, the boundedness of $\|\nabla f(\mathbf{x}_t)\|_*$ is inferred from finite suboptimality gaps. In contrast, for the stochastic case, it holds that $\|\mathbf{x}_{t+1} - \mathbf{x}_t\| \leq \|\mathbf{g}_t\|_*$, where the challenge lies in bounding $\|\mathbf{g}_t\|_*$. Clearly, the self-bounding property in Lemma 3.4 fails to bridge the stochastic gradient and the suboptimality gap. To address this, we leverage the noise model in Assumption J.1 to connect the noise level with the dual norm of the true gradient. As the suboptimality gap is bounded with high probability, the boundedness of the true gradient follows directly, implying that the noise is almost surely bounded via $\sigma(\cdot)$. Finally, we apply a union bound over these events to establish bounds on the stochastic gradient.

Based on the above reasoning, our analysis framework significantly differs from that of Li et al. (2023a, Section 5.2), who utilize a stopping-time technique (Williams, 1991) to show that the two key quantities are bounded before the stopping time $T$ with high probability. However, their method has two limitations: (i) it requires prior knowledge of the total number of iterations $T$; (ii) it lacks a convergence guarantee for the outputs generated before the final round. Moreover, their approach is ill-suited for SMD, as it heavily relies on the Euclidean inner product such that $\langle \mathbf{x},\mathbf{x}\rangle = \|\mathbf{x}\|_2^2$. Our main result is presented in Theorem J.3, whose proof is deferred to Appendix J.4.

**Theorem J.3.** *Under Assumptions 2.5, 3.3, J.1 and J.2, set*

$$0 < \eta_t \leq \min\left\{\frac{D}{\sigma(\|\nabla f(\mathbf{x}_{t-1})\|_*)\sqrt{t}}, \frac{\min\{1, \|\nabla f(\mathbf{x}_{t-1})\|_* / \sigma(\|\nabla f(\mathbf{x}_{t-1})\|_*)\}}{2\ell(2\|\nabla f(\mathbf{x}_{t-1})\|_*)}\right\}. \tag{172}$$

*With high probability, we have the following anytime convergence rate:*

$$f(\overline{\mathbf{x}}_t) - f^* = \widetilde{O}\left(\frac{1}{\sqrt{t}}\right), \quad \forall t \in \mathbb{N}_+, \tag{173}$$

*where $\overline{\mathbf{x}}_t = \sum_{s=1}^t \eta_s \mathbf{x}_s / \sum_{s=1}^t \eta_s$. Moreover, the gradients along the trajectory satisfy $\|\nabla f(\mathbf{x}_t)\|_* = \widetilde{O}(1), \forall t \in \mathbb{N}$.*

*Remark* J.4. Theorem J.3 achieves the classic $\widetilde{O}(1/\sqrt{t})$ anytime convergence rate (Shamir & Zhang, 2013; Zhang et al., 2024a; Bai et al., 2025) in the classic SCO. Note that existing results rely on a deterministic learning rate schedule proportional to $1/\sqrt{t}$. However, this strategy is too coarse to capture the local features of the objective under $\ell*$-smoothness. Consequently, we incorporate local curvature information into the learning rates, akin to the approach in Xie et al. (2024b), who consider online convex optimization under $\ell$-smoothness.

### J.3. Comparison of Noise Assumptions

Below, we briefly compare Assumption J.1 with commonly used assumptions in stochastic optimization.

**Bounded noise**  When $\sigma(\cdot)$ degenerates into a positive constant, Assumption J.1 recovers the uniformly bounded noise condition in the seminal work on $(L_0, L_1)$-smoothness (Zhang et al., 2020b), and it is also commonly adopted in other subfields (Koloskova et al., 2023; Zhang et al., 2023; Yu et al., 2024).

**Affine noise**  If $\sigma(\alpha) = \sigma_0 + \sigma_1\alpha$ or $\sigma(\alpha) = \sqrt{\sigma_0^2 + \sigma_1^2\alpha^2}$, Assumption J.1 recovers the almost sure $(\sigma_0, \sigma_1)$-affine noise condition in Liu et al. (2023b); Hong & Lin (2024) and Attia & Koren (2023), respectively. Its in-expectation version is also widely adopted in the existing literature (Shi et al., 2021; Jin et al., 2021; Wang et al., 2023a; Jiang et al., 2024).

**Finite moment**  Another popular setting is that the stochastic gradient has bounded $p$-th moment for some $p \in (1, 2]$ (Sadiev et al., 2023; Liu et al., 2024), which includes the uniformly bounded variance condition (Ghadimi & Lan, 2012; 2013; Arjevani et al., 2023) with $p = 2$. We underline that the generalized bounded noise is never stronger than this finite moment condition (cf. Proposition J.5).

**Sub-Gaussian noise**  This condition is strictly stronger than the finite variance condition and is extremely effective in deriving high-probability bounds (Juditsky et al., 2011; Liu & Zhou, 2024). As demonstrated by Proposition J.5, Assumption J.1 can not imply sub-Gaussian noise.

**Generalized heavy-tailed noise**  Recently, Liu & Zhou (2025b) propose a relaxation of the traditional heavy-tailed noises assumption (e.g., finite moment condition) in the form $\mathbb{E}_{t-1}[\|\boldsymbol{\epsilon}_t\|_2^p] \leq \sigma_0^p + \sigma_1^p \|\nabla f(\mathbf{x}_t)\|_2^p$ for some $p \in (1, 2], \sigma_0, \sigma_1 \in \mathbb{R}_+$. Yu et al. (2026, Assumptions 4a and 4b) further consider a coordinate-wise variant, as well as a matrix-geometry-aware heavy-tailed noise model. The almost sure version of the generalized heavy-tailed noise corresponds to our formulation with $\sigma(\alpha) = \sigma_0^p + \sigma_1^p\alpha^p$.

In real-world applications, it is common that the noise level is positively correlated with the true signal, as considered in most multiplicative noise models (Sancho et al., 1982; López-Martínez & Fabregas, 2003; Aubert & Aujol, 2008; Hodgkinson & Mahoney, 2021). To this end, we leverage an *arbitrary* non-decreasing polynomial $\sigma(\cdot)$ to capture this relation. We believe this formulation is expressive enough to accommodate diverse stochastic environments where the noise is potentially unbounded (Hwang, 1986; Loh & Wainwright, 2011; Diakonikolas et al., 2023) or has heavy tails (Zhang & Zhou, 2018; Lu et al., 2019; Xue et al., 2020; Gurbuzbalaban et al., 2021; Cutkosky & Mehta, 2021; Gou et al., 2023; Xue et al., 2023; Liu et al., 2024; Yu et al., 2026).

Lastly, we present a simple proof to demonstrate the relative strength of our generalized bounded noise assumption compared to the finite moment condition. A similar argument is presented in Liu & Zhou (2025b, Example A.1), who construct a counterexample that satisfies the generalized heavy-tailed noise condition but does not satisfy the finite moment condition.

**Proposition J.5.** *Assumption J.1 does not imply the finite moment condition given below:*

$$\forall t \in \mathbb{N}, p \in (1, 2], \quad \mathbb{E}_{t-1}[\|\boldsymbol{\epsilon}_t\|_*^p] < +\infty, \tag{174}$$

*where the expectation $\mathbb{E}_{t-1}$ is conditioned on the past stochasticity $\{\boldsymbol{\epsilon}_s\}_{s=0}^{t-1}$.*

*Proof.* Consider the special case where

(i) $p = 2$; (ii) $\sigma(\alpha) = \sigma_0 + \sigma_1\alpha$ for some $\sigma_0, \sigma_1 > 0$; (iii) $\|\boldsymbol{\epsilon}_t\|_* = \sigma(\|\nabla f(\mathbf{x}_t)\|_*)$ for all $t \in \mathbb{N}$.

In this case, the finite moment condition degenerates to the finite variance assumption (Ghadimi & Lan, 2012; 2013; Arjevani et al., 2023). It suffices to show that the variance could be unbounded under Assumption J.1 with the equality condition in

(iii). We proceed with

$$
\begin{aligned}
\mathbb{E}_{t-1}\left[\|\epsilon_t\|_*^2\right] &= \mathbb{E}_{t-1}\left[\sigma_0 + \sigma_1\|\nabla f(\mathbf{x}_t)\|_*\right]^2 \\
&= \sigma_0^2 + 2\sigma_0\sigma_1\mathbb{E}_{t-1}\left[\|\nabla f(\mathbf{x}_t)\|_*\right] + \sigma_1^2\mathbb{E}_{t-1}\left[\|\nabla f(\mathbf{x}_t)\|_*^2\right].
\end{aligned}
\tag{175}
$$

Intuitively, $\|\nabla f(\mathbf{x}_t)\|_*$ can not be directly controlled and thus could potentially diverge to infinity. For instance, consider a more extreme case where $f(\mathbf{x}) = \mathbf{x}^2$ and the algorithm outputs $\mathbf{x}_t = t\mathbf{x}_0$. In this way, we have

$$
\begin{aligned}
\mathbb{E}_{t-1}\left[\|\epsilon_t\|_*^2\right] &= \sigma_0^2 + 4\sigma_0\sigma_1\mathbb{E}_{t-1}\left[\|\mathbf{x}_t\|_*\right] + 4\sigma_1^2\mathbb{E}_{t-1}\left[\|\mathbf{x}_t\|_*^2\right] \\
&= \sigma_0^2 + 4t\sigma_0\sigma_1\mathbb{E}_{t-1}\|\mathbf{x}_0\|_* + 4t^2\sigma_1^2\mathbb{E}_{t-1}\|\mathbf{x}_0\|_*^2 \to \infty, \text{ if } t \to \infty,
\end{aligned}
\tag{176}
$$

which clearly contradicts the finite variance condition. Hence, Assumption J.1 is not strictly stronger than the finite moment condition. $\square$

## J.4. Proof of Theorem J.3

The lemmas in this subsection hold under the conditions in Theorem J.3. We do not restate this point for simplicity. First, we present the following two technical lemmas. Lemma J.6 is the Hoeffding-Azuma inequality for martingales, which is often used as a standard tool to derive the high probability bound. Lemma J.7 is a powerful technique to transform the convergence behavior from the average-iterate to the last-iterate.

**Lemma J.6.** *(Cesa-Bianchi & Lugosi, 2006) Let $V_1, V_2, \ldots$ be a martingale difference sequence with respect to some sequence $X_1, X_2, \ldots$ such that $V_i \in [A_i, A_i + c_i]$ for some random variable $A_i$, measurable with respect to $X_1, \ldots, X_{i-1}$ and a positive constant $c_i$. If $S_n = \sum_{i=1}^n V_i$, then for any $t > 0$,*

$$
\mathbb{P}[S_n > t] \le \exp\left(-\frac{2t^2}{\sum_{i=1}^n c_i^2}\right).
\tag{177}
$$

**Lemma J.7.** *(Orabona, 2020, Lemma 1) Let $\{\eta_s\}_{s=1}^t$ be a non-increasing sequence of positive numbers and $\{q_s\}_{s=1}^t$ be a sequence of positive numbers. Then*

$$
\eta_t q_t \le \frac{1}{t}\sum_{s=1}^t \eta_s q_s + \sum_{k=1}^{t-1}\frac{1}{k(k+1)}\sum_{s=t-k+1}^t \eta_s\left(q_s - q_{t-k}\right).
\tag{178}
$$

Next, we move on to the dynamics of SMD defined in (16). For convenience, we define the following quantities, which are frequently used in our analysis.

$$
\forall t \in \mathbb{N}, \quad G_t := \|\nabla f(\mathbf{x}_t)\|_*, \ L_t := \ell(2G_t), \ \sigma_t := \sigma(G_t)
\tag{179}
$$

**Lemma J.8.** *If $\eta_{t+1} \le \min\{1/(2L_t), G_t/(2L_t\sigma_t)\}$, then for any $\mathbf{x} \in \mathcal{X}$ we have*

$$
\eta_{t+1}[f(\mathbf{x}_{t+1}) - f(\mathbf{x})] \le B(\mathbf{x}, \mathbf{x}_t) - B(\mathbf{x}, \mathbf{x}_{t+1}) + \eta_{t+1}\langle\epsilon_t, \mathbf{x} - \mathbf{x}_t\rangle + \eta_{t+1}^2\sigma_t^2.
\tag{180}
$$

*Proof.* By Lemma H.1, we can derive

$$
\begin{aligned}
\forall \mathbf{x} \in \mathcal{X}, \ \langle\eta_{t+1}\nabla f(\mathbf{x}_t), \mathbf{x}_{t+1} - \mathbf{x}\rangle \le &\, B(\mathbf{x}, \mathbf{x}_t) - B(\mathbf{x}, \mathbf{x}_{t+1}) - B(\mathbf{x}_{t+1}, \mathbf{x}_t) \\
&+ \eta_{t+1}\langle\nabla f(\mathbf{x}_t) - \mathbf{g}_t, \mathbf{x}_{t+1} - \mathbf{x}\rangle.
\end{aligned}
\tag{181}
$$

In view of the step size $\eta_{t+1} \le \min\{1/(2L_t), G_t/(2L_t\sigma_t)\}$, we exploit the generalized smoothness property as follows

$$
\begin{aligned}
\|\mathbf{x}_{t+1} - \mathbf{x}_t\| &\le \eta_{t+1}\|\mathbf{g}_t\|_* \le \eta_{t+1}(\|\nabla f(\mathbf{x}_t)\|_* + \|\epsilon_t\|_*) \\
&\le \frac{G_t}{2L_t} + \frac{G_t}{2L_t\sigma_t}\cdot\sigma(\|\nabla f(\mathbf{x}_t)\|_*) \le \frac{G_t}{L_t},
\end{aligned}
\tag{182}
$$

where the first step is due to Lemma E.1; the last step leverages the non-decreasing property of $\sigma(\cdot)$. Now that (182) holds, we can utilize the local smoothness property in Lemma 2.7 and we follow the same steps as in (67) and (68) to derive

$$\langle \eta_{t+1} \nabla f(\mathbf{x}_t), \mathbf{x}_{t+1} - \mathbf{x}_* \rangle \geq \eta_{t+1}(f(\mathbf{x}_{t+1}) - f^*) - \frac{\eta_{t+1}L}{2} \|\mathbf{x}_{t+1} - \mathbf{x}_t\|^2. \tag{183}$$

Combining the above inequality with (181), we obtain

$$
\begin{aligned}
\eta_{t+1}[f(\mathbf{x}_{t+1}) - f(\mathbf{x})] \leq & B(\mathbf{x}, \mathbf{x}_t) - B(\mathbf{x}, \mathbf{x}_{t+1}) - B(\mathbf{x}_{t+1}, \mathbf{x}_t) + \frac{\eta_{t+1}L}{2} \|\mathbf{x}_{t+1} - \mathbf{x}_t\|^2 \\
& + \eta_{t+1}\langle \boldsymbol{\epsilon}_t, \mathbf{x} - \mathbf{x}_t \rangle + \eta_{t+1}\langle \boldsymbol{\epsilon}_t, \mathbf{x}_t - \mathbf{x}_{t+1} \rangle \\
\leq & B(\mathbf{x}, \mathbf{x}_t) - B(\mathbf{x}, \mathbf{x}_{t+1}) + \eta_{t+1}\langle \boldsymbol{\epsilon}_t, \mathbf{x} - \mathbf{x}_t \rangle - \frac{1}{2} \|\mathbf{x}_{t+1} - \mathbf{x}_t\|^2 \\
& + \frac{1}{4} \|\mathbf{x}_{t+1} - \mathbf{x}_t\|^2 + \eta_{t+1}^2 \|\boldsymbol{\epsilon}_t\|_*^2 + \frac{1}{4} \|\mathbf{x}_{t+1} - \mathbf{x}_t\|^2 \\
\leq & B(\mathbf{x}, \mathbf{x}_t) - B(\mathbf{x}, \mathbf{x}_{t+1}) + \eta_{t+1}\langle \boldsymbol{\epsilon}_t, \mathbf{x} - \mathbf{x}_t \rangle + \eta_{t+1}^2 \sigma_t^2,
\end{aligned}
\tag{184}
$$

where the second inequality uses Lemma D.2 and $\eta_{t+1} \leq 1/(2L_t)$. $\qquad \square$

**Lemma J.9.** *For any $\delta \in (0,1)$, define*

$$
\begin{aligned}
\widetilde{F}_t &:= 30 \max\left\{1, \frac{\sigma_{s-1}}{G_{s-1}}\right\} L_{t-1} D^2 \sqrt{\ln \frac{16}{\delta}} + 12 D\sigma_{t-1}\sqrt{\ln \frac{16}{\delta}} + 4D\sigma_{t-1}\sqrt{\ln \frac{2t^3}{\delta}} \ln t, \\
\widetilde{G}_t &:= \sup\left\{\alpha \in \mathbb{R}_+ | \alpha^2 \leq 2\widetilde{F}_t \ell(2\alpha)\right\}, \quad \forall t \in \mathbb{N}_+.
\end{aligned}
\tag{185}
$$

*Under Assumptions 3.3 and J.1, with probability at least $1 - \delta$, the following holds simultaneously for all $t \in \mathbb{N}_+$:*

$$f(\mathbf{x}_t) - f^* \leq \widetilde{F}_t = \widetilde{O}(1), \; G_t \leq \widetilde{G}_t = \widetilde{O}(1), \; L_t = \widetilde{O}(1), \; \sigma_t = \widetilde{O}(1). \tag{186}$$

*Proof.* We apply Lemma J.8 to get

$$\sum_{s=t_1}^{t_2} \eta_s \left[f(\mathbf{x}_s) - f(\mathbf{x})\right] \leq B(\mathbf{x}, \mathbf{x}_{t_1-1}) - B(\mathbf{x}, \mathbf{x}_{t_2}) + \sum_{s=t_1-1}^{t_2-1} \left(\eta_{s+1}\langle \boldsymbol{\epsilon}_s, \mathbf{x} - \mathbf{x}_s \rangle + \eta_{s+1}^2 \sigma_s^2\right), \tag{187}$$

which holds for any $\mathbf{x} \in \mathcal{X}$ and any indices $t_1 \leq t_2, t_1, t_2 \in \mathbb{N}$. For convenience, we define

$$q_s := f(\mathbf{x}_s) - f^*, \; V_s := \eta_{s+1}\langle \boldsymbol{\epsilon}_s, \mathbf{x}_* - \mathbf{x}_s \rangle, \; U_s^k := \eta_{s+1}\langle \boldsymbol{\epsilon}_s, \mathbf{x}_{t-k} - \mathbf{x}_s \rangle \tag{188}$$

for all $0 \leq s \leq t$, where $\mathbf{x}_* \in \operatorname{argmin}_{\mathbf{x} \in \mathcal{X}} f(\mathbf{x})$ (the existence of $\mathbf{x}_*$ can be implied by Assumptions 2.5 and J.2). Obviously, we have $q_s \geq 0$ and $\{V_s\}_{s=0}^{t-1}, \{U_s^k\}_{s=t-k+1}^{t-1}$ are martingale difference sequences. Then we have

$$\sum_{s=1}^{t} \eta_s q_s \leq B(\mathbf{x}_*, \mathbf{x}_0) + \sum_{s=0}^{t-1} V_s + \eta_{s+1}^2 \sigma_s^2 \overset{\text{Assumption J.2}}{\leq} D^2 + \sum_{s=1}^{t} \eta_s^2 \sigma_{s-1}^2 + \sum_{s=0}^{t-1} V_s. \tag{189}$$

The above bound usually relates to the average-iterate. To incorporate Lemma J.7, we consider the following relation:

$$
\begin{aligned}
\sum_{s=t-k+1}^{t} \eta_s(q_s - q_{t-k}) &= \sum_{s=t-k+1}^{t} \eta_s[f(\mathbf{x}_s) - f(\mathbf{x}_{t-k})] \\
&\overset{(187)}{\leq} B(\mathbf{x}_{t-k}, \mathbf{x}_{t-k}) - B(\mathbf{x}_{t-k}, \mathbf{x}_t) + \sum_{s=t-k}^{t-1} \left(\eta_{s+1}\langle \boldsymbol{\epsilon}_s, \mathbf{x}_{t-k} - \mathbf{x}_s \rangle + \eta_{s+1}^2 \sigma_s^2\right) \\
&\leq \sum_{s=t-k+1}^{t} \eta_s^2 \sigma_{s-1}^2 + \sum_{s=t-k+1}^{t-1} U_s^k.
\end{aligned}
\tag{190}
$$

With these preparations, we invoke Lemma J.7 to deduce

$$
\begin{aligned}
\eta_t q_t \leq & \frac{1}{t} \sum_{s=1}^{t} \eta_s q_s + \sum_{k=1}^{t-1} \frac{1}{k(k+1)} \sum_{s=t-k+1}^{t} \eta_s \left( q_s - q_{t-k} \right) \\
\leq & \frac{D^2}{t} + \frac{1}{t} \sum_{s=1}^{t} \eta_s^2 \sigma_{s-1}^2 + \frac{1}{t} \sum_{s=0}^{t-1} V_s^t \\
& + \sum_{k=1}^{t-1} \frac{1}{k(k+1)} \sum_{s=t-k+1}^{t} \eta_s^2 \sigma_{s-1}^2 + \sum_{k=1}^{t-1} \frac{1}{k(k+1)} \sum_{s=t-k+1}^{t-1} U_s^k .
\end{aligned}
\tag{191}
$$

Next, we bound the terms in (191). Recalling the definition of $\eta_s \leq D/(\sigma_{s-1}\sqrt{s})$, we have

$$
\begin{aligned}
\sum_{s=1}^{t} \eta_s^2 \sigma_{s-1}^2 &\leq D^2 \sum_{s=1}^{t} \frac{1}{s} \leq D^2 + D^2 \int_1^t \frac{1}{s} \mathrm{d}s = D^2 \left( 1 + \ln t \right), \\
\sum_{s=t-k+1}^{t} \eta_s^2 \sigma_{s-1}^2 &\leq D^2 \sum_{s=t-k+1}^{t} \frac{1}{s} \leq D^2 \int_{t-k}^t \frac{1}{s} \mathrm{d}s = D^2 \ln \left( \frac{t}{t-k} \right) \leq \frac{kD^2}{t-k}.
\end{aligned}
\tag{192}
$$

Then

$$
\begin{aligned}
\sum_{k=1}^{t-1} \frac{1}{k(k+1)} \sum_{s=t-k+1}^{t} \eta_s^2 \sigma_{s-1}^2 &\leq \sum_{k=1}^{t-1} \frac{D^2}{k(k+1)} \cdot \frac{k+1}{t-k} \\
= \sum_{k=1}^{t-1} \frac{D^2}{kt} + \sum_{k=1}^{t-1} \frac{D^2}{(t-k)t} &= \frac{2D^2}{t} \sum_{k=1}^{t-1} \frac{1}{t} \leq \frac{2D^2[1+\ln(t-1)]}{t}.
\end{aligned}
\tag{193}
$$

Next, we utilize Lemma J.6 to bound the sum of the martingale difference sequence. In view of $\|\mathbf{x}_1 - \mathbf{x}_2\| \leq 2\sqrt{2}D, \forall \mathbf{x}_1, \mathbf{x}_2 \in \mathcal{X}$ (cf. (Yu et al., 2024, Proposition A.6)), we have

$$
\begin{aligned}
V_s &\leq \eta_{s+1} \|\boldsymbol{\epsilon}_s\|_* \|\mathbf{x}_* - \mathbf{x}_s\| \leq \frac{D}{\sigma_s \sqrt{s+1}} \sigma_s \cdot 2\sqrt{2}D = \frac{2\sqrt{2}D^2}{\sqrt{s+1}}, \\
U_s^k &\leq \eta_{s+1} \|\boldsymbol{\epsilon}_s\|_* \|\mathbf{x}_{t-k} - \mathbf{x}_s\| \leq \frac{D}{\sigma_s \sqrt{s+1}} \sigma_s \cdot 2\sqrt{2}D = \frac{2\sqrt{2}D^2}{\sqrt{s+1}}.
\end{aligned}
\tag{194}
$$

where we make use of $\eta_s \leq D/(\sigma_{s-1}\sqrt{s})$. Hence, we invoke Lemma J.6 to deduce that with probability at least $1 - \delta/t$,

$$
\sum_{s=0}^{t-1} V_s^t \leq \sqrt{\frac{1}{2} \sum_{s=0}^{t-1} \left( \frac{2\sqrt{2}D^2}{\sqrt{s+1}} \right)^2 \ln \frac{1}{\delta}} \leq 2D^2 \sqrt{(1+\ln t)\ln \frac{t}{\delta}}.
\tag{195}
$$

Similarly, for a given index $k$, with probability at least $1 - \delta/t$,

$$
\sum_{s=t-k+1}^{t-1} U_s^k \leq \sqrt{\frac{1}{2} \sum_{s=t-k+1}^{t-1} \left( \frac{2\sqrt{2}D^2}{\sqrt{s+1}} \right)^2 \ln \frac{1}{\delta}} \leq 2D^2 \sqrt{\ln \left( \frac{t}{t-k+1} \right) \ln \frac{t}{\delta}}.
\tag{196}
$$

Taking the union bound over $k \in [t-1]$ and $\{V_s^t\}_{s=0}^{t-1}$, we conclude that with probability at least $1 - \delta$,

$$
\sum_{s=0}^{t-1} V_s^t \leq 2D^2 \sqrt{(1+\ln t)\ln \frac{t}{\delta}}, \quad \sum_{s=t-k+1}^{t-1} U_s^k \leq 2D^2 \sqrt{\ln \left( \frac{t}{t-k+1} \right) \ln \frac{t}{\delta}},
\tag{197}
$$

hold simultaneously for all $k \in [t-1]$. We proceed to bound the second term.

$$\sum_{k=1}^{t-1} \frac{1}{k(k+1)} \sum_{s=t-k+1}^{t-1} U_s^k$$

$$\leq 2D^2 \sqrt{\ln \frac{t}{\delta}} \sum_{k=1}^{t-1} \frac{1}{k(k+1)} \cdot \ln\left(1 + \frac{k}{t-k}\right) \cdot \frac{1}{\sqrt{\ln\left(\frac{t}{t-k}\right)}}$$

$$\leq 2D^2 \sqrt{\ln \frac{t}{\delta}} \underbrace{\sum_{k=1}^{t-1} \frac{1}{k(t-k)} \cdot \frac{1}{\sqrt{\ln\left(\frac{t}{t-k}\right)}}}_{\texttt{term-A}} \cdot \tag{198}$$

Then,

$$\texttt{term-A} = \sum_{k=1}^{t-1} \frac{1}{kt} \cdot \frac{1}{\sqrt{\ln\left(\frac{t}{t-k}\right)}} + \sum_{k=1}^{t-1} \frac{1}{t(t-k)} \cdot \frac{1}{\sqrt{\ln\left(\frac{t}{t-k}\right)}}$$

$$= \frac{1}{t} \sum_{k=1}^{t-1} \frac{1}{k} \left( \frac{1}{\sqrt{\ln\left(\frac{t}{t-k}\right)}} + \frac{1}{\sqrt{\ln\left(\frac{t}{k}\right)}} \right) \tag{199}$$

$$\overset{(200)}{\leq} \frac{1}{t} \sum_{k=1}^{t-1} \frac{1}{k} \left( \sqrt{t} + \frac{1}{\sqrt{\ln t}} \right) \leq \frac{1 + \ln t}{\sqrt{t}} + \frac{1 + \ln t}{t\sqrt{\ln t}}$$

where we use the relation $x/(1+x) \leq \ln(1+x), \forall x > -1$ and the following fact

$$\frac{1}{\sqrt{\ln\left(\frac{t}{t-k}\right)}} + \frac{1}{\sqrt{\ln\left(\frac{t}{k}\right)}} \in \left( \frac{2}{\sqrt{\ln 2}}, \frac{1}{\sqrt{\ln\left(1 + \frac{1}{t-1}\right)}} + \frac{1}{\sqrt{\ln t}} \right]. \tag{200}$$

Now that each term in (191) has been calculated, we combine all these results to obtain

$$\eta_t q_t \leq \frac{D^2}{t} + \frac{D^2(1 + \ln t)}{t} + \frac{2D^2[1 + \ln(t-1)]}{t}$$

$$+ \frac{2D^2\sqrt{(1 + \ln t)\ln \frac{t}{\delta}}}{t} + 2D^2 \sqrt{\ln \frac{t}{\delta}} \left( \frac{1 + \ln t}{\sqrt{t}} + \frac{1 + \ln t}{t\sqrt{\ln t}} \right). \tag{201}$$

which holds with probability at least $1 - \delta$. Dividing by $\eta_t$ and taking union bound on $\{q_s\}_{s \in \mathbb{N}_+}$ (Zhang et al., 2024a), we obtain

$$\forall t \in \mathbb{N}_+, \quad q_t \leq \frac{2L_{t-1}D^2(4 + 3\ln t)}{t} + \frac{4L_{t-1}D^2\sqrt{(1 + \ln t)\ln \frac{2t^3}{\delta}}}{t}$$

$$+ 4L_{t-1}D^2 \sqrt{\ln \frac{2t^3}{\delta}} \left( \frac{1 + \ln t}{\sqrt{t}} + \frac{1 + \ln t}{t\sqrt{\ln t}} \right)$$

$$+ \frac{D\sigma_{t-1}(4 + 3\ln t)}{\sqrt{t}} + \frac{2D\sigma_{t-1}\sqrt{(1 + \ln t)\ln \frac{2t^3}{\delta}}}{\sqrt{t}} \tag{202}$$

$$+ 2D\sigma_{t-1}\sqrt{\ln \frac{2t^3}{\delta}}(1 + \ln t) + 2D\sigma_{t-1}\sqrt{\ln \frac{2t^3}{\delta}} \frac{1 + \ln t}{\sqrt{t}\ln t},$$

where we use the well-known fact $\sum_{t=1}^{+\infty} 1/t^2 = \pi^2/6 < 2$. Denote by $F_t$ the RHS of the above inequality. For all $2 \leq t \leq T$, we have

$$
\begin{aligned}
F_t \leq &30 \max\left\{1, \frac{\sigma_{s-1}}{G_{s-1}}\right\} L_{t-1} D^2 \sqrt{\ln \frac{16}{\delta}} \\
&+ 12D\sigma_{t-1}\sqrt{\ln \frac{16}{\delta}} + 4D\sigma_{t-1}\sqrt{\ln \frac{2t^3}{\delta}} \ln t = \widetilde{F}_t.
\end{aligned}
\tag{203}
$$

We will show $\widetilde{F}_t = \widetilde{O}(1)$ by induction in the sequel. For the base case, we have $G_0 = \|\nabla f(\mathbf{x}_0)\|_* < \infty$. Hence $L_0, \sigma_0$ are absolute constants. Since the high probability bound (202) holds, we clearly have $F_1 \leq \widetilde{F}_1 < \infty$ and $G_1 \leq \widetilde{G}_1 < \infty$ by Lemma D.3. For the induction step, we have $G_{t-1} = \widetilde{O}(1)$ from the induction basis. Then under Assumptions 3.3 and J.1,

$$
L_{t-1} = \ell(2G_{t-1}) = \widetilde{O}(1), \ \sigma_{t-1} = \sigma(G_{t-1}) = \widetilde{O}(1),
\tag{204}
$$

hold with probability at least $1 - \delta$, which implies $F_t = \widetilde{O}(1)$. Then we invoke Lemma D.3 again to deduce that $G_t = \widetilde{O}(1)$. Combining the base case and the induction step finishes the proof. $\qquad \square$

*Proof of Theorem J.3.* In the proof of Lemma J.9, we have shown that with probability at least $1 - \delta$ (see (189), (192), (197) and (202)),

$$
\sum_{s=1}^{t} \eta_s q_s \leq D^2 + D^2(1 + \ln t) + 2D^2\sqrt{(1 + \ln t)\ln \frac{2t^3}{\delta}}, \quad \forall t \in \mathbb{N}_+.
\tag{205}
$$

For the average-iterate up to $t$-th round, by the convexity of $f$ and Jensen's Inequality, we have

$$
\begin{aligned}
f(\overline{\mathbf{x}}_t) - f^* &\leq \frac{\sum_{s=1}^{t} \eta_s q_s}{\sum_{s=1}^{t} \eta_s} \leq \frac{\overbrace{D^2(2 + \ln t) + 2D^2\sqrt{(1 + \ln t)\ln \frac{2t^3}{\delta}}}^{D_t}}{\sum_{s=1}^{t} \eta_s} \\
&\leq \frac{D_t}{\sum_{s=1}^{t} \frac{\min\{1, G_{s-1}/\sigma_{s-1}\}}{2L_{s-1}}} + \frac{D_t}{\sum_{s=1}^{t} \frac{D}{\sigma_{s-1}\sqrt{s}}} \leq \frac{\widetilde{L}_{t-1}^{\max} D_t}{t} + \frac{\sigma_{t-1}^{\max} D_t}{D\sqrt{t}},
\end{aligned}
\tag{206}
$$

where the second line is derived via the flooring technique (Liu et al., 2023c; Liu & Zhou, 2024); the last line uses $\sum_{s=1}^{t} 1/\sqrt{s} \geq \sqrt{t}$ and the definitions below:

$$
\widetilde{L}_t^{\max} := \max\left\{L_0, \max_{1 \leq s \leq t}\left[L_s \max\left(1, \frac{\sigma_{s-1}}{G_{s-1}}\right)\right]\right\}, \quad \sigma_t^{\max} := \max_{0 \leq s \leq t} \sigma_s.
\tag{207}
$$

By Lemma J.9, we have $G_t = \widetilde{O}(1)$, $L_t = \widetilde{O}(1)$, $\sigma_t = \widetilde{O}(1)$ that holds with high probability along the trajectory. Therefore, we have $\sigma_{t-1}^{\max} = \widetilde{O}(1), \widetilde{L}_{t-1}^{\max} = \widetilde{O}(1)$. In view of $D_t = \widetilde{O}(1)$, we conclude our proof by combining these quantities:

$$
\begin{aligned}
f(\overline{\mathbf{x}}_t) - f^* \leq &\frac{\widetilde{L}_{t-1}^{\max}\left(D^2(2 + \ln t) + 2D^2\sqrt{(1 + \ln t)\ln \frac{2t^3}{\delta}}\right)}{t} \\
&+ \frac{\sigma_{t-1}^{\max}\left(D(2 + \ln t) + 2D\sqrt{(1 + \ln t)\ln \frac{2t^3}{\delta}}\right)}{\sqrt{t}} = \widetilde{O}\left(\frac{1}{\sqrt{t}}\right).
\end{aligned}
\tag{208}
$$

$\qquad \square$

# K. Analysis for Non-convex Composite Mirror Descent

First, we introduce a standard yet useful lemma.

**Lemma K.1** (Lemma 6.4 in Lan (2020)). *Let $\mathcal{G}_t$ be defined in (18), it holds that*

$$
\langle \nabla f(\mathbf{x}_t), \mathcal{G}_t \rangle \geq \|\mathcal{G}_t\|^2 + \frac{\phi(\mathbf{x}_{t+1}) - \phi(\mathbf{x}_t)}{\eta}.
$$

The proof of Theorem 5.1 stems from the textbook analysis of mirror descent in the non-convex case (Lan, 2020, Theorem 6.5), together with our previous generalized smooth analysis.

*Proof of Theorem 5.1.* First, we provide a quick and intuitive explanation of our proof outline. Recall Lemma E.2, which bounds the suboptimality gaps $\{f(\mathbf{x}_t) - f^*\}$ along the optimization trajectory *without convexity*. This is exactly the key to the derivations in the non-convex case. Since we have $f(\mathbf{x}_t) - f^* \leq F$ and $\|\nabla f(\mathbf{x}_t)\|_* \leq G$ for all $t \in \mathbb{N}$, we immediately leverage the effective smoothness constant $L$ and reduce the analysis for $\ell_*$-smooth functions to the textbook analysis of standard $L$-smooth functions. Below, we present formal reasoning by induction.

When $t = 0$, the induction basis is trivially satisfied. Suppose for all $s \leq t - 1$ we have bounded suboptimality gaps and bounded gradients. Now that the conditions of Lemma 2.7 is satisfied, we apply Lemma 2.7 to the $t$-th iterate:

$$
\begin{aligned}
f(\mathbf{x}_{t+1}) \leq &\, f(\mathbf{x}_t) + \langle \nabla f(\mathbf{x}_t), \mathbf{x}_{t+1} - \mathbf{x}_t \rangle + \frac{L}{2} \|\mathbf{x}_{t+1} - \mathbf{x}_t\|^2 \\
\leq &\, f(\mathbf{x}_t) - \eta \langle \nabla f(\mathbf{x}_t), \mathcal{G}_t \rangle + \frac{\eta^2 L}{2} \|\mathcal{G}_t\|^2 \\
\overset{\text{Lemma K.1}}{\leq} &\, f(\mathbf{x}_t) - \eta \|\mathcal{G}_t\|^2 + \phi(\mathbf{x}_t) - \phi(\mathbf{x}_{t+1}) + \frac{\eta^2 L}{2} \|\mathcal{G}_t\|^2 .
\end{aligned}
$$

Since $\eta \leq 1/L$, after rearrangement we get

$$
f(\mathbf{x}_{t+1}) \leq F(\mathbf{x}_{t+1}) \leq F(\mathbf{x}_t) - \frac{\eta}{2} \|\mathcal{G}_t\|^2 \leq F(\mathbf{x}_t) \leq \cdots \leq F(\mathbf{x}_0), \tag{209}
$$

which implies $f(\mathbf{x}_{t+1}) - f^* \leq f(\mathbf{x}_0) - f^* + \phi(\mathbf{x}_0)$. So we deduce $\|\nabla f(\mathbf{x}_{t+1})\|_* \leq G$ by Lemma D.3. Hence, throughout the optimization trajectory, the bounded suboptimality gaps as well as gradients hold uniformly. Thus, we can start from (209) to perform a telescoping:

$$
\frac{1}{T} \sum_{t=0}^{t-1} \|\mathcal{G}_t\|^2 \leq \frac{F(\mathbf{x}_0) - F(\mathbf{x}_T)}{\eta T} \leq \frac{F(\mathbf{x}_0) - F^*}{\eta T},
$$

which completes the proof. $\qquad\square$

