# OpenReview forum: "Mirror Descent Under Generalized Smoothness"
_ICML.cc/2026/Conference — ICML 2026 regular_

### Official Review · Reviewer_fb7F · 2026-02-18

**Soundness:** 2
**Presentation:** 2
**Significance:** 4
**Originality:** 3
**Overall Recommendation:** 4
**Confidence:** 4

**Summary:**

This work studies the mirror descent under generalized smoothness with a non-Euclidean norm. It contains multiple contributions: (1) It proposes a new concept of generalized smoothness, which generalizes the traditional $\ell$-smoothness to a more general norm. (2) To reflect the effectiveness under a different norm, this paper considers the mirror descent algorithm and establishes the convergence guarantee for a series of algorithms. (3) A novel technique (a generalized self-bounding property --- Lemma 3.4) is developed.

**Compliance With Llm Reviewing Policy:**

Affirmed.

**Final Justification:**

The rebuttal didn't address my concern about the function class and have address my other issues. So I keep my score 4 as indicated in my review.

**Key Questions For Authors:**

1. On the page 1, the domain of the function $f$ is denoted by $\mathcal{X}$. Later, in the Notations (on page 2), it considers the Banach space $\mathcal{E}$. The relation between these two spaces is not very clear.  $\mathcal{E}$ might be the whole space and $\mathcal{X}$ is a subset of that. Do I understand it correctly? Do you have further requirements on $\mathcal{X}$, e.g. $\mathcal{X}$ should be a compact set or an open set?

2. I would suggest the author to list all contributions using (1), (2), ... in Section 1. The current version is too dense.

3. The author didn't use "mapsto $\mapsto$" correctly in Definition 2.4 (and throughout this paper). This notation is not used for showing the domain and range. See this: https://math.stackexchange.com/questions/936558/use-of-mapsto-and-to

4. On page 3, what is the definition of "strict twice-differentiability" (Line 139)?

5. On page 3, Assumption 2.5. What is the definition of "closed"? I might consider it as this one "every sub-level set is closed": https://en.wikipedia.org/wiki/Closed_convex_function

    But this definition may not be sufficient to imply the Part 2 of the proof of Proposition 2.6. It results in the following correctness issue --- see the point 6 below:

6. (correctness issue) In Line 1463, how does the definition of $s_b$ and Assumption 2.5 imply $\lim f(x(s))=\infty$?   It seems that Assumption 2.5 should be strengthen to "every sub-level set is compact" (it is equivalent to require the function $f$ to be coercive, that is, the function value tends to infinite near the boundary --- same as the used one $\lim f(x(s))=\infty$). Then this step would be fixed. **However**, this strengthen assumption cannot by implied by the continuity anymore (stated in Line 114--Line116) and the coercive function class is obviously more restrictive.

7. Also on page 27, the Part 1 proof of Proposition 2.6: I am not very clear about the covering technique. The proof seems to be very simple and it doesn't require the discretization. Here is my thought: As the domain $\mathcal{X}$ is second-countable (as it is a subset of the Euclidean space), every open covering has countable sub-covering. For every open set in this sub-covering, we define the zero-measure set using Rademacher's theorem. Then the measure of union of countable zero-measure sets is 0. Is there anywhere you have to use discretization?

8. Assumption 3.3 is not very clear. Is there any example showing that if this assumption doesn't hold, then sometime the algorithm will not converge? Or is this assumption made just for convenience?

---

I am unlikely further raise my positive score (rating 4) as I have already assumed all above issues will be addressed/clarified by the author, where I assume the correctness issue will be fixed by using a stronger assumption (it will make the function class less general).

**Strengths And Weaknesses:**

## Strengths

It is obvious that this paper delivers complete results for mirror descent under generalized smoothness, which is also absolutely significant. Mirror descent has wide applications in modern machine learning including the policy gradient method and has attracted increasing attention in recent years.

Moreover, all components that should be considered (including the convergence rate, the difference under different norms, and the extension to non-convex case) have been considered, which further improves the significance of this work.

The motivation is clear and the derivation for the new $\ell_*$-smoothness technically sounds and is an indeed original result.

## Weaknesses

However, there are two major weaknesses: (1) the presentation is not satisfying. A few steps are missing and some notations/definitions are confusing. And (2) a technical result Proposition 2.6 seems to be wrong (and it is likely to be caused by the confusing definition).

I put these concerns in the Question section below.

---

> ### Author Rebuttal · Authors · 2026-03-28
>
> We sincerely thank the reviewer for their precious time, thoughtful review, and valuable feedback. Below, we address your concerns in detail.
>
> ---
>
> **Q1: Correctness of Proposition 2.6 and Assumption 2.5**
>
> We truly apologize for this confusion. **We respectfully clarify that our derivation in Line 1463 is mathematically sound as currently written, and a coercive condition is not needed.** The reviewer is correct about the closeness definition, which we will make clearer in the next revision. The crux of Line 1463 is that we require an **open domain** $\mathcal X$ in Assumption 2.5. According to Boyd & Vandenberghe (2004, Section A.3.3), combining a closed function with an open domain mathematically guarantees that $f(x) \to \infty$ as $x$ approaches the boundary of the domain. This directly supports our deduction that $\lim f(x(s))=\infty$, eliminating the need for extra coercive assumptions. The exact formulation is also adopted by Li et al. (2023a, Assumption 1). We will gladly add a clarifying sentence and a reference to Boyd & Vandenberghe (2004) in the revision to ensure this property is explicitly clear to all readers.
>
> ---
>
> **Q2: Covering Technique in Proposition 2.6**
>
> We thank the reviewer for pointing out this elegant simplification. Our use of "discretization" referred to manually partitioning the domain into countable subsets based on gradient norm intervals to construct covers, an approach adopted in Li et al. (2023a). However, we fully agree with your assessment: applying Rademacher's theorem directly to the domain's inherent countable sub-cover yields the measure-zero result much more cleanly. We will replace our original derivation with this standard topological argument in the revised appendix.
>
> ---
>
> **Q3: Necessity of Assumption 3.3**
>
> **Assumption 3.3 is strictly necessary to prevent divergence and is not merely a technical convenience**. Consider the problem setup in Section 5. We focus on the Euclidean special case where $\\|\cdot\\|=\\|\cdot\\|_2,\psi(x)=\\|x\\|_2^2/2,\phi(x)=0$​, our mirror descent update reduces exactly to gradient descent (GD), and the convergence criterion (gradient mapping in Eq.(16)) becomes $\mathcal{G}_t=\nabla f(x_t)$. Li et al. (2023a, Theorem 5.4) show that for quadratic growth $\ell$, where $\ell(\alpha)=L_0+L_1\alpha^2$, and for any stepsize $\eta>0,f(x_0)-f^*\le\Delta_0$, there exists an objective $f$ such that GD either fails to reach a $1$-stationary point or requires at least $e^{L_1\Delta_0/8}/6$ iterations. Hence, if Assumption 3.3 is dropped, the guarantee of our Theorem 5.1 is false in general, as it is easy to see that the constants $G$ and $L$ are polynomial functions of the problem parameters like $L_0,L_1,\Delta_0$. This exponential lower bound makes GD highly inefficient and drastically diverges from traditional optimization theory under standard smoothness. The above negative example justifies the necessity of Assumption 3.3.
>
> Furthermore, we refer the reviewer to the generalized self-bounding property in Lemma 3.4, which guarantees that $\\|\nabla f(x)\\|\_\*^2\le2\ell(2\\|\nabla f(x)\\|\_\*)(f(x)-f^*)$. If $\ell$ were to grow quadratically, this inequality fails to provide meaningful upper bounds for gradient norms. The effective smoothness parameter $L$ in Eq.(5) will no longer be an absolute constant.
>
> While Assumption 3.3 is proven necessary for non-convex and stochastic settings, its strict necessity for convex deterministic mirror descent remains an open problem (as noted by Li et al., 2023a), which we plan to explore in future work.
>
> ---
>
> **Q4: Domain Spaces**
>
> The reviewer is correct regarding the relationship between the Banach space $\mathcal{E}$ and the domain $\mathcal{X}$. The domain $\mathcal{X}$ is a subset of $\mathcal{E}$. As stated in Assumption 2.5, we further require $\mathcal{X}$ to be an open set. We will clarify this relationship early in the notation section.
>
> ---
>
> **Q5: Contribution summary**
>
> Thank you for this great advice. We will reorganize the introduction to state our contributions point-by-point in a much clearer format.
>
> ---
>
> **Q6: Mapsto notation**
>
> We apologize for the imprecise use of the $\mapsto$ notation. We will ensure this is corrected to standard domain/range notation throughout the manuscript.
>
> ---
>
> **Q7: Strict twice-differentiability**
>
> The function class considered in this paper allows $f$ to be second-order differentiable almost everywhere in $\mathcal X$, i.e., it can fail on a set of measure zero. One could also follow Zhang et al. (2020b) to further relax this condition, but it requires more careful handling of integrations, which we omitted for simplicity. Similar arguments are utilized in Zhang et al. (2020b); Li et al. (2023a), as well.
>
> ---
>
> We believe most of these questions stem from our presentation and notation, which we will substantially improve in the final version to avoid further confusion. We are happy to respond to any remaining concerns.

---

> > ### Author Rebuttal · Reviewer_fb7F · 2026-03-31
> >
> > Thanks for the clarification and for providing the detailed references. I checked the "Boyd and Vandenberghe, 2004" and it gives a strong support for the result used in the proof; your proof is correct and I indeed made a mistake in my previous review.
> >
> > However, I would like to check it with the author:
> > * **This statement is correct** (according to Boyd & Vandenberghe (2004, Section A.3.3)): As $s\_n$ tends to the boundary point $s\_b$ of the domain of continuous function $f$, then $f(s\_n)$ tends to infinite. This is equivalent to the definition of closed function (when it is continuous).
> > * **This statement  is not correct:** When the optimization problem degenerates to the unconstrained setting, this assumption holds naturally for all continuous functions. (On page 3, Assumption 2.5).
> >
> >     **Here is why:** In the unconstrained setting (e.g. the domain of $f$ is $\mathcal{X} = \mathcal{E}$), the boundary of domain of $f$ is an empty set). In this case, you cannot use this condition (from Boyd & Vandenberghe (2004, Section A.3.3)) to characterize the closedness of a continuous function. In the proof of Proposition 2.6, it requires to take $s\_b \notin \mathcal{X}$ (Line 1461). This step in the proof is correct, but it requires the domain $\mathcal{X} \neq \mathcal{E}$. That is, the domain cannot be the whole space.
> >
> > ---
> >
> > To summarize my question:
> >
> > Therefore, I hope confirm with the author if my above understanding is correct or not. For example, the following continuous functions will be excluded by Assumption 2.5:
> > 1. $f(x)=x^2$ or $f(x)=x$ defined on the domain $\mathcal{X}=(-1,1)$. The boundary of the domain is $\pm 1$.
> > 2. All constant functions defined on any open set $\mathcal{X}$, possibly the whole space $\mathcal{E}$.
> > 3. $f(x)=\cos x$.
> >
> > Also, to make the proof of Proposition 2.6 correct, the domain $\mathcal{X}$ must be a proper subset of $ \mathcal{E}$ (i.e. they cannot be equal).

---

> > > ### Author Response · Authors · 2026-04-01
> > >
> > > We sincerely thank the reviewer for the prompt follow-up and for confirming the correctness of our current proof. Below, we address the remaining concerns in detail.
> > >
> > > ---
> > >
> > > > _**This statement is not correct:** When the optimization problem degenerates to the unconstrained setting, this assumption holds naturally for all continuous functions. (On page 3, Assumption 2.5)._
> > >
> > > We truly apologize for this confusion. **We respectfully clarify that this statement, also seen in Li et al. (2023a, last line on page 3), is correct**. We first formally introduce the definition of closeness to avoid further misunderstanding: a function $f$ is closed if its sub-level sets $\\{\mathbf{x}\in\mathcal{X}|f(\mathbf{x})\le a\\}$ are closed sets. By the definition of continuity, the preimage of a closed interval $(-\infty,a]$ under _any_ continuous mapping is always a closed set (see, e.g., Rudin (1976), _Principles of Mathematical Analysis_, Theorem 4.8 and its Corollary). Therefore, if the domain is the whole space ($\mathcal{X}=\mathcal{E}$), **every continuous function inherently has closed sub-level sets and is thus a closed function.**
> > >
> > > >_**Here is why:** In the unconstrained setting (e.g. the domain of $f$ is $\mathcal{X}=\mathcal{E}$), the boundary of domain of $f$ is an empty set). In this case, you cannot use this condition (from Boyd & Vandenberghe (2004, Section A.3.3)) to characterize the closedness of a continuous function._
> > >
> > > The reviewer correctly notes that the boundary of the whole space $\mathcal{E}$ is the empty set ($\partial\mathcal{E}=\emptyset$). Because there are no boundary points, the condition "$f(\mathbf{x})\to\infty$ as $\mathbf{x}$ approaches the boundary" is **vacuously true**. It does not exclude continuous functions like $\cos(x)$; it simply means that for $\mathcal{X}=\mathcal{E}$, there are no boundary points that need to be verified.
> > >
> > > >_In the proof of Proposition 2.6, it requires to take $s_b\notin\mathcal X$ (Line 1461). This step in the proof is correct, but it requires the domain $\mathcal{X}\ne\mathcal{E}$. That is, the domain cannot be the whole space._
> > >
> > > The reviewer expressed concern that Line 1461 ($\mathbf{x}(s_b)\notin\mathcal{X}$) requires $\mathcal{X}$ to be a proper subset of $\mathcal{E}$. **We respectfully clarify that this is not needed, and the current proof remains correct.** In our proof, we use contradiction to show that $\mathcal{B}(\mathbf{x},\tilde r(\\|\nabla f(\mathbf{x})\\|_*))\subset\mathcal{X}$. If $\mathcal{X}=\mathcal{E}$, then this first step is **trivially true**: every ball centered at $\mathbf{x}\in\mathcal E$ is automatically contained in $\mathcal E$ (as all points in the ball belong to $\mathcal E$), rendering the contradiction branch unnecessary. In other words, the "suppose $\tilde{\mathbf{x}}\notin\mathcal X$" argument in Line 1461 is in fact simply the opposite scenario, making the proof straightforward. From the above reasoning, **there is no need to assume that domain $\mathcal X$ must be a proper subset of $\mathcal E$.**
> > >
> > > We apologize if this corner case was not explicitly delineated, and we will add a clarifying remark in the revised proof to distinguish the trivial unconstrained case from the constrained case.
> > >
> > > >_For example, the following continuous functions will be excluded by Assumption 2.5..._
> > >
> > > 1. $x$ or $x^2$ on $(-1,1)$: excluded as they are not closed on this open domain.
> > > 2. constant functions: If the domain is the whole space $\mathcal{E}$, they are **included** since they are continuous functions defined on the whole space. If defined on a proper open subset of $\mathcal{E}$, they are excluded.
> > > 3. $\cos x$: **Included**, as it is a continuous function defined on the whole space.
> > >
> > > ---
> > >
> > > We greatly appreciate this engaging and rigorous discussion, which has undoubtedly helped us improve the clarity and precision of our manuscript. We hope these detailed clarifications fully resolve your remaining concerns.

---

### Official Review · Reviewer_jDPj · 2026-02-24

**Soundness:** 3
**Presentation:** 2
**Significance:** 2
**Originality:** 2
**Overall Recommendation:** 4
**Confidence:** 4

**Summary:**

The authors introduce the notions of $\ell*$-smoothness and $(\ell,r)*$-smoothness. Under these assumptions, the authors derive convergence guarantees for mirror descent and related first-order methods, with rates matching existing results.

**Compliance With Llm Reviewing Policy:**

Affirmed.

**Final Justification:**

Thank you for your careful responses to the follow-up questions. From a practical standpoint in MD, the choice of $\psi$ is important, so it is somewhat disappointing that $\ell\ast$-smoothness does not appear to improve the situation over the standard setting. For this reason, there is still some room for debate about how significant this contribution is relative to prior work. Important questions also remain, such as which function classes are covered, which are not, and how well the approach works in practice (I understand that addressing these points is beyond the scope of the paper and should be left for future work). At the same time, the authors suggest that the stepsize and the constant in the complexity bounds may be improved, so I do not think this setting is without practical value. In light of the authors' response from a practical perspective, I have raised my score to 4.

**Key Questions For Authors:**

1. How does the choice of Bregman function affect the link function $\ell$? Does the proposed framework suggest any principled way to choose the the Bregman function to obtain a ``better'' $\ell$ or smaller constants?

2. Assumption 3.3 requires $\ell$ to be sub-quadratic. How restrictive is this assumption? Please also provide explicit examples of (i) objectives satisfying $\ell\ast$-smoothness with a sub-quadratic $\ell$, and (ii) objectives that are $\ell\ast$-smooth but only admit a non-sub-quadratic $\ell$ (if such examples exist). What qualitative property of $f$ determines whether the corresponding link function $\ell$ can be chosen sub-quadratic? An intuitive explanation of the underlying mechanism would be helpful.

3. Why is $\ell\ast$-smoothness (as opposed to existing generalized smoothness) the right assumption for analyzing MD and its variants? Which parts of the analysis would fail under, e.g., $(L_0,L_1)$-smoothness, $\ell$-smoothness, locally $L$-smoothness, or relative smoothness? In particular, if analogous convergence guarantees can already be obtained under existing frameworks (and it seems plausible that relative smoothness could yield such results), what is the essential reason to introduce $\ell\ast$-smoothness as a new assumption?

**Limitations:**

yes

**Strengths And Weaknesses:**

***Strengths.***

- The authors propose $\ell\ast$-smoothness and $(\ell,r)\ast$-smoothness, generalized smoothness notions that extend $(L_0,L_1)$-smoothness and $\alpha$-symmetric generalized smoothness, respectively.
- Under $\ell\ast$-smoothness, the authors provide convergence analyses for MD and several variants (e.g., accelerated MD, optimistic MD, and mirror prox), achieving rates comparable to the classical smooth case.

***Weaknesses.***

1. While Appendix A summarizes multiple related notions ($(L_0,L_1)$-smoothness, $\alpha$-symmetric generalized smoothness, $\ell$-smoothness, and relative smoothness), their inclusion relations with $\ell\ast$-smoothness and $(\ell,r)\ast$-smoothness are not made explicit. In particular, it is unclear whether there are natural and separating examples that are $\ell\ast$-smooth (or $(\ell,r)\ast$-smooth) but do not fall into the other frameworks, and whether such examples arise in practical applications.

2. The contribution is primarily analytical: the algorithms themselves are standard, and practical guidance for settings with unknown $\ell$ and $G$ (and induced constants such as $L$) (e.g., adaptive stepsizes, line-search, or robust estimation of $\ell$ in optimization) is limited.

3. The authors do not discuss how to choose the Bregman function in relation to $\ell\ast$-smoothness (or $(\ell,r)\ast$-smoothness). Convergence results depend on the Bregman term $B(x_\ast,x_0)$ but in practice, the choice of Bregman function plays a crucial role in the performance of the algorithm. Clearer guidance on the role of the Bregman function and how it interacts with the link function $\ell$ would strengthen the paper.

---

> ### Author Rebuttal · Authors · 2026-03-28
>
> Many thanks for the constructive reviews!
>
> ---
> **Q1: Relation between Bregman function and link function**
>
> **The key point is that the link function $\ell$ is determined by the objective $f$ together with the chosen primal-dual geometry $(\\|·\\|,\\|·\\|_*)$, not directly by the Bregman function itself**. In Definition 2.1, $\ell*$-smoothness is _immediately defined_ after fixing a norm pair, through the inequality $\\|\nabla^2 f(x)h\\|\_\*\le\ell(\\|\nabla f(x)\\|\_\*)\\|h\\|$. By contrast, the Bregman function _enters later_ through a distance-generating function $\psi$ that is required to be $1$-strongly convex with respect to that same norm. Therefore, if we change $\psi$ but keep the same underlying norm geometry, the admissible link function $\ell$ does not change; what changes are the prox mapping, the Bregman term $B_\psi(x^*,x_0)$, and the initialization $x_0=\arg\min \psi$, which then affect the constants in the convergence bound.
>
> This also clarifies Weakness 3. The principled way to choose the Bregman function is: **first choose the geometry/norm in which the objective admits the most favorable link function $\ell$; then choose a computationally convenient strongly convex $\psi$ for that geometry.** In other words, $\ell$ is a property of $f$ under a chosen geometry, whereas $\psi$ is the algorithmic regularizer used to realize mirror descent in that geometry. **This is exactly analogous to the classical $L$-smooth setting**: even for standard $L$-smooth functions, the smoothness constant $L$ is determined by $f$ and the norm, while the Bregman function used by mirror descent is a separate design choice. Our framework follows the same logic, except that $L$ is replaced by the more general link function $\ell$. Proposition 2.8 illustrates this clearly: for logistic kernel / logistic regression, changing the geometry from Euclidean to $\ell_1/\ell_\infty$ changes the link from $\\|w\\|\_2\alpha$ to $\\|w\\|\_\infty\alpha$, which can improve the constants.
>
> ---
> **Q2: On Assumption 3.3**
>
> Assumption 3.3 is not very restrictive as **all explicit examples we study admit sub-quadratic link functions**. In Section 2.2, the logistic kernel / logistic regression examples have linear links. In Section 2.3, the GPT2 smoothness model is affine, of the form $\ell_i(\alpha)=L_0^i+L_1^i\alpha$ (and also the empirical example in Appendix C.3). Appendix C.2 further gives a constant-link example on the simplex. The reviewer could also check examples in our response to `Reviewer 12xo Q2`. Thus, all explicit examples currently appearing in the manuscript are constant, affine, or linear, and hence sub-quadratic.
>
> For non-sub-quadratic examples, consider $f(x)=x-\log x-1,x>0$. It's easy to show that $f$ is $\ell$-smooth with $\ell(\alpha)=(\alpha+1)^2$. This also gives the intuitive mechanism: _a sub-quadratic link is possible when the curvature grows strictly slower than $\\|\nabla f\\|^2$, and it becomes impossible when the Hessian is intrinsically quadratic in the gradient magnitude_.
>
> **We also refer the reviewer to our response to `Reviewer fb7F Q3` to see why Assumption 3.3 is necessary**. Briefly speaking, there exist hardness results on non-sub-quadratic $\ell$ that either cause divergence or have exponentially bad complexity.
>
> ---
> **Q3: Why is $\ell\*$-smoothness the right model?**
>
> 1. **$\ell\*$-smoothness naturally fits MD due to the usage of arbitrary primal-dual norm pairs, while previous generalized smoothness models only support Euclidean norms**. MD and its variants inherently rely on Bregman functions adapting to various geometries via diverse norms, Euclidean smoothness is not sufficient to analyze MD.
> 2. **$\ell\*$-smoothness has dimensional improvements over Euclidean models**. As seen in our examples in Section 2.2, Appendix C.1-C.3, response to `Reviewer 12xo Q2`, our smoothness reduces dimension factors compared to $\ell$-smoothness, ultimately bringing faster convergence (Appendix C.4).
> 3. **$(L_0,L_1)$- and relative smoothness fail to fit in our analysis framework directly**. See Section 3.2 & Appendix A.5 for details.
>
> ---
> **Q4: Related notions**
>
> $\ell*$-smoothness naturally encompasses $\ell$-smoothness (by choosing Euclidean geometry), $(L_0,L_1)$-smoothness (by Euclidean geometry plus affine $\ell$), $\alpha$-symmetric generalized smoothness (almost equivalent to $(L_0,L_1)$-smooth (Chen et al., 2023)). For relative smoothness, the comprehensive discussions on how $\ell*$-smoothness is more practical and meaningful can be found in Appendix A.5.
>
> ---
> **Q4: Practical guidance**
>
> Due to space limitations, we refer the reviewer to Riabinin et al. (2025, Appendix E.3.3), which contains practical guidelines for choosing learning rate $\eta$ (ultimate goal of estimating $\ell,G$) based on AdamW training trajectory. Since their smoothness is a special case of $\ell*$-smoothness, the methodology can be transferred to our setting. We will make sure to add this important note in the final version.

---

> > ### Author Rebuttal · Reviewer_jDPj · 2026-04-03
> >
> > Thank you for your rebuttal and for the helpful clarifications.
> >
> > While $(\ell,r)\ast$-smoothness appears to play an important role in the analysis, it remains unclear to me to what extent it improves the actual performance of mirror descent, since the method still depends crucially on the choice of the Bregman function $B$ (Q1). Indeed, the guarantees in Theorems 3.5, 3.7, 3.9, 3.10, and 4.2 all contain the term $B(x_\ast,x_0)$, and the associated constants may worsen depending on the choice of $B$. This is also true in the classical mirror descent setting, which makes it all the more important to clarify whether $(\ell,r)\ast$-smoothness offers a concrete advantage beyond the analysis itself. In your rebuttal, you wrote that one may "then choose a computationally convenient strongly convex $\psi$ for that geometry." However, this still leaves Q1 insufficiently clarified for me. I would appreciate a more concrete explanation of how such a $\psi$ should be chosen in practice.
> >
> > Regarding whether $(\ell,r)\ast$-smoothness (or $\ell\ast$-smoothness) is the right modeling framework (Q3), I would find the paper more compelling if it provided results showing that, for functions satisfying several relevant smoothness notions, such as $\ell\ast$-smoothness, $(\ell,r)\ast$-smoothness, $(L_0,L_1)$-smoothness, $\ell$-smoothness, local $L$-smoothness, or relative smoothness, the $(\ell,r)\ast$-smoothness framework yields improved step-size rules or better complexity bounds. Such evidence would help clarify the specific benefit of this framework, that is, $(\ell,r)\ast$-smoothness is the right model. At present, I do not think that Q3 has been sufficiently clarified either.

---

> > > ### Author Response · Authors · 2026-04-03
> > >
> > > We sincerely thank the reviewer for the thoughtful follow-up and for finding our earlier clarifications helpful. Below, we address the remaining concerns in detail.
> > >
> > > ---
> > > >_While $\ell*$-smoothness appears to play an important role in the analysis, it remains unclear to me to what extent it improves the actual performance of mirror descent, since the method still depends crucially on the choice of the Bregman function (Q1)._
> > >
> > > We respectfully clarify that the Bregman term $B(x_*,x_0)$ is exactly the same type of term as in classical mirror descent (MD). **The source of improvement in our framework is not this term, but the generalized smoothness constant**. For example, in Theorem 3.5, setting $\eta=1/L$ gives the rate $LB(x_*,x_0)/T$, where $L=\ell(2G)<\infty$ is the effective smoothness constant. By contrast, an $\ell*$-smooth objective also satisfies classic smoothness with constant $L_0=\sup_x\\|\nabla^2f(x)\\|$ whenever this quantity is finite. Textbook analysis of MD under classic smoothness yields a rate of $L_0B(x_*,x_0)/T$. **The key point is that $L_0$​ may be extremely large, or even infinite, because $\ell\*$-smooth functions need not have globally bounded Hessian. Our framework shows that the relevant quantity is instead the effective constant $L=\ell(2G)$, which remains finite by Lemma D.3**. Therefore, the improvement comes from replacing the crude worst-case curvature $L_0$​ by the much sharper effective smoothness constant $L$. The same principle applies throughout the paper for the other MD variants as well.
> > >
> > > >_I would appreciate a more concrete explanation of how such a $\psi$ should be chosen in practice._
> > >
> > > Based on previous reasoning, $\psi$ can be chosen in essentially the same way as standard MD in practice. For example, when optimizing within a simplex, one may consider negative entropy functions. When operating in Euclidean geometries, one may simply use square functions (degenerate to gradient descent).
> > >
> > > >_Regarding whether $\ell*$-smoothness is the right modeling framework (Q3), I would find the paper more compelling if it provided results showing that, for functions satisfying several relevant smoothness notions, the $\ell*$-smoothness framework yields **improved stepsize rules or better complexity bounds**._
> > >
> > > For **improved stepsize rules**, we refer the reviewer to Theorem 3.7 and our remarks in Lines 301-319. It is evident that our refined analysis yields a better $1/L$ stepsize, whereas previous $\ell$-smoothness has to use a smaller $1/L^2$ stepsize.
> > >
> > > For **better complexity bounds**, we refer the reviewer to Appendix C.4, where we explicitly show that $\ell*$-smoothness can improve complexity by factors as large as $n/\ln n$ and $n^{0.4}/\ln n$. These gains arise because $\ell*$-smoothness uses a primal-dual norm pair adapted to the problem geometry, which can lead to a substantially better link function $\ell$ with reduced dimensional dependence. The relevant examples include Proposition 2.8, Appendices C.2&C.3, and our rebuttal to `Reviewer RJ8N (3)`.
> > >
> > > More broadly, we would like to clarify why we believe $\ell*$-smoothness is a meaningful modeling framework.
> > > 1. **$\ell\*$-smoothness is the first non-Euclidean generalized smoothness notion, supporting arbitrary primal-dual norm pairs**. Previous notions such as $\ell$-smoothness and $(L_0,L_1)$-smoothness are inherently Euclidean and are stated in terms of $\\|·\\|_2$. This is exactly why they do not naturally fit MD, whose geometry is fundamentally non-Euclidean. If one insists on using Euclidean smoothness to analyze MD, one must pass through norm equivalence: \\[\\|\nabla f(x)-\nabla f(y)\\|\_\*\le C_1\\|\nabla f(x)-\nabla f(y)\\|_2\overset{\\#}{\le}C_1L\\|x-y\\|\_2\le C_1C_2L\\|x-y\\|,\\] where $C_1,C_2$​ are generally dimension-dependent norm-equivalence constants, and the marked step $\\#$ is the only place where we can exploit Euclidean smoothness. This inevitably introduces dimension-dependent degradation. By contrast, $\ell*$-smoothness is stated directly in the geometry used by MD, avoiding this loss.
> > > 2. **Even outside the context of MD, $\ell\*$-smoothness is still meaningful in the sense that (i) it captures a wider range of objectives in practice; and (ii) it can be used to analyze other non-Euclidean steepest descent methods, including SignSGD [1], Muon [2], and ASGO [3]**. For (i), please see our rebuttal to `Reviewer RJ8N`. For (ii), please see, e.g., our rebuttal to `Reviewer 12xo Q1`, where our smoothness model encompasses that of ASGO. We also cover the smoothness models in [4,5] & Riabinin et al. (2025), where SignSGD, Muon, and Scion are analyzed.
> > >
> > > ---
> > >
> > > **References**
> > >
> > > [1] SignSGD: Compressed optimisation for non-convex problems. ICML'18
> > >
> > > [2] Muon: An Optimizer for Hidden Layers in Neural Networks
> > >
> > > [3] ASGO: Adaptive structured gradient optimization. NeurIPS'25
> > >
> > > [4] Improved Analysis for Sign-based Methods with Momentum Updates. arXiv2507.12091
> > >
> > > [5] On the Convergence Analysis of Muon. arXiv2505.23737

---

### Official Review · Reviewer_RJ8N · 2026-03-11

**Soundness:** 3
**Presentation:** 3
**Significance:** 3
**Originality:** 3
**Overall Recommendation:** 5
**Confidence:** 3

**Summary:**

This paper considers a non-smooth minimization problem and extends mirror descent under a generalized smoothness condition - $\ell_*$-smoothness, which includes non-Euclidean geometries. Specifically, this paper provides convergence results for mirror-descent-type algorithms under the generalized smoothness conditions, which match the rates under the classic smoothness. Moreover, this paper establishes convergence for stochastic convex optimization and non-convex optimization.

**Compliance With Llm Reviewing Policy:**

Affirmed.

**Final Justification:**

I will maintain my positive score.

**Key Questions For Authors:**

See above.

**Limitations:**

Yes

**Strengths And Weaknesses:**

Strengths:
1. Based on the main paper, this paper is technically sound.Theoretical results and proof sketch make sense for me.
2. This paper is clearly written and well structured.
3. This paper extends the mirror descent to non-Euclidean geometries by introducing a generalized smoothness assumption, which is important from my perspective.

Weaknesses:
1. This paper shows experimentally that the $\ell_*$-smooth assumption aligns with the optimization trajectory of LLMs, it would be better if the authors can show more practical applications/examples that satisfy the generalized smooth assumption rather than classic smooth assumption.

---

> ### Author Rebuttal · Authors · 2026-03-28
>
> We sincerely thank the reviewer for their precious time, thoughtful review, and valuable feedback. Below, we provide more practical examples that satisfy our $\ell*$-smoothness assumption. All added figures and experimental results can be found in https://anonymous.4open.science/r/ICML2026_Submission8210-69C4.
>
> ---
>
> **(1) Language modeling**
>
> To further support our smoothness model in LLM training regimes, we provide more empirical evidence by varying model size and model architecture.
>
> Firstly, by following the same setup as in Section 2.3 and Figure 1 (nanoGPT implementation of GPT-2-small (124M)), we validate $\ell*$-smoothness on GPT-2-medium (355M) and GPT-2-large (770M). The results are shown in [1, Figure 1(a) and 1(b)], which shows that our smoothness aligns closely with practice.
>
> Secondly, we test the smoothness property by training a 6-layer transformer on WMT'16 multimodal machine translation de-en dataset, which is built upon [3, Section 5.2]. The smoothness estimation and experimental details follows from [3, Section 5.2]. The result in [1, Figure 1(d)] shows that our $\ell*$-smoothness model again closely characterizes the real curvature.
>
> **(2) Computer vision**
>
> As suggested by [4], CNN and Transformer typically exhibit distinct Hessian spectrums and smoothness properties. Thus, we also evaluate the trajectory smoothness during computer vision tasks (image classification on CIFAR-10), where the backbone model is CNN. We mostly follow the similar strategy as the GPT2 experiments to estimate smoothness. The rest experimental details remains the same as [2, Section 5.2]. The smoothness property of CNN head weight is shown in [1, Figure 1(c)], where a strong agreement between $\ell*$-smoothness and true smoothness is evident. We also plot the rest layers of CNN (`layer1.conv2.weight`, `layer1.norm2.bias`, `layer2.conv1.weight`, `layer2.norm1.bias`, `layer3.conv1.weight`) in https://anonymous.4open.science/r/ICML2026_Submission8210-69C4/fig_cnn/, where the results are consistent. The above empirical findings further enhances the practical applicability of $\ell*$-smoothness in computer vision tasks.
>
> **(3) Numerical examples**
>
> In Appendix C.3, we present a numerical example function defined in Eq.(25), whose smoothness properties are well-modeled by $\ell*$-smoothness in Figure 3. Figure 4 also depicts the advantages of our $\ell*$-smoothness over Euclidean $\ell$-smoothness in Li et al. (2023a).
>
> There is also another example in our response to `Reviewer 12xo Q2`, where $f(x)=(\mathbf{1}^\top x)^4/4,x\in\mathbb{R}^d$ with $\mathbf 1$ being all-ones vector. The structure of $f$ is ubiquitous in non-linear regression, polynomial neural networks, and self-attention mechanism approximations. We can show that $f\in\mathcal{F}\_\ell(\\|\cdot\\|\_2)$ with $\ell(\alpha)=d+2\sqrt{d}\alpha$, and that $f\in\mathcal{F}\_{\ell^* }(\\|\cdot\\|\_1)$ with $\ell^* (\alpha)=1+2\alpha$. The clear dimensional factor improvement further highlights the practical value of $\ell*$-smoothness.
>
> ---
>
> We sincerely appreciate the reviewer's advice. We will include the above examples and discussions in the final version, which will undoubtedly strength the paper.
>
> **References**
>
> [1] https://anonymous.4open.science/r/ICML2026_Submission8210-69C4/all.pdf. Submission8210 Authors
>
> [2] Riabinin et al. _Gluon: Making Muon & Scion Great Again! (Bridging Theory and Practice of LMO-based Optimizers for LLMs)_. arXiv preprint arXiv:2505.13416
>
> [3] Crawshaw et al. _Robustness to Unbounded Smoothness of Generalized SignSGD_. In NeurIPS 2022
>
> [4] Zhang et al. *Why transformers need Adam: A Hessian perspective*. In NeurIPS 2024

---

> > ### Author Rebuttal · Reviewer_RJ8N · 2026-04-01
> >
> > I will keep the score unchanged.

---

> > > ### Author Response · Authors · 2026-04-01
> > >
> > > Dear Reviewer RJ8N,
> > >
> > > Thank you for your positive feedback and for confirming that your concerns are fully resolved. We will incorporate the additional examples into the final version to further strengthen the paper.
> > >
> > > Best,
> > >
> > > Authors

---

### Official Review · Reviewer_12xo · 2026-03-13

**Soundness:** 3
**Presentation:** 3
**Significance:** 2
**Originality:** 3
**Overall Recommendation:** 4
**Confidence:** 3

**Summary:**

This paper studies mirror descent under a relaxed notion of smoothness that goes beyond the classical Lipschitz gradient assumption. The authors introduce $\ell*$‑smoothness, which measures Hessian growth using a general norm and its dual, allowing the analysis to extend naturally to non‑Euclidean geometries. Under this framework, the paper establishes convergence guarantees for mirror descent and several related variants—including accelerated, optimistic, mirror prox, stochastic, and composite mirror descent. These algorithms match the rates known under standard smoothness.

**Compliance With Llm Reviewing Policy:**

Affirmed.

**Final Justification:**

The paper analyzes several algorithms for a broad class of $\ell*$‑smoothness functions. The theoretical results are clean and easy to follow.

**Key Questions For Authors:**

1. Can authors comment about the smoothness assumption in this paper, and its connection to the conditions in the following two references?

(a) An, Kang, et al. "Asgo: Adaptive structured gradient optimization." arXiv preprint arXiv:2503.20762 (2025).
(b) Zhang, Yushun, et al. "Why transformers need adam: A hessian perspective." NeurIPS 2024.

2. In Section 2.2, the function $f_2$ is presented as a motivating example for the $\ell$-smoothness condition.  $f_2$ smoothness constant $L$ depends on $\|w\|^2$. However, even if we characterize it with $\ell$-smoothness condition, the $\ell$ function still depends on $\|w\|^2$.  Could the authors clarify in what sense this example demonstrates an advantage over classical smoothness assumptions?

3. In Figures 1 and 2, how are $L_1$ and $L_0$ decided? In Equation (3), is the $\nabla_i f$ computed with all data without noise?

**Limitations:**

yes.

**Strengths And Weaknesses:**

**Strength**

1. Compared with the standard $L$-smoothness assumption, this paper relaxes the assumption a lot. It also covers the $(L_0, L_1)$-smoothness, and $\ell$-smoothness in the Euclidean sense. It can attain a much bigger function class.

2. The authors introduce this $\ell*$‑smoothness condition, and in Section 2, several properties and characterizations are provided.

3. The paper is very systematic.  A broad family of mirror-type algorithms is analyzed, and the resulting convergence theorems are clearly stated and easy to follow.

4. The main ideas are well articulated. The analysis proceeds in a logical sequence: function descent is first established with appropriate step sizes, Lemma 3.4 is then used to bound the gradient’s dual norm, and finally the $\ell*$-smoothness condition is leveraged to obtain effective smoothness bounds along the optimization trajectory.


**Weaknesses**

1. Although the paper is systematic, I am not very excited about it. The $\ell*$‑smoothness condition is extended from the $\ell$‑smoothness. While there are technical challenges, the novelty of the idea is limited.

2. The practical relevance of the generalized smoothness assumptions remains somewhat unclear. As far as I know, there is little theoretical evidence showing $(L_0, L_1)$-smoothnesshold in the neural network landscape.

3. The current analysis framework may be hard to extend to the stochastic setting without sub‑Gaussian noise assumptions.


Overall, while the paper is technically careful and well written, I find the contribution borderline in terms of novelty and practical impact.

---

> ### Author Rebuttal · Authors · 2026-03-28
>
> Many thanks for the constructive reviews!
>
> ---
> **Q1: Smoothness comparison**
>
> We thank the reviewer for pointing out two important works on the block diagonal structural properties of Hessians in neural networks. **We underline that their smoothness models are in fact a strict special case of our $\ell\*$-smoothness**. For ASGO, the authors assume 1-smoothness w.r.t. $\\|·\\|\_{L}$, which is equivalent to $-\mathbf{L}\preceq\nabla^2f(x)\preceq\mathbf{L}$ (per ASGO's notation, $\mathbf{L}=diag(L,..,L)$). Then, it holds that
> \\[
> \\|\nabla^2f(x)h\\|\_{\mathbf{L}^{-1}}=\\|\mathbf{L}^{-1/2}\nabla^2f(x)h\\|\_2\le\\|\mathbf{L}^{-1/2}\nabla^2f(x)\mathbf{L}^{-1/2}\\|\_{op}\\|\mathbf{L}^{1/2}h\\|\_2\le1·\\|h\\|\_{\mathbf L},
> \\]
> which implies $f\in\mathcal{F}\_{\ell}(\\|·\\|\_{\mathbf L})$ with $\ell(\alpha)\equiv1$. Our framework is strictly more general, because it additionally allows the curvature scale to depend on the gradient magnitude through a nonconstant link $\ell$, and it is not restricted to quadratic matrix-induced norms like $\\|·\\|_{\mathbf L}$.
>
> ---
> **Q2: Example function $f_2$**
>
> We refer the reviewer to Appendix C.1.2, where we consider $f_2$ with scaling factor $C$ in Eq.(20). Classical smoothness constant $L=C\\|w\\|_2^2$, while Euclidean generalized smoothness ($\ell$-smoothness) gives $\ell(\alpha)=\\|w\\|\_2\alpha$. Although $\ell$ still depends on $\\|w\\|\_2$, it eliminates the scaling factor $C$ and reduces the quadratic dependence on $\\|w\\|\_2$ to linear dependence. The first advantage brings favorable scale-invariant properties and the second advantage is also meaningful when the domain $\mathcal X$ is large. However, this gain becomes more prominent when it comes to our $\ell\*$-smoothness, which gives $\ell(\alpha)=\\|w\\|\_\infty\alpha$. For practical high-dimensional settings or the cases where $w$ is dense, our $\ell\*$-smoothness can reduce the dimension factor up to $d$ plus a potentially large scaling factor $C$ compared to classical smoothness. For Euclidean $\ell$-smoothness, there is still $C\sqrt d$ improvement.
>
> To further support our smoothness model beyond the examples in Appendix C.1, C.2 & C.3, here we provide another example to **(i) justify the necessity of generalized smoothness**, and **(ii) show the advantage of $\ell\*$-smoothness over Euclidean $\ell$-smoothness**. Consider the polynomial function $f(x)=(\mathbf{1}^\top x)^4/4$, where $x\in\mathbb{R}^d$ and $\mathbf 1$ is all-ones vector. This type of structure frequently appears in non-linear regression, polynomial neural networks, and self-attention mechanism approximations. The gradient and Hessian are given by $\nabla f(x)=(\mathbf{1}^\top x)^3\mathbf{1}$ and $\nabla^2f(x)=3(\mathbf{1}^\top x)^2\mathbf{1}\mathbf{1}^\top$, respectively. Classic smoothness constant is $L=\\|\nabla^2f(x)\\|_2=3d(\mathbf{1}^\top x)^2$, which can grow arbitrarily large. For $\ell$ and $\ell*$-smoothness, it's easy to show that $\ell(\alpha)=d+2\sqrt{d}\alpha$ and $\ell^*(\alpha)=1+2\alpha$. Evidently, $\ell\*$-smoothness exhibits a substantial dimensional improvement over $\ell$-smoothness, highlighting its practical edge.
>
> ---
> **Q3: Figures 1&2**
>
> The fitted $L_0,L_1$​ values are obtained in the same way as in Riabinin et al. (2025, Section 5.1). Specifically, we first estimate the trajectory smoothness constant in Line 1031, and then fit $L_0,L_1$​ by minimizing the discrepancy between this estimate and the affine approximation in Line 1036.
>
> Eq.(3) is our theoretical model, which is stated for all data without noise. Empirically, however, we did not utilize Eq.(3) to compute smoothness. Instead, only the current batch is accessed and used to calculate the trajectory curvature in Line 1031.
>
> ---
> **Q4: Practical relevance of $(L_0,L_1)$-smoothness**
>
> It is indeed difficult to prove the existence of $(L_0,L_1)$-smooth NN landscapes. However, **our $\ell\*$-smoothness relaxes the linear $(L_0,L_1)$-relation and allows more general, geometry-dependent links, which may capture a wider range of objectives in practice**. We have conducted more empirical validations in our response to `Reviewer RJ8N`, which further verifies the practical value of $\ell*$-smoothness.
>
> ---
> **Q5: Sub-Gaussian noise**
>
> **In Appendix J, we introduce a new noise model (Assumption J.1) that is never stronger than sub-Gaussian noise assumption (see Appendix J.3) and establish sharp convergence rates under this condition (Theorem J.3)**. The whole analysis framework is essentially the same as the deterministic case: bounding the gradients along the optimization trajectory via controlling suboptimality gaps.
>
> **On the other hand, sub-Gaussian noise is necessary even under standard $L$-smoothness**. In fact, our Theorem 4.2 uses the same noise model as the previous best-known convergence under standard smoothness (Liu & Zhou, 2024). We can't adopt a weaker noise condition without deteriorating convergence rates, as the current one already matches state-of-the-art.

---

> > ### Author Rebuttal · Reviewer_12xo · 2026-04-03
> >
> > Thank you to the authors for the helpful response. I would like to follow up on two points:
> >
> > 1) In most existing literature, $(L_0, L_1)-smoothness is verified empirically along the optimization trajectory. I am not fully convinced that this is an intrinsic local/global property of the loss landscape, rather than an effect induced by the algorithmic dynamics.
> >
> > 2) While sub-Gaussian noise is necessary for obtaining this high-probability result, I wonder whether the current analysis can be extended to obtain convergence in expectation without a sub-Gaussian assumption.

---

> > > ### Author Response · Authors · 2026-04-04
> > >
> > > We sincerely thank the reviewer for the valuable follow-up and for finding our earlier response helpful. Below, we address the remaining concerns in detail.
> > >
> > > ---
> > > **Q1: About $(L_0,L_1)$-smoothness**
> > >
> > > We agree that trajectory-based verification alone does not fully prove that $(L_0,L_1)$-smoothness is an intrinsic global property of the loss landscape. Our claim is more modest: **existing evidence suggests that this phenomenon is probably optimizer-agnostic**. In particular, Riabinin et al. (2025) (as well as our own experimental runs) report similar $(L_0,L_1)$-curvature across training runs of AdamW, Adam, unScion, and Gluon, which indicates that algorithmic dynamics may have limited influence on this property.
> > >
> > > To further isolate the optimizer effect, we additionally conduct **full-batch** smoothness verification along the optimization trajectory, replacing the mini-batch-based calculation used in our rebuttal to `Reviewer 12xo Q3`. The results remain consistent: https://anonymous.4open.science/r/ICML2026_Submission8210-69C4/fig_cnn_full_batch/. We only perform this experiment for CNNs, since full-batch verification is computationally infeasible for LLMs; this is also why prior work typically verifies generalized smoothness along the trajectory.
> > >
> > > More importantly, **our main theoretical contribution does not rely on generalized smoothness being an intrinsic global characterization of practical neural networks**. Our core result is a non-Euclidean mirror-descent analysis under $\ell\*$-smoothness. In addition, _Proposition 2.6 shows that **global $\ell\*$-smoothness** and **local $(\ell,r)\*$-smoothness** are equivalent, so a **global/intrinsic property** can be translated into a **local/trajectory one**_. This further supports trajectory verification. From an optimization viewpoint, the curvature along the actual trajectory is the most relevant quantity, since it governs the algorithmic behavior, including the admissible stepsize.
> > >
> > > ---
> > > **Q2: About sub-Gaussian noise**
> > >
> > > We refer the reviewer to Appendix J, where we establish a sharp convergence guarantee for stochastic mirror descent under a distinct noise model. Specifically, we propose the generalized bounded noise model in Assumption J.1, which is neither stronger nor weaker than the standard sub-Gaussian or finite variance conditions (cf. Appendix J.3). Under this new noise model, we demonstrate a **time-uniform** high probability convergence (i.e., the bound holds for *all $t\in\mathbb N$ simultaneously*) of $\tilde O(1/\sqrt t)$ for the average-iterate $\bar x_t$ in Theorem J.3. The rate matches the classical **anytime** (i.e., the bound holds for _each $t\in\mathbb N$_, stricly weaker than time-uniform) convergence (Shamir & Zhang, 2013; Zhang et al., 2024a; Bai et al., 2025) in SCO, certifying the sharpness of our analysis. Although Theorem J.3 is stated in high probability, we can use standard probability-to-expectation tools (see, for instance, Lemma D.1 in [1]) to convert this high-probability guarantee to an in-expectation guarantee. A concrete example of such conversion can be found in Lemma B.4 in [1], where the in-expectation bound is built upon the high-probability bound.
> > >
> > > Beyond Assumption J.1, however, it is indeed difficult to derive in-expectation guarantees under strictly weaker noise assumptions. This difficulty is closely tied to the nature of generalized smoothness: it is a local property between $x_t$ and $x_{t+1}$. To exploit it, one must control $\\|x_{t+1}-x_t\\|\le\eta_t\\|g_t\\|\_*$ for all $t$ along the trajectory. This naturally leads to a chain of events that is most naturally handled through high-probability analysis [2-7], whereas purely in-expectation arguments often cannot maintain the required iterate-wise control.
> > >
> > > We would also like to remark that sub-Gaussian noise is not a very restrictive assumption. Prior work of generalized smoothness (including both $\ell$-smoothness and $(L_0,L_1)$-smoothness) also utilizes sub-Gaussian noise, or even the stronger bounded noise conditions to analyze stochastic gradient descent [2,3,4,5]. Although [6,7] establishes convergence under weaker noise assumptions, how to transfer their Euclidean-based analysis into non-Euclidean mirror descent remains open. We agree that this is an interesting direction for future work.
> > >
> > > ---
> > >
> > > **References**
> > >
> > > [1] _Efficient Stochastic Approximation of Minimax Excess Risk Optimization._ ICML'24
> > >
> > > [2] _Toward a Unified Theory of Gradient Descent under Generalized Smoothness._ ICML'25
> > >
> > > [3] _Convergence of Clipped SGD on Convex $(L_0, L_1)$-Smooth Functions._ NeurIPS'25
> > >
> > > [4] _Why Gradient Clipping Accelerates Training: A Theoretical Justification for Adaptivity._ ICLR'20
> > >
> > > [5] _Robustness to Unbounded Smoothness of Generalized SignSGD._ NeurIPS'22
> > >
> > > [6] _Convex and Non-convex Optimization Under Generalized Smoothness._ NeurIPS'23
> > >
> > > [7] _Convergence of Clipped-SGD for Convex $(L_0,L_1)$-Smooth Optimization with Heavy-Tailed Noise._ arXiv2505.20817

---

### Decision · Program_Chairs · 2026-04-30

**Decision:**

Accept (regular)

**Comment:**

The paper introduces an \ell*-smoothness as a generalization of the standard Lipschitz smoothness for non-euclidean geometries. it then establishes convergence guarantees for mirror descent and its variants and shows that they match known optimal rates. reviews are generally positive and the theoretical analysis is interesting and clean. Also the llm experiments are nice.